# 2′-*O*-Methyl-guanosine RNA fragments antagonize TLR7 and TLR8 to limit autoimmunity

Arwaf S. Alharbi [1,2,3,19], Sunil Sapkota [1,2,19] ✉, Zhikuan Zhang[4,19], Ruitao Jin[5,19], Erandi Rupasinghe[1,2], W. Samantha N. Jayasekara[1,2], Dingyi Yu[6], Mary Speir[1,7,8], Lorna Wilkinson-White[9], Liza Cubeddu[10,11], Julia I. Ellyard [12], Refaya Rezwan [1,2], Daniel S. Wenholz[7,8], Alexandra L. McAllan [1,2], Rui Gao[1,2], Le Ying[1,2], Rasan M. Sathiqu[13], Hani Hosseini Far[1,2], Josiah Bones[5], Sitong He[5], Marina R. Alexander[13], Kim A. Lennox[14], Paul J. Hertzog [1,2], Claudia A. Nold-Petry [15,16], Cameron R. Stewart[13], Carola G. Vinuesa [17], Mark A. Behlke[14], Umeharu Ohto[4,18], Olivier F. Laczka [7,8], Roland Gamsjaeger[10,11], Ben Corry [5,20], Toshiyuki Shimizu [4,20] & Michael P. Gantier [1,2,20] ✉

Recognition of RNA fragments by Toll-like receptor 7 (TLR7) and TLR8 helps to initiate the innate immune response against pathogens. An outstanding question is why RNA fragments generated during clearance of apoptotic cells fail to activate TLR7 and TLR8 signaling. Here we show that select 2′-*O*-methyl (2′-OMe) guanosine RNA fragments, including those derived from host RNAs, function as potent TLR7 and TLR8 antagonists and reduce TLR7 sensing in vivo. Mechanistically, these fragments bind to an antagonistic site on these proteins via their 5′-end 2′-OMe guanosine. These findings indicate that host RNAs evade detection because abundant ribosomal 2′-OMe-modified fragments naturally antagonize TLR7 and TLR8. Crucially, rare TLR7 and TLR8 mutations at this antagonist binding site decrease inhibition by 2′-OMe guanosine RNA fragments, leading to autoimmunity in patients. Collectively, this work redefines TLR7 and TLR8 sensing by introducing 2′-OMe guanosine as a natural immune checkpoint for their activation.

Chromosome X-encoded endosomal Toll-like receptor 7 (TLR7) and TLR8 are essential for initiating innate immune responses by detecting phagocytosed RNAs. TLR7 deficiencies increase susceptibility to severe viral infections, such as SARS-CoV-2[1], whereas rare mutations in TLR7/8 disrupt self-RNA evasion and promote autoimmunity in patients[2–4]. Critically, the mechanism distinguishing pathogenic from host RNAs remains unclear.

Initially thought to sense single-stranded RNAs (ssRNAs) as short as ~20 bases[5–7], structural studies revealed that TLR7/8 dimers bind

RNA degradation products via two distinct sites[8–11]. Site 1, conserved in both receptors, binds single nucleosides (uridine for TLR8 and guanosine or guanosine 2′,3′-cyclic phosphate [2′,3′-cGMP] for TLR7) and can also interact with small-molecule agonists like imiquimod and resiquimod[9–12]. Site 2 binds short uridine-rich motifs (for example, UUU, UG), which synergize with site 1 to enhance receptor activation[9,10]. RNase T2 and RNase 2 are critical for generating these fragments, as their absence impairs TLR7 and TLR8 activation by ssRNA and bacterial RNA[13,14].

---

Endogenous RNA modifications, particularly 2′-O-methyl (2′-OMe) on ribose, were shown to inhibit TLR7/8 sensing of ssRNAs[15,16]. Mammalian ribosomal RNA, comprising 70% to 80% of cellular RNA, contains over 100 such modifications[17], suggesting a role in preventing self-RNA recognition[15]. Bacterial transfer RNAs (tRNA) with 2′-OMe motifs (for example, Gm18), similarly antagonize TLR7 recognition[18]. However, the precise mechanism of TLR7/8 antagonism by 2′-OMe-modified RNA remains unresolved.

Here we show that 2′-OMe-modified oligonucleotides with a 5′-guanosine function as potent TLR7/TLR8 antagonists by binding a distinct site separate from sites 1 and 2. Cryo-electron microscopy (Cryo-EM) and molecular dynamics studies confirm this interaction, whereas mutations in this antagonist binding site reduce inhibition, leading to autoimmunity in patients. Thus, abundant 2′-OMe guanosine motifs in ribosomal RNA and its fragments, serve as natural immune checkpoints, preventing aberrant TLR7/8 activation. These insights not only clarify self-RNA discrimination but also highlight the therapeutic potential of synthetic 2′-OMe guanosine oligonucleotides to control TLR7/8-driven inflammation.

## Results

### Three-base-long 2′-OMe-modified oligonucleotides modulate TLR8 sensing

Short DNA and 2′-OMe-modified phosphorothioate (PS; denoted with [PS]) oligonucleotides (oligos) can potentiate TLR8 sensing, as exemplified with homopolymers of ≥8 bases of deoxythymidine (dT) (Extended Data Fig. 1a)[19,20]. To better understand TLR8 potentiation, we tested 20-mer PS gapmer antisense oligos (ASOs) with variable 5′ ends, including those with $_mU_mC$ motifs, based on our previous observations that these motifs were enriched in TLR8-potentiating oligos[21] (Fig. 1a,b and Extended Data Fig. 1b–d). These experiments revealed that the $_mU_mC$ motif was necessary but not sufficient for potentiation, as its addition to oligo #1-UC[PS], but not oligo #2-LNA-UC[PS], enhanced TLR8 sensing. In addition, $_mU_mC$ mutation or sugar-modification in the 2′-OMe oligo #660[PS] decreased TLR8 potentiation[21] (Fig. 1a,b and Extended Data Fig. 1b–d). Emphasizing the significance of the 2′-OMe region of oligo #660 for TLR8 potentiation, a 5-mer PS oligo reproducing its 5′-end 2′-OMe region (designated #660-5[PS]) was sufficient to induce potentiation of R848 sensing (Fig. 1c and Extended Data Fig. 1e).

Critically, analysis of the three possible 2′-OMe-modified 3-mer PS oligos in the 5-mer region demonstrated that the $_mU_mC_mG^{PS}$ oligo was the only one that potentiated TLR8 sensing (Fig. 1d). This potentiation by $_mU_mC_mG^{PS}$ averaged approximately twofold over a range of R848 concentrations, suggesting cooperation with site 1 activation (Fig. 1e). The analysis of 2-mer PS oligos in the 5-mer region showed that $_mC_mG^{PS}$ only somewhat potentiated TLR8 sensing but less potently that the 3-mer oligos such as $_mU_mC_mG^{PS}$ (Fig. 1f).

To further define the landscape of TLR8 modulation by short 2′-OMe oligos, we conducted an unbiased screen of all 64 possible 3-mer 2′-OMe PS oligos on TLR8 function in THP-1 and HEK TLR8 cells (Fig. 1g,h and Supplementary Table S1). Both cell lines revealed a robust TLR8-potentiating activity of 3-mer oligos restricted to a few motifs, including $_mU_mC_mG^{PS}$ and $_mC_mG_mG^{PS}$ (Fig. 1g,h and Extended Data Fig. 1f). Notably, these oligos did not have any impact on TLR8 sensing in the absence of site 1 ligands, including R848 and motolimod (Extended Data Fig. 1g). This potentiation was validated in induced pluripotent stem cell (iPSC)-derived macrophages (Fig. 1i).

Unexpectedly, a small number of 3-mer 2′-OMe-modified oligos with a $_mG_mA_mX^{PS}$ motif (X being A, U, G or C) inhibited TLR8 sensing in THP-1 and HEK TLR8 cells (Fig. 1g,h). In addition, most PS-modified DNA 3-mer oligos modestly inhibited, rather than potentiated, TLR8 sensing (Supplementary Table S1). This finding suggests that modulation of TLR8 sensing by short 3-mer oligos can result in opposing responses in a motif-dependent manner.

### 2′-OMe-modified 3-mers modulate TLR7 sensing

20-mer PS 2′-OMe-modified oligos generally inhibit TLR7 (ref. 21). Chemically modifying the sugar moiety of the three 5′-end 2′-OMe bases in a 20-mer oligo reduced human and mouse TLR7 inhibition in HEK TLR7 cells and mouse RAW264.7 macrophages, highlighting the significance of 2′-OMe bases for TLR7 antagonism (Fig. 2a and Extended Data Fig. 2a). Accordingly, an $_mG_mU_dA^{PS}$ 3-mer oligo at the 5′-end of the dC[PS] oligo (where $_d$ denotes a DNA nucleotide) significantly inhibited human and mouse TLR7, whereas other 3-mer and 2-mer oligos covering this region were less inhibitory (Fig. 2b,c and Extended Data Fig. 2b). The fully 2′-OMe 3-mer $_mG_mU_mA^{PS}$ was also a potent, dose-dependent inhibitor of human TLR7 with notable inhibition still seen at up to 5 μg/mL R848 (Extended Data Fig. 2c,d).

We next tested the effects of the panel of 64 2′-OMe 3-mer PS oligos on TLR7 inhibition. The results revealed that $_mG_mU_mX^{PS}$ 3-mer oligos were the most potent inhibitors of human TLR7, with $_mG_mU_mC^{PS}$ being the best inhibitor (Fig. 2d). 2′-OMe guanosine was the preferred 5′-end base (13 out of the 16 most inhibitory 3-mers) and, unlike for TLR8, none of the 3-mer oligos robustly potentiated TLR7 sensing (Fig. 2d and Supplementary Table S1). Notably, the most potent inhibitor of mouse TLR7 sensing was $_mG_mG_mC^{PS}$, followed by $_mG_mA_mC^{PS}$ and $_mG_mA_mG^{PS}$, whereas $_mG_mU_mC^{PS}$ inhibited signaling by less than 30%, suggesting that structural differences between human and mouse TLR7 affect the interaction with the 3-mers (Fig. 2e).

### 2′-OMe guanosine 3-mers with DNA modulate TLR7 and TLR8 sensing

The observation that $_mG_mU_dA^{PS}$ retained inhibitory activity on human TLR7 (Fig. 2c), despite having a DNA moiety, prompted the question of whether the immunomodulatory activity of the 3-mers extended beyond 2′-OMe modification noting, however, that fully PS-DNA-3-mers did not substantially inhibit TLR7 (Supplementary Table S1). One or two 2′-OMe bases were systematically replaced with DNA bases in $_mG_mU_mC^{PS}$ (best human TLR7 inhibitor), $_mG_mG_mC^{PS}$ (best mouse TLR7 inhibitor), $_mC_mG_mG^{PS}$ (best human TLR8 potentiator) and $_mG_mA_mG^{PS}$ (best human TLR8 inhibitor and a good mouse TLR7 inhibitor; Fig. 2f–i and Extended Data Fig. 2e). For TLR7 sensing, DNA moieties incorporated at the 5′-end reduced the activity of 3-mer 2′-OMe oligos (see GUC-v3/v6[PS], GGC-v3/v6[PS], and GAG-v3/v6[PS] in Fig. 2f,g, and Extended Data Fig. 2e). Similarly, whereas both TLR8-potentiating and TLR8-inhibitory 3-mers tolerated DNA bases at the third position (see $_mC_mG_dG^{PS}$ and $_mG_mA_dG^{PS}$, that is CGG-v1[PS] and GAG-v1[PS], respectively), DNA modification of their 5′-end base ablated both activities (Fig. 2h,i; see CGG-v3/v6[PS] and GAG-v3/v6[PS]). Interestingly, the DNA substitution at the 3′-end base instead increased TLR7 inhibitory activity of the GUC-v1[PS] ($_mG_mU_dC^{PS}$) (Fig. 2j). Notably, 3′-end extension of GAG-v1[PS] with three dC bases significantly increased its TLR7 antagonism while decreasing its TLR8 antagonism, suggesting key structural differences in TLR7 and TLR8 antagonism for longer oligonucleotides (Extended Data Fig. 2f).

Given that several oligos with a 5′-end 2′-OMe guanosine and two DNA bases retained potent TLR7 or TLR8-inhibitory activity (for example, $_mG_dT_dC^{PS}$, $_mG_dG_dC^{PS}$ and $_mG_dA_dG^{PS}$), a panel of 16 $_mG_mX_dX^{PS}$ and 16 $_mG_dX_dX^{PS}$ 3-mer PS oligos was also tested. Analyses of the results revealed close alignment with the initial screens, identifying $_mG_mU_dX^{PS}$ sequences as the most potent inhibitors of human TLR7, with $_mG_mU_dC^{PS}$ (GUC-v1) being the best antagonist (Fig. 2k). Similarly, $_mG_mG_dC^{PS}$ (GGC-v1), $_mG_mA_dC^{PS}$, $_mG_dG_dC^{PS}$ (GGC-v4) and $_mG_dA_dG^{PS}$ (GAG-v4) were the most potent inhibitors of mouse TLR7 (Fig. 2l). Several DNA-modified 3-mers also inhibited TLR8 sensing, including $_mG_mA_dX^{PS}$, and the double-DNA-modified $_mG_dA_dG^{PS}$ (GAG-v4) (Fig. 2m). These analyses also revealed novel potentiators of TLR8 sensing with an $_mG_dC_dX^{PS}$ motif, with $_mG_dC_dC^{PS}$ (GCC-v4) being the strongest (Fig. 2m). The potentiating activity of GCC-v4 and the inhibiting activity of $_mG_mA_dT^{PS}$ were validated in iPSC-derived macrophages, where they robustly modulated TLR8

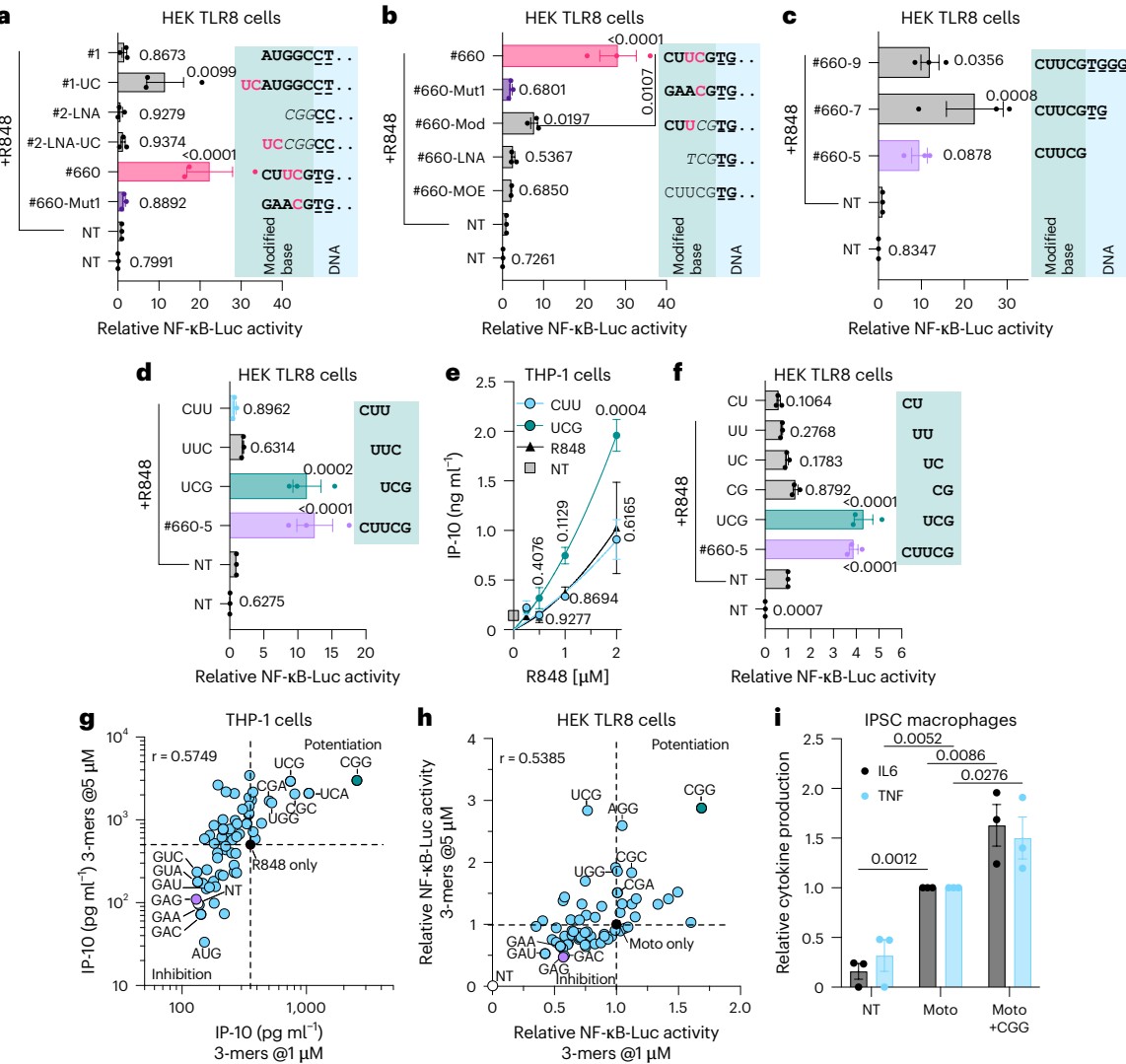

**Fig. 1 | Modulation of TLR8 sensing by 3-mer oligos. a–d,f,** HEK TLR8 cells were pretreated for ~30 min with 500 nM (**a, b**), 2 µM (c), or 5 µM (**d, f**) of the indicated oligos prior to overnight stimulation with 1 µg ml⁻¹ of R848 followed by luciferase assay. **a–d,f,** Data were background-corrected using the non-treated (NT) condition and are shown as expression relative to R848-only (± standard error of the mean (s.e.m.) and one-way analysis of variance (ANOVA) with uncorrected Fisher's LSD tests shown compared to R848-only condition (**a, b, d, f:** $P < 0.0001$; **c:** $P = 0.0034$). **b,** Unpaired two-sided *t*-test comparing #660 to #660-Mod conditions is shown. **e,** Monocytic THP-1 cells were incubated overnight with 1 µM oligo and stimulated with increasing concentrations of R848 (0.250, 0.5, 1 and 2 µg ml⁻¹) for 8 h before IP-10 ELISA (± s.e.m. and two-way ANOVA with uncorrected Fisher's LSD tests shown compared to R848-only condition). **g,** Monocytic THP-1 cells were incubated overnight with 1 or 5 µM of fully 2′-OMe-modified PS 3-mers and stimulated with 1 µg ml⁻¹ R848 for 8 h before IP-10 ELISA analysis. **h,** HEK TLR8 cells were pretreated for ~30 min with 1 or 5 µM of fully 2′-OMe-modified PS 3-mers and stimulated with 600 nM motolimod (Moto) overnight before luciferase assay. Data were background-corrected using the NT condition and are shown as expression relative to motolimod only. **i,** iPSC-derived macrophages were pretreated for 30 min with 5 µM of ₘCₘGₘG 3-mer before stimulation with 400 nM motolimod for 6 h followed by IL-6 and TNF ELISA. Cytokine levels were normalized to the motolimod-only condition (± s.e.m. and two-way ANOVA with uncorrected Fisher's LSD tests shown compared to the motolimod-only condition). **a–f,i,** Data are shown as mean of $n = 3$ independent experiments. **g,h,** Data are averaged from two or three biological replicates for each screen, and the screens at the different oligo concentrations were conducted on independent days. **a–d,f,** Oligos are modified as follows: bold pink or black is 2′-OMe, italic is LNA, non-bold is 2′-MOE. Pink highlights ₘUₘC motifs. DNA bases are underlined, and "·" denotes that the sequence is truncated. See Supplementary Table S6 for full-length sequences. All statistics are available in Source Data Fig. 1.

agonist-induced tumor necrosis factor (TNF) and/or IL-6 production (Extended Data Fig. 2g).

To confirm the relevance of our observations on sensing of known TLR7 and TLR8 RNA agonists[7,14], phosphodiester (PO) RNA9.2s^PO and ssRNA40^PO, the top 30 3-mer oligos modulating TLR7 and TLR8 were tested in healthy donor peripheral blood mononuclear cells (PBMCs). These analyses confirmed that ₘGₘAₘG^PS/ₘG_dA_dG^PS oligos strongly reduced TLR7 (IFNα) and TLR8 (TNF) sensing (Extended Data Fig. 2h,i and Supplementary Table S2). Notably, ₘGₘUₘX/ₘG_dU_dX^PS oligos selectively blunted IFNα levels with a minimal effect on TNF levels, highlighting their preferential activity on TLR7. Analysis of TLR8-selective

IL-12p70 and IFNγ levels confirmed the more selective inhibitory effect of ₘG_dA_dG^PS and related sequences on TLR8 sensing, which was much weaker for ₘGₘUₘC^PS/ₘGₘU_dC^PS oligos (Supplementary Table S2). The potentiating effect of ₘG_dC_dC^PS was also confirmed for both RNA ligands, with a preference for TNF, although this was more limited than in the context of small-molecule site 1 agonists, and ₘUₘCₘU^PS was consistently superior across TNF/IL-12p70/IFNγ levels (Extended Data Fig. 2h,i and Supplementary Table S2). Analyses of the thirty 3-mer oligo panel on TLR9 sensing in PBMCs did not reveal any significant IFNα inhibition by the 3-mers, confirming their selective activity on TLR7/8 over TLR9 (Table S2).

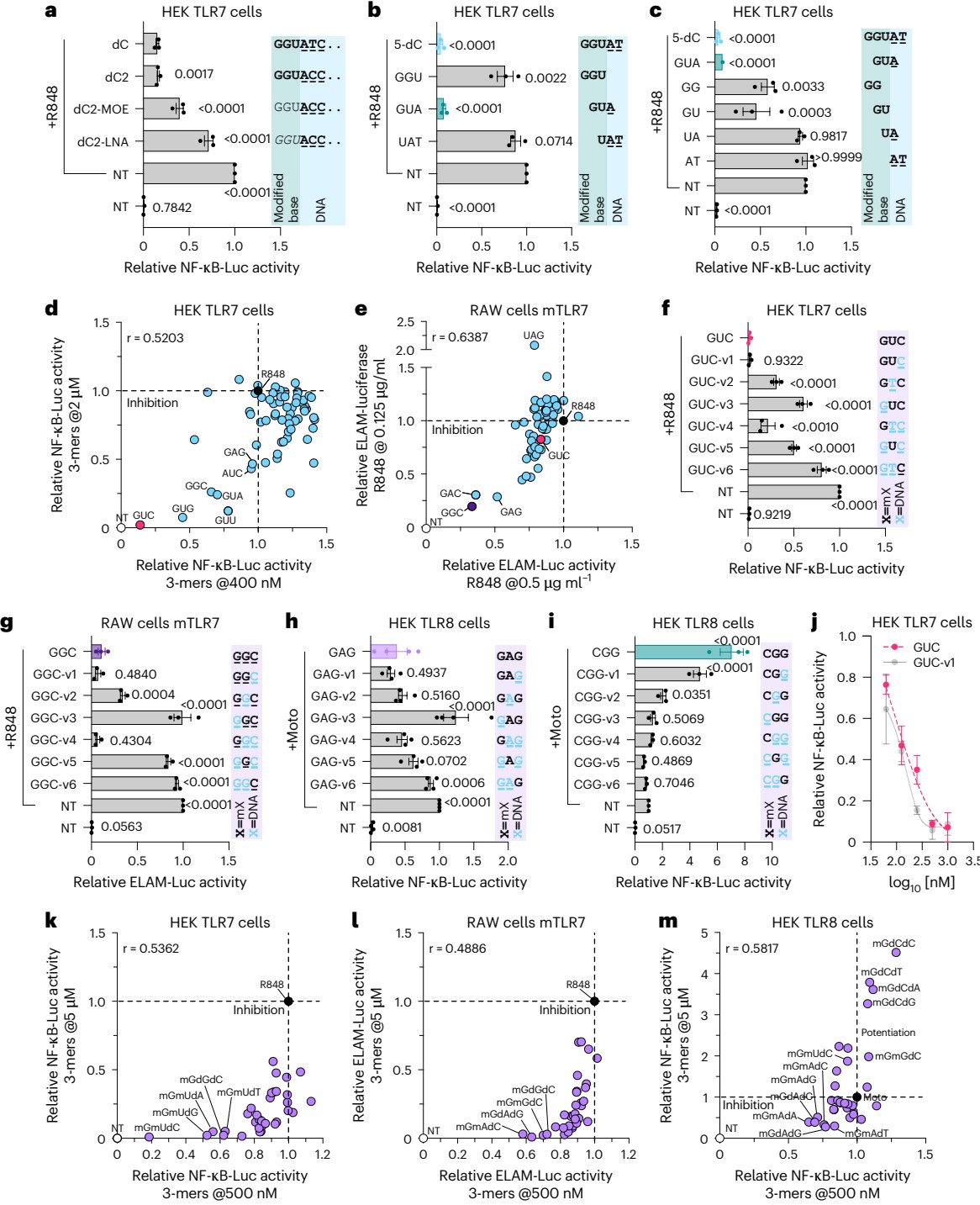

**Fig. 2 | TLR7 inhibition by 3-mer oligos. a–d,f,j,k,** HEK TLR7 cells were pretreated for ~30 min with 100 nM (**a**), 400 nM and 2 μM (**d**), 500 nM (**κ**), 5 μM (**b**, **c**, **f**, **k**) or dose-response (0.0625, 0.125, 0.25, 0.5 and 1 μM in **j**) of the indicated oligos before overnight stimulation with 1 μg ml⁻¹ R848 followed by luciferase assay. **e,g,l,** RAW-ELAM cells were pretreated with 500 nM (**l**) or 5 μM (**e**, **g**, **l**) of the indicated oligos prior to overnight stimulation with 0.5 μg ml⁻¹ (**e**) or 0.125 μg ml⁻¹ (**e,g,l**) of R848 and luciferase assay. **h,i,m,** HEK TLR8 cells were pretreated for ~30 min with 500 nM (**m**) or 5 μM (**h**, **i**, **m**) of the indicated oligos prior to overnight stimulation with 600 nM motolimod followed by luciferase assay. **a–m,** Data were background-corrected using the non-treated (NT) condition and are shown as expression relative to R848/motolimod-only conditions (± s.e.m. and one-way ANOVA with uncorrected Fisher's LSD tests shown compared to dC+R848 [**a**], R848-only [**b**, **c**], to $_m$G$_m$U$_m$C$^{PS}$ + R848 [**f**], to $_m$G$_m$G$_m$C$^{PS}$ + R848 [**g**],

to $_m$G$_m$A$_m$G$^{PS}$+Moto [**h**], or to Moto only [**i**] – **a–c**, **f–i**: $P < 0.0001$). Data are mean of $n=3$ (**a–c**, **f**, **g**, **i**, **j**) or $n=4$ (**h**) independent experiments. **d,e,k,l,m,** Data are averaged from three biological replicates for each screen, and the screens at the different oligo concentrations were conducted on independent days ($r$ values are provided on each graph, and correlation $P$ values were: **d**, **e**: $P < 0.0001$; **k**: $P = 0.0011$; **l**: $P = 0.0034$; **m**: $P = 0.0003$). **a–c**, Bold is 2′-OMe, italic is LNA, non-bold is 2′-MOE. DNA bases are underlined, and "· ·" denotes that the sequence is truncated. See Supplementary Table S6 for full-length sequences. **d,e,** Fully 2′-OMe-modified 3-mer PS oligos were assessed. **f–i,** Black bases in bold denote 2′-OMe modification and light blue bases underlined denote DNA modifications. **k–m,** $_m$G$_m$X$_d$X$^{PS}$ and $_m$G$_d$X$_d$X$^{PS}$ 3-mers were assessed. All the oligos were PS modified. All statistics are available in Source Data Fig. 2.

## Selective chiral configurations of 3-mers modulate TLR7/8 sensing

We also confirmed the capacity of GUC-v1$^{PS}$ to inhibit the human TLR7-specific agonist gardiquimod and $_mG_mG_mC^{PS}$ to inhibit gardiquimod, CL075 and ssRNA-driven activation of mouse TLR7 (Extended Data Fig. 3a–c). Similarly, $_mG_dC_dC^{PS}$ significantly potentiated TLR8 sensing of uridine in iPSC-derived macrophages, whereas $_mG_mA_mG^{PS}$ and GAG-v1$^{PS}$ significantly inhibited uridine and ssRNA-driven TLR8 sensing in HEK cells (Extended Data Fig. 3d,e). Analysis of a panel of eleven 3-mers on RNA9.2s$^{PO}$-sensing by mouse TLR7 in primary bone-marrow-derived dendritic cells (DCs) revealed that $_mG_mG_dC^{PS}$ and $_mG_dG_dC^{PS}$ had the strongest inhibitory effect on TNF production while also halving IFNα production (Extended Data Fig. 3f and Supplementary Table S2). Notably, $_mG_dA_dG^{PS}$ had the strongest inhibitory effect on IFNα, but not TNF, indicating the 3-mers may have different activities in different Flt3L-derived-DC subsets. However, none of these 3-mers significantly impacted TLR9-driven TNF production in Flt3L-derived DCs (Extended Data Fig. 3g and Supplementary Table S2). In addition, mouse TLR7 sensing of transfected bacterial RNA was also significantly inhibited by GGC-v1$^{PS}$, $_mG_dA_dG^{PS}$ (GAG-v4$^{PS}$) and GAG-v1$^{PS}$ (Extended Data Fig. 3h).

The activity of the mouse TLR7 inhibitory sequence GGC-v1$^{PS}$ was also tested on primary bone-marrow-derived macrophages (BMDMs) derived from *Tlr7$^{Y264H}$* mutant mice, which constitutively engage TLR7 via an increased affinity for guanosine[3]. Overnight treatment of *Tlr7$^{Y264H}$* mutant BMDMs with GGC-v1$^{PS}$ significantly down-regulated 20 out of the 22 genes that were down-regulated by Enpatoran[22] (Fig. 3a,b). Several genes confirmed to be significantly down-regulated by both inhibitors were previously reported as top imiquimod-induced genes in a mouse model of psoriatic-like skin inflammation (for example, *Slc13a3, Fpr1, Fpr2, Cd300e*)[23] (Fig. 3c).

To assess the direct interaction of the lead 3-mer oligos on recombinant human TLR8 and *Macaca mulatta* TLR7 (mmTLR7), surface plasmon resonance (SPR) was used. SPR analyses showed GUC-v1$^{PS}$ had an average $K_D$ of 5.6 μM to mmTLR7, whereas the weaker TLR7 inhibitor GAG-v1$^{PS}$ bound to mmTLR7 with an average $K_D$ of 20.2 μM (Fig. 3d, Extended Data Fig. 3i and Supplementary Tables S3a,b). GUC-v6$^{PS}$ ($_dG_dT_mC^{PS}$) and GCC-v4$^{PS}$ showed negligible binding to TLR7. Conversely, both GAG-v1$^{PS}$ and GCC-v4$^{PS}$ bound to human TLR8, with averaged $K_D$ values of 4 μM and 8 μM, respectively, whereas GUC-v6$^{PS}$ had negligible binding (Fig. 3e,f and Supplementary Tables 3a,b). The SPR binding profiles of GAG-v1$^{PS}$ and GCC-v4$^{PS}$ to human TLR8 differed substantially (on and off rates), indicative of a different TLR8 binding profile (Fig. 3f).

Importantly, all the oligos tested above were synthesized using PS internucleotide linkages. Unlike natural achiral PO linkages, the two chiral PS internucleotide linkages in the 3-mers were synthesized in a stereo-random fashion, leading to a mixture of four stereoisomers. Stereopure 3-mer oligos of GUC-v1$^{PS}$ and GAG-v1$^{PS}$ were synthesized with the four possible PS configurations (referred to as RR, RS, SR and SS) to test their effect on TLR7/8 antagonism. The RR and RS configurations of GUC-v1$^{PS}$ and GAG-v1$^{PS}$ displayed significantly less inhibition of human TLR7 than the SR and SS variants (Fig. 3g,h). Conversely, TLR8 inhibition by GAG-v1$^{PS}$ was significantly less with the SR and SS stereoisomers compared to the RR and RS configurations (Fig. 3i). Finally, having observed that GUC-v1$^{PS}$ acted as a mild potentiator of TLR8 (Fig. 2m), its stereoisomers were tested on TLR8 sensing, which revealed that the RS and SS configurations blunted potentiation (Fig. 3j). These results supported that TLR8 potentiation and inhibition relate to different configurations of the oligos required for activity (RR and RS for inhibition with GAG-v1$^{PS}$, and RR or SR for potentiation with GUC-v1$^{PS}$).

## 2′-OMe 3-mers inhibit TLR7 through its antagonist binding site

Reported crystal structures of mmTLR7-RNA complexes indicate the presence of a conserved RNA binding site (site 2) near the dimerization interface, where binding of short RNA fragments, including $_rG_rU_rC_rC_rC$, encourages the active form of the dimer (Fig. 4a)[11]. Interestingly, the first three bases of $_rG_rU_rC_rC_rC$ RNA are the only ones that directly form interactions with the receptor (Fig. 4b,c and Extended Data Fig. 4a). We next investigated whether $_mG_mU_mC^{PS}$ inhibited TLR7 via an increased affinity to site 2. In silico CpHMD analyses revealed that, whereas the truncated 3-mer GUC RNA could stably bind to TLR7 site 2, the 2′-OMe $_mG_mU_mC$ analogue did not remain stably bound to this site. Specifically, the central $_mU$ base retreated from the conserved TLR7 binding pocket due to a steric clash of the 2′-OMe group with the protein (Fig. 4d). This led to a large movement away from site 2, as seen with the molecule root-mean-square deviation (RMSD) analysis over time and the decreased intermolecular interactions of the 2′-OMe uridine with all the associated TLR7 residues (Fig. 4e and Extended Data Fig. 4b,c). Therefore, the molecular dynamics simulation suggested that the presence of a 2′-OMe uridine group in $_mG_mU_mC^{PS}$ or GUC-v1$^{PS}$ was detrimental to the interaction with TLR7 site 2.

Cryo-EM analysis of the mmTLR7 ectodomain was performed in the presence of GUC-v1$^{PS}$ (mixture of RR, RS, SR and SS configurations) and the structure of the TLR7/GUC-v1$^{PS}$ complex was solved with an overall resolution of 3.0 Å (Fig. 4f and Supplementary Table S4). The TLR7/GUC-v1$^{PS}$ complex formed a *C*2 symmetric open-form dimer, similar to the previously reported small-molecule antagonist-bound TLR7 structures[24] (Extended Data Fig. 4d). Densities for GUC-v1$^{PS}$ were clearly observed at the antagonist binding sites between two TLR7 protomers (Fig. 4f). Unlike the closed form of the agonist-bound TLR7 dimer, the two TLR7 protomers in the open form are separated at the C termini, which hinders the proximity of the intracellular TIR domains for activation, thus representing an inhibited state (Extended Data Fig. 4d). Although the cryo-EM map may represent the average densities for the mixture of GUC-v1$^{PS}$ stereoisomers, each PS stereoisomer could also be reasonably fitted to the cryo-EM map (Extended Data Fig. 4e). Figure 4g shows the structures of the RR and SS stereoisomers, which are essentially identical in terms of the recognition of the modified nucleotide, with minor variations observed only at the PS linkage portion (Fig. 4g,h and Extended Data Fig. 4f,g). Hereafter, the representative TLR7/GUC-v1$^{PS}$-*SS* complex structure is described because of the relatively stronger inhibitory activity of GUC-v1$^{PS}$-*SS* (Figs. 3g and 4h).

The 5′-end 2′-OMe guanosine ($_mG_1$) of GUC-v1$^{PS}$ deeply inserts into the antagonist binding site and makes extensive contacts with TLR7 (Fig. 4h). The guanine moiety is stacked by F351$^A$ and F507$^B$ and is surrounded by bulky aromatic residues, including Y264$^A$, F408$^A$ and F506$^B$. The guanine N1 amino, C2 amino and C6 carbonyl groups form hydrogen bonds with E352$^A$ and V355$^A$ main-chain O atoms, and with the Q354$^A$ main-chain N atom, respectively. The intimate contacts and hydrogen bonding pattern explain the preference for a guanine base at this position. The 5′-OH group of $mG_1$ also forms hydrogen bonds with the F506$^B$ backbone carbonyl and S530$^B$ backbone amine groups. In addition, the modified 2′-OMe group of $mG_1$ points to a small hydrophobic patch formed by the F351$^A$, V381$^A$ and F408$^A$ side chains, strengthening the interactions. These structural features are in agreement with the stronger inhibitory effect of the 3-mer oligos with a 2′-OMe guanosine at the 5′-end (Fig. 2d). For the phosphate backbone, the first PS group forms hydrogen bonds with N265$^A$ and S530$^B$, and the second PS group forms weak electrostatic interactions with R553$^B$ and H578$^B$. Compared to the stringent recognition of $mG_1$, the following $_mU_2$ and $_dC_3$ are loosely recognized. The two pyrimidine rings successively stack onto the F349$^A$ side chain and are also surrounded by the P267$^A$ and F268$^A$ side chains on the opposite side, thereby occupying the entrance of the antagonistic site. Additionally, the N3 amino group of $_mU_2$ and the C4 amino group of $_dC_3$ form hydrogen bonds with the Q323$^A$ side-chain and R262$^A$ and Y264$^A$ main-chain O atoms, respectively. The 2′-OMe group of $_mU_2$ is oriented toward the solvent and positioned between the F349$^A$ side chain and the ribose of $_dC_3$. Notably, alchemical

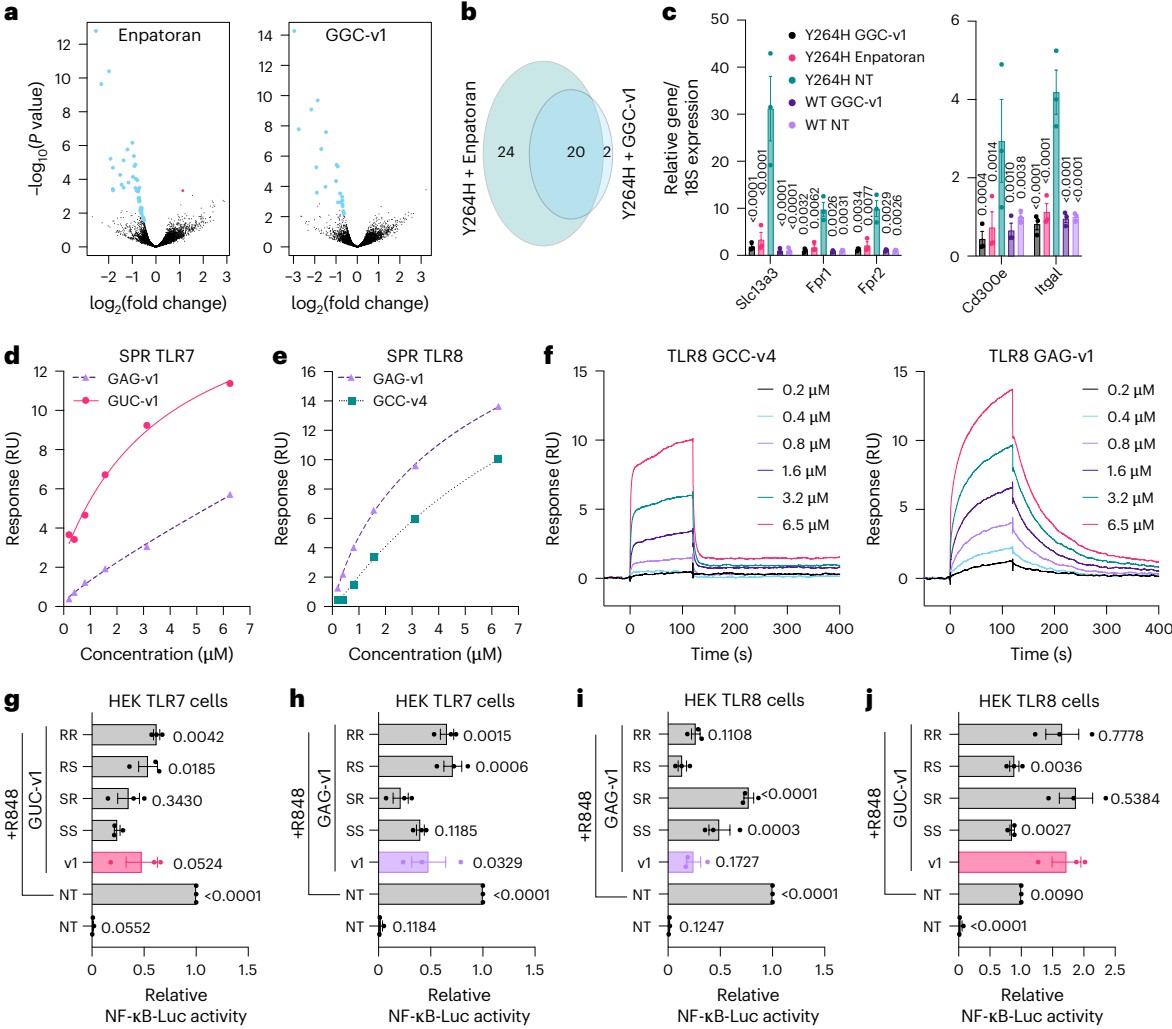

**Fig. 3 | 3-mer oligos bind to TLR7/8 to modulate their function. a–c**, BMDMs from *Tlr7^{Y264H}* mice were stimulated for 24 h with 5 μM GGC-v1 or 100 nM Enpatoran before RNA purification for RNA sequencing (**a,b**) or RT-qPCR analyses (**c**). **a**, Volcano plot of the genes significantly impacted compared to non-treated (NT) condition (blue are down-regulated and red is upregulated) were compared between GGC-v1 and Enpatoran treatments (**b**). **c**, RT-qPCR analyses of *Slc13a3/18S*, *Fpr1/18S*, *Fpr2/18S*, *Cd300e/18S* and *Itgal/18S* in RNA lysates from primary BMDMs from three independent *Tlr7^{Y264H}* and wild-type (WT) mice. Data are shown relative to the NT condition from WT mice (± s.e.m. and two-way ANOVA with uncorrected Fisher's LSD tests shown compared to the NT *Tlr7^{Y264H}* condition). **d–f**, Surface plasmon resonance (SPR) analyses of recombinant monkey TLR7 (**d**) and human TLR8 (**e,f**) with the indicated concentrations of 3-mers. Data shown are representative of five

or six independent analyses (Supplementary Table S3). **g–j**, HEK TLR7 cells (**g,h**) or HEK TLR8 cells (**i,j**) were pretreated with 200 nM (**g**) or 5 μM (**h–j**) of _mG_mU_dC^{PS} or _mG_mA_dG^{PS} 3-mers synthesized as stereopure isomers of RR, RS, SR or SS configurations, before overnight stimulation with 1 μg ml^{−1} of R848 followed by luciferase assay. Non-stereopure oligos were included as controls (shown as "v1" conditions). Data were background-corrected using the NT condition and are shown as expression relative to the R848-only condition (± s.e.m. and one-way ANOVA with uncorrected Fisher's LSD tests shown compared to GUC-v1-SS + R848 (**g**), GAG-v1-SR + R848 (**h**), GAG-v1-RS + R848 (**i**) or GUC-v1 + R848 (**j**); **g–j**: *P* < 0.0001). **g–j**, Data are shown as mean of *n* = 3 independent experiments. All the oligos were PS modified. All statistics are available in Source Data Fig. 3.

free energy perturbation calculation indicates that the relative binding energy difference for _rG_rU_dC^{PO} compared to _mG_mU_dC^{PO} to this site of TLR7 is about 16 kJ mol^{−1}, which is equal to an ~550-fold change in $K_D$ compared to _mG_mU_dC. This finding supports that normal RNA molecules cannot compete with 2′-OMe RNA molecules at the antagonist binding site (Extended Data Fig. 4h).

## 2′-OMe 3-mers modulate TLR7 sensing in vivo

We next investigated the capacity of GGC-v1^{PS} to antagonize mouse TLR7 sensing of R848 in vivo. Prophylactic intravenous (i.v.) administration of GGC-v1^{PS} complexed with the commercial polycationic agent in vivo-jetPEI significantly decreased the splenic induction of several key nuclear factor kappa B (NF-κB) targets driven by intraperitoneal (i.p.) injection of R848 (for example, *Tnf*, *Il6* and *Il10*),

leading to a significant decrease in circulating TNF protein levels in the sera of WT mice (Fig. 5a,b). Similarly, pre-treatment of the skin of mice with GGC-v1 formulated in 30% F127 Pluronic gel significantly reduced a TLR7-dependent gene signature driven by repeated topical administration of Aldara cream containing imiquimod (including pro-inflammatory *Tnf*, *Cxcl1* and *Il17*, and specific genes reported to be induced in this model; for example, *Fpr1* and *Slc13a3* (ref. 23)) (Fig. 5c). This reduction in TLR7-driven gene expression in the skin was partially dose-dependent and concurrent with a significant decrease in CD45^+ immune infiltrates in the skin and overall decreased skin redness and scaliness (Fig. 5d and Extended Data Fig. 5a,b). Importantly, pre-treatment of the skin with GGC-v1^{PS} did not alter the splenomegaly seen in this model, suggesting that its anti-inflammatory effect on TLR7 was primarily localized to the skin (Extended Data Fig. 5c,d).

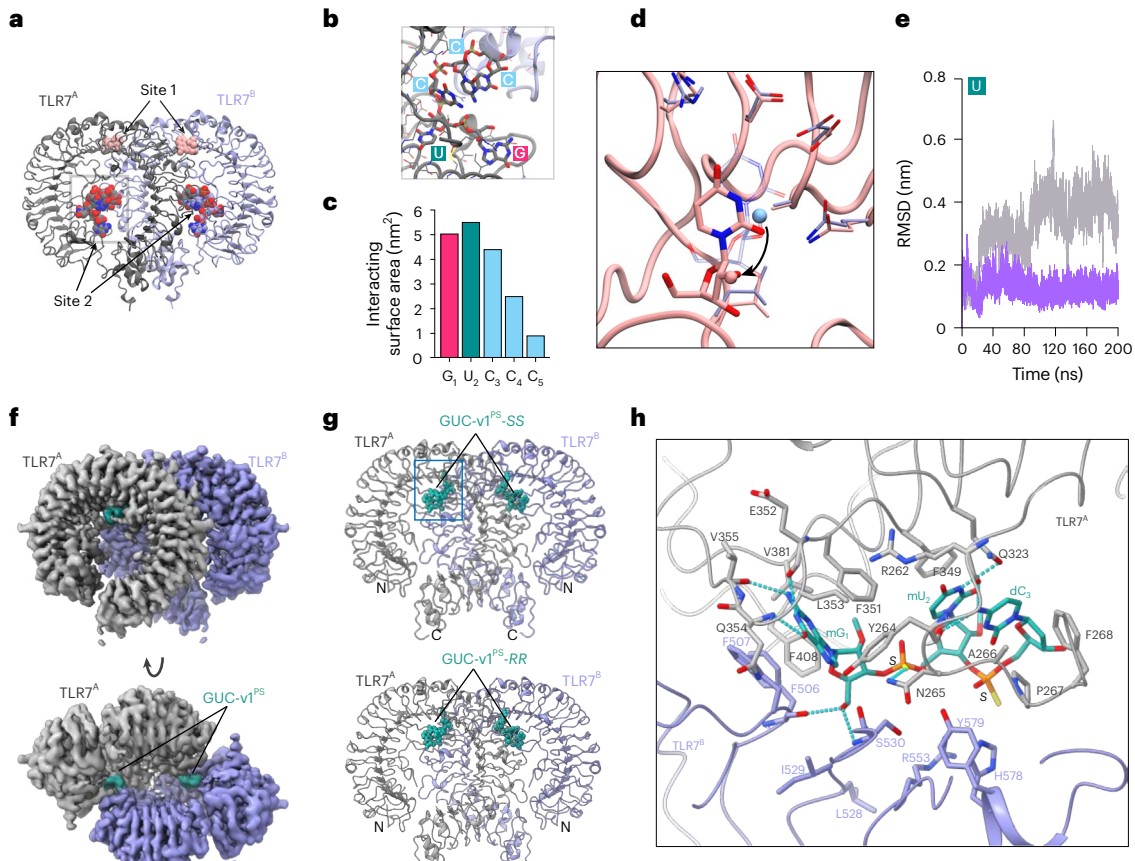

**Fig. 4 | 3-mer oligos bind to an antagonist binding site in TLR7. a,b,** Crystal structure of TLR7 dimer in complex with IMQD at site 1 (pink ball representation) and $_rG_rU_rC_rC_rC^{PO}$ motif at site 2 (pink) (zoom view in **b** with details of the RNA binding site shown (PDB: 5ZSE)). The two protomers are colored in dark gray and light blue. **c,** Molecular surface area of each nucleotide interacting with protein residues forming the binding site 2, the first three nucleotides contribute the majority of molecular surface ( ~ 80%). **d,** The retreat of uracil in $_mG_mU_mC^{PO}$ from the conserved binding pocket is shown in solid licorice as per MD simulation at pH 5. Conversely, the native uracil from $_rG_rU_rC^{PO}$ in the binding pocket is shown in transparent licorice. The protein ribbon and side chains are shown in pink in $_mG_mU_mC^{PO}$ simulations, and in light blue in $_rG_rU_rC^{PO}$ simulations. The location of the uracil 2′-residues are indicated by a sphere in each structure. **e,** Root mean square deviation (RMSD) of $_rG_rU_rC^{PO}$ (purple) and $_mG_mU_mC^{PO}$ (gray) versus time shows that the methylated version moves away from the $_rG_rU_rC$ binding site, indicating the introduced 2′-O-methyl moieties on sugar backbones destabilize the binding of $_mG_mU_mC^{PO}$, compared with native $_rG_rU_rC^{PO}$. **f,** Unsharpened cryo-EM map of the TLR7/GUC-v1$^{PS}$ complex shown as surface representations. Densities for the two TLR7 protomers and GUC-v1$^{PS}$ are respectively colored gray, purple and green. **g,** Overall structure of the TLR7/GUC-v1$^{PS}$-SS and TLR7/GUC-v1$^{PS}$-RR complexes. Two TLR7 protomers and GUC-v1$^{PS}$-SS and GUC-v1$^{PS}$-RR are shown in cartoon and ball-stick representations, respectively. Color schemes are the same as in **f. h,** Close-up view of GUC-v1$^{PS}$-SS recognition at the antagonistic site. Residues (within 4.5 Å from the ligand) are shown in stick representations and are colored by atom, with the N, O, S and P atoms colored by blue, red, yellow and orange, respectively. Dashed lines in cyan indicate hydrogen bonds (cutoff distance <3.5 Å).

Collectively, these results established the capacity of GGC-v1$^{PS}$ to antagonize TLR7 sensing of R848 and imiquimod in vivo.

To determine whether 3-mer oligos inhibited RNA sensing by TLR7 in vivo, we studied the activity of GGC-v1$^{PS}$ co-administered with 5′-capped T7-synthesized Firefly luciferase (Fluc) mRNA containing unmodified uridine using FDA-approved ALC-0315-based lipid nanoparticles (LNPs) (Methods)[25]. Following validation that GGC-v1$^{PS}$ could be co-packaged with mRNA molecules in LNPs (Extended Data Fig. 5e), mice were injected i.v. with LNPs containing Fluc mRNA with or without GGC-v1$^{PS}$. Although the co-delivery of GGC-v1$^{PS}$ did not decrease the expression of Fluc mRNA in the liver, it halved the production of many key pro-inflammatory cytokines in the sera (for example, IFNα, IFNγ, IL-6, IL-10, IL-12p40, MCP1, MIP1α/β and CCL5) (Fig. 5e–g and Extended Data Fig. 5f). This finding is consistent with the concept that the reactogenicity of unmodified mRNAs is at least partially dependent on TLR7 (ref. 15) and indicates that the GGC-v1$^{PS}$ oligo is capable of dampening TLR7 activation in response to natural ligands in vivo. Taken together, these findings demonstrate the capacity of synthetic 2′-OMe 3-mer oligos to functionally modulate TLR7 in the animal models tested.

## RNAs containing select 2′-OMe motifs are natural antagonists of TLR7/8

Given the essential role of $_mG_1$ in the interaction of GUC-v1$^{PS}$ with TLR7 (Fig. 4h), we reasoned that select $_mG_rX_rX$ motifs occurring naturally in endogenous RNA molecules were likely to modulate TLR7/8 sensing. Focusing on 5′-$_mG$ 3-mers, we screened a panel of 16 $_mG_rX_rX^{PS}$ oligos on human TLR7 and TLR8 sensing (Fig. 6a). Similar to our observations with fully 2′-OMe-modified PS 3-mers, $_mG_rU_rC^{PS}$ was one of the most potent human TLR7 inhibitors, whereas $_mG_rA_rG^{PS}$ was a strong inhibitor of TLR8 (Fig. 6a and Supplementary Table S1). Moreover, additional inhibitors for both TLR7 ($_mG_rG_rA^{PS}$) and TLR8 ($_mG_rA_rA^{PS}$) were identified (noting $_mG_rA_rA^{PS}$ was the most potent TLR8 inhibitor). Eight of 16 oligos inhibited both receptors by more than 25%.

Using transfected $_mG_rU_rC^{PO}$ and $_mG_rA_rA^{PO}$ oligos, we confirmed that oligos with a natural PO backbone could also significantly antagonize TLR7/8 sensing of R848 (Fig. 6b), although this activity was milder than PS-modified oligos. We attributed the lower potency of these PO oligos to intracellular degradation, which was supported by a time-dependent selective increase in $_mG$ levels following transfection

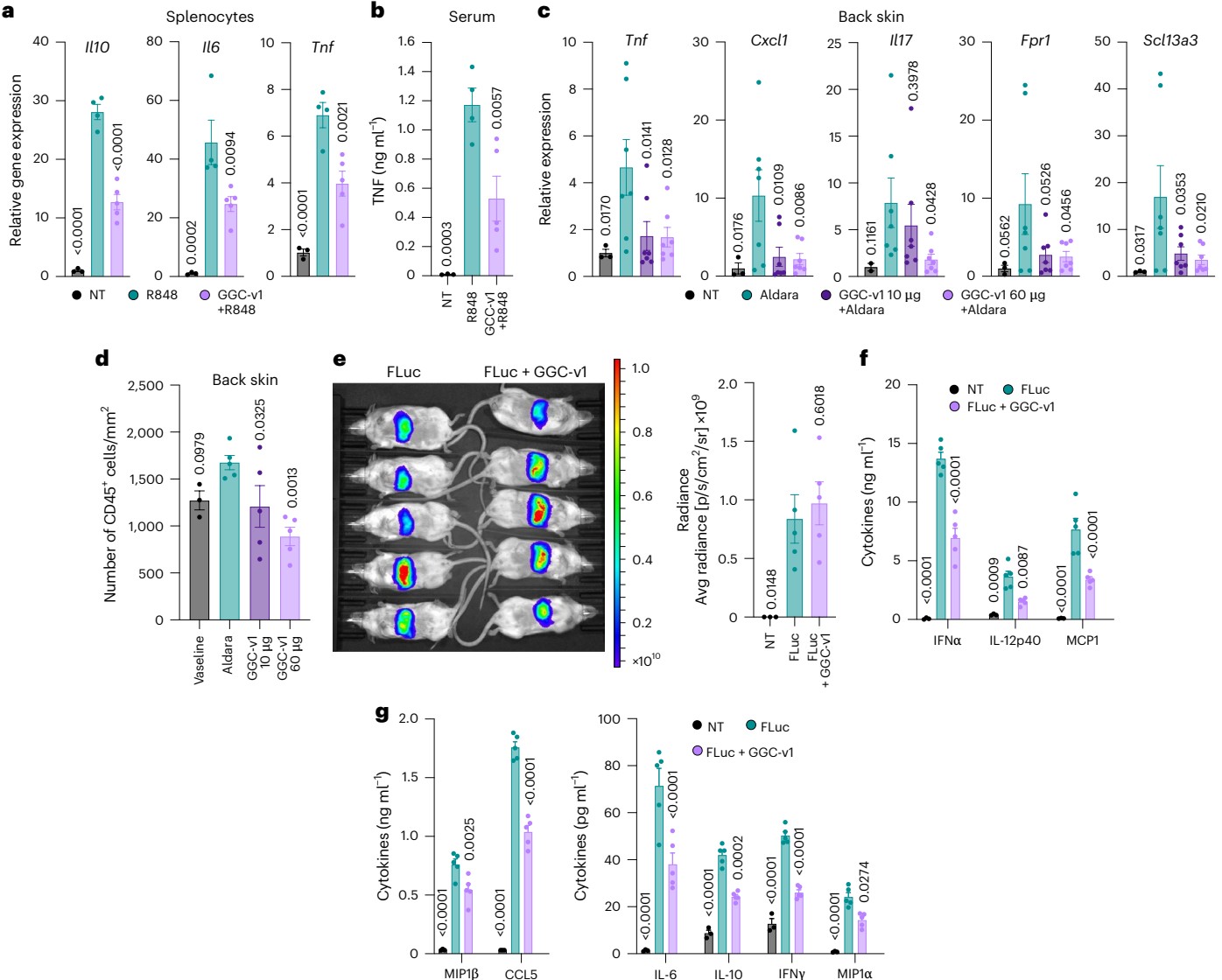

**Fig. 5 | 2′-OMe 3-mer oligos antagonize TLR7 function in vivo. a,b**, WT C57/BL6 mice were injected i.v. with 200 µg GGC-v1^PS complexed with *in vivo*-jetPEI for 1 h before i.p. injection of 25 µg R848 for 2 h before collection of spleens (**a**) and sera (**b**). **a**, RT-qPCR analyses of *Tnf/Gapdh*, *Il6/Gapdh* and *Il10/Gapdh* from spleen lysates; data are reported relative to the non-treated (NT) condition. **b**, TNF levels were quantified by LegendPlex assay. **a,b**, Mean of *n* = 3 NT mice, *n* = 4 R848 mice and *n* = 5 R848 + GGC-v1^PS mice are shown (± s.e.m. and one-way ANOVA with uncorrected Fisher's LSD tests shown compared to R848-only group; **a**: Tnf/ Il10 P = 0.0001; **a**: Il6 *P* = 0.0006; **b**: *P* = 0.0009). **c,d**, Aldara cream was applied topically to the back of WT C57/BL6 mice directly following, or not (*n* = 7 mice), application of 10 µg (*n* = 7 mice) or 60 µg (*n* = 7 mice) GGC-v1^PS formulated in F127 Pluronic gel for 4 days. Non-Aldara treated control mice received Vaseline (*n* = 3 mice). Mice were humanely euthanized and the back skin collected for RNA purification (**c**) or histology (**d**). **c**, RT-qPCR analyses of indicated genes reported to that of 18S expression, relative to NT mice. Mean of *n* = 3 (Vaseline) and *n* = 7 (all other groups) mice/group (± s.e.m. and one-way ANOVA with

uncorrected Fisher's LSD tests shown compared to Aldara-only group; **c** Tnf *P* = 0.0247; Cxcl1 *P* = 0.0194; Il17 *P* = 0.1539; Fpr1 *P* = 0.1055; Slc13a3 *P* = 0.0535). Data are representative of three independent experiments. **d**, CD45^+ positive cells in the back skin were quantified by fluorescent histology (Methods). Mean of *n* = 3 (Vaseline) and *n* = 5 (all other groups) mice/group are shown (± s.e.m. and one-way ANOVA with uncorrected Fisher's LSD tests shown compared to Aldara-only group; *P* = 0.0113). **e–g**, WT 129×1/SvJ mice were injected i.v. with LNPs containing 20 µg Fluc mRNA alone (*n* = 5 mice) or 17.5 µg Fluc mRNA conjugated to 2.5 µg GGC-v1^PS (*n* = 5 mice) (Methods). *n* = 3 mice were not treated (NT). **e**, IVIS measurement of radiance was conducted at 6 h after LNP injection and 3–5 min after injection of d-luciferin potassium in all mice, and 6 h sera were collected for multiplex cytokine analyses (**f,g**). **e–g**, Mean of *n* = 3 (NT) and *n* = 5 (all other groups) mice/group (± s.e.m. and one-way (**e**) or two-way (**f,g**) ANOVA with uncorrected Fisher's LSD tests shown compared to Fluc-only LNP group; **e**: *P* = 0.0167). All statistics are available in Source Data Fig. 5.

of a 4-mer $_m$G$_r$A$_r$A$_r$A$^{PO}$ oligo, indicative of its rapid complete nuclease processing (Extended Data Fig. 6a). Critically, we determined the cryo-EM structure of TLR7 in complex with $_m$G$_r$A$_r$A$^{PS}$ and $_m$G$_r$U$_r$C$^{PO}$, representing more natural 2′-OMe RNA fragments, at a resolution of 2.7 Å and 2.9 Å, respectively (Fig. 6c–f, Extended Data Fig. 6b–d and Supplementary Table S4).

Densities of $_m$G$_r$A$_r$A$^{PS}$ and $_m$G$_r$U$_r$C$^{PO}$ were clearly observed at the antagonist binding site of TLR7 (Extended Data Fig. 6e). Similar to

GUC-v1$^{PS}$, $_m$G$_r$A$_r$A$^{PS}$ also represented a mixture of four PS stereoisomers. We modeled the two representative *SS* and *RR* stereoisomers and focused on the TLR7/$_m$G$_r$A$_r$A$^{PS}$-*SS* complex. In both structures complexed with $_m$G$_r$A$_r$A$^{PS}$-*SS* and with $_m$G$_r$U$_r$C$^{PO}$, TLR7 formed dimers in the open conformation stabilized by the $_m$G$_r$X$_r$X 3-mers interacting with the antagonist binding site, in a manner similar to the TLR7/GUC-v1$^{PS}$ complex (Extended Data Fig. 4d). The overall mode of recognition for $_m$G$_r$A$_r$A$^{PS}$-*SS* and $_m$G$_r$U$_r$C$^{PO}$ is generally similar to GUC-v1$^{PS}$ (Fig. 6e–g and

Extended Data Fig. 6d). The 5′-end $_mG_1$ is recognized by TLR7 through three major types of interactions in the same manner as the $_mG_1$ of GUC-v1$^{PS}$, including (1) compact hydrophobic packing around the guanine moiety (with Y264$^A$, F351$^A$, F408$^A$, F506$^B$ and F507$^B$), (2) key hydrogen bond networks around the guanine functional groups and the main-chain atoms of E352$^A$, Q354$^A$, V355$^A$ and T406$^A$, as well as the 5′-OH group with carbonyl group of F506$^B$ and (3) the conserved hydrophobic interactions originating from the particular 2′-OMe of $_mG_1$ and the side chains of F351$^A$, V381$^A$ and F408$^A$ (Fig. 6e,f). These conserved interactions observed for each $_mG_1$ in complexes of TLR7 with GUC-v1$^{PS}$, $_mG_rA_rA^{PS}$ and $_mG_rU_rC^{PO}$ highlight the key role played by 2′-OMe guanosine at the 5′-end of these sequences, particularly visible on the overlay of the three structures (Fig. 6g). On the other hand, the positions of the second and third nucleotides, including the PS or PO linkages, are slightly shifted. The purine rings of $_rA_2$ and $_rA_3$, as well as the pyrimidine rings of $_rU_2$ and $_rC_3$ are similarly stacked with each other and occupy the entrance of the antagonist binding site. Additionally, S530$^B$, R553$^B$, and H578$^B$ side chains form hydrogen bonds or electrostatic interactions with the PS and phosphate groups. As in the TLR7/GUC-v1$^{PS}$ complex, the recognition of the second and third nucleotides of $_mG_rA_rA^{PS}$ and $_mG_rU_rC^{PO}$ is less extensive, indicating the compatibility to accommodate different nucleotides at these positions.

Strikingly, akin to the interaction with GUC-v1$^{PS}$, the guanine moiety of the natural $_mG_rU_rC^{PO}$ and $_mG_rA_rA^{PS}$-$SS$ fragments underwent aromatic stacking with F507$^B$, indicative of an essential role for F507 in antagonism of TLR7 by natural 2′-OMe RNA fragments, and consistent with recent reports of F506 and F507 gain of function (GOF) mutations in SLE patients[2,3,26]. Accordingly, in vitro SPR analyses of F506L, F507L and F507S recombinant mutants of mmTLR7 confirmed the critical interactions between residues F506/F507 and the guanosine moiety of our oligos; all three mutations decreased binding to the $_mG_rU_rC^{PS}$ oligo by disrupting the F507 aromatic stacking interaction, with F507S being the most deleterious (Fig. 6h and Supplementary Table S3c). These binding results are directly concordant with our independent in silico analyses of the influence of the mutations on the binding free energy of GUC-v1$^{PS}$ and $_mG_rA_rA^{PS}$, obtained from alchemical free energy perturbation calculations (Extended Data Fig. 6f,g). In these in silico assays, the affinity of both GUC-v1$^{PO}$ and $_mG_rA_rA^{PO}$ was reduced by more than 200-fold for F507S and 10-fold for F507L compared to WT (Fig. 6i). Functionally, the antagonistic activity of GUC-v1$^{PS}$ on

gardiquimod sensing by TLR7 was blunted in cells transiently expressing the GOF TLR7 F507S variant, but not the TLR7 L528I mutant (Fig. 6j). Similarly, although cooperative sensing of a site 2 ligand RNA (RNA9.2$^{PS}$ (refs. 10,12)) combined with guanosine was seen with WT, F507S, and L528I expression, the F507S mutant was the only variant where antagonism by GUC-v1$^{PS}$ was impaired (Fig. 6k and Extended Data Fig. 6h). Antagonism of poly(dT)$_{20}$ was also impaired in F507S mutant cells, confirming that DNA molecules can also engage with the antagonist binding site of TLR7 (Extended Data Fig. 6i).

Distal mutation at residue P435 also impacted the binding of TLR7 antagonists, as evidenced in our SPR assays with the P435S recombinant mutant, which displayed robust decreased binding to the $_mG_rU_rC^{PS}$ oligo (Fig. 6h and Supplementary Table S3c). We predict this relates the interaction of P435 with F506 and F507, indirectly affecting its stacking with the guanine residue (Extended Data Fig. 6j). We also investigated the impact of residues S530 and T406, which are close to the $_mG_1$. The S530A and T406S mutant proteins instead increased binding to the $_mG_rU_rC^{PS}$ oligo by SPR, indicating their important role in forming the antagonistic site (Fig. 6h and Supplementary Table S3c).

Noting the conservation of amino acids around residues F506/F507 of TLR7 with TLR8 (aligned to position F494/F495) (Extended Data Fig. 6k), and based on prior characterization of the structure of TLR8 in complex with the small-molecule TLR8 antagonist CU-CPT8m[27] (Extended Data Fig. 6l), we posited that the F494/F495 residues were also involved in TLR8 antagonism by 2′-OMe RNA fragments. Although the structure of TLR8 in complex with GAG-v1$^{PS}$/$_mG_rA_rA^{PS}$ could not be resolved, we successfully docked $_mG_rA_rA^{PO}$ in the TLR8 antagonist binding site of an inactive dimer structure[28], with $_mG_1$ forming direct interactions with F494/F495, and with conserved interactions with the 2′-OMe as seen for TLR7 (Fig. 6l and Extended Data Fig. 6l). In vitro SPR analyses of F495S TLR8 recombinant protein, mimicking the F507S mutant of TLR7, entirely ablated measurable binding to $_mG_rA_rA^{PS}$, whereas the I403S mutation reduced binding by approximately fourfold (Fig. 6m and Supplementary Table S3d). This aligns with the predicted role of F495 forming an aromatic stacking interaction with guanine, and with the prediction that I403S would decrease this interaction by reducing hydrophobic interactions with the 2′-OMe moiety of $_mG_1$ (Fig. 6l). Accordingly, the TLR8 F494L GOF mutation reported in a neutropenic patient[4] was resistant to GAG-v1 antagonism upon R848 activation of TLR8, unlike another TLR8 GOF mutation G572D (Fig. 6n).

**Fig. 6 | 2′-OMe guanosine RNA fragments act as natural antagonists of TLR7 and TLR8. a**, HEK TLR7 cells (x-axis) and HEK TLR8 cells (y-axis) were pretreated with 2 μM $_mG_rX_rX^{PS}$ 3-mers before overnight stimulation with 1 μg ml$^{-1}$ R848 followed by luciferase assay. Data were background-corrected using the non-treated (NT) condition and are shown as expression relative to the R848-only condition. **b**, HEK TLR7 and HEK TLR8 cells were transfected with 2 μM (for TLR7 cells) or 5 μM (for TLR8 cells) indicated oligo with DOTAP before overnight stimulation with 1 μg ml$^{-1}$ R848 followed by luciferase assay. Data were background-corrected using the non-treated (NT) condition and are shown as expression relative to the R848-DOTAP condition (± s.e.m. and two-way ANOVA with uncorrected Fisher's LSD tests shown compared to the DOTAP + R848 condition). **c,d**, Unsharpened cryo-EM map (**c**), and structure (**d**) of the TLR7/$_mG_rU_rC^{PO}$ complex. The two TLR7 protomers and $_mG_rU_rC^{PO}$ are colored gray, purple and brown, respectively. **e,f**, Close-up view of $_mG_rU_rC^{PO}$ (**e**) and $_mG_rA_rA^{PS}$-$SS$ (**f**) recognition at the antagonistic site. Residues (within 4.5 Å from the ligand) are shown in stick representations and are colored by atom, with the N, O, S and P atoms colored blue, red, yellow and orange, respectively. Dashed lines in cyan indicate hydrogen bonds (cutoff distance < 3.5 Å). **g**, Comparison of the conformations of GUC-v1$^{PS}$-$SS$, $_mG_rA_rA^{PS}$-$SS$ and $_mG_rU_rC^{PO}$ at the TLR7 antagonist binding site. **h**, Surface plasmon resonance (SPR) analyses of recombinant monkey TLR7 point mutants with the indicated concentrations of $_mG_rU_rC^{PS}$. Data shown are representative of 3 independent analyses (Supplementary Table S3). **i**, The relative binding free energy difference of antagonistic 3-mers (cyan: GUC-v1$^{PS}$, purple: $_mG_rA_rA^{PS}$) between the WT and F507S or F507L mutant (ΔΔG= ΔG$_{mutant}$ − ΔG$_{WT}$). **j,k**, HEK-293T cells co-transfected overnight with the

indicated TLR7 mutants, NF-κB-Luc, and UNC93B1 were treated with 1 μM of GUC-v1$^{PS}$ for ~30 min before overnight stimulation with 5 μg ml$^{-1}$ gardiquimod (GDQ) (**j**), or were pretreated 30 min with 1 μM of GUC-v1$^{PS}$ before the addition of 1 μM of naked RNA9.2$^{PS}$ for 1 h and overnight stimulation with 500 μM guanosine (**g**) and luciferase assay (**k**). **l**, The binding of the best docked pose of $_mG_rA_rA^{PO}$ to the TLR8 antagonist binding site in the inactive dimer (receptor template PDB: 8PFI). Dashed lines indicate the hydrophobic interactions formed among the 2′-O-methyl and the hydrophobic residues highly conserved between TLR7 and TLR8. Carbon atoms and protein ribbons in different protomers are colored in gray and steel blue, carbon atoms in $_mG_rA_rA^{PO}$ are colored in purple, O, N, and P atoms are in red, blue, and orange respectively. **m**, Surface plasmon resonance (SPR) analyses of recombinant human TLR8 point mutants with the indicated concentrations of $_mG_rA_rA^{PS}$. Data shown are representative of three independent analyses (Supplementary Table S3). **n**, HEK-293T cells co-transfected overnight with indicated TLR8 mutant, NF-κB-Luc, and UNC93B1 were treated with 5 μM of GAG-v1 for ~30 min before 6 to 8 h stimulation with 1 μg ml$^{-1}$ R848. **o**, HEK-293T cells co-transfected overnight with indicated TLR8 mutant, NF-κB-Luc, and UNC93B1 were pretreated 30 min with 5 μM GAG-v1$^{PS}$ before the addition of 1 μM of naked ssRNA40$^{PS}$ for 1 h and overnight stimulation with 2.5 mM uridine (U) followed by luciferase assay. **a**, Data are averaged from three biological replicates for each screen. **j,k,n,o**, Data are shown normalized to NT condition (± s.e.m. and two-way ANOVA with uncorrected Fisher's LSD tests shown compared to the GDQ condition (**j**), RNA9.2$^{PS}$ + G (**k**), R848 only (**n**) and ssRNA40$^{PS}$ + U (**o**)). Data are shown as the mean of n = 3 (**b,j,n,o**) or n = 4 (**k**) independent experiments. All statistics are available in Source Data Fig. 6.

In addition, although cooperative sensing of a TLR8 site 2 ligand RNA (ssRNA40) combined with uridine was seen with WT, F494L and G572D expression, the F494L mutant was the only variant lacking antagonism by GAG-v1[PS] (Fig. 6o and Extended Data Fig. 6m). Collectively, these observations establish that fragments of 2′-OMe-guanosine-modified RNAs can act as natural TLR7/8 antagonists and suggest that this natural antagonism may contribute to the maintenance of TLR7/8 homeostasis.

## Ribosomal RNA is a source of natural TLR7 and TLR8 antagonists

In mammalian cells, 106 2′-OMe sites have been characterized to-date in 5.8S, 18S and 28S ribosomal RNA (rRNA)[17]. Thirty-one of these 106 rRNA sites contain a 2′-OMe guanosine, among which the most frequent is the TLR7 inhibitory $_mG_rG_rA$ motif (Fig. 7a). Ribosomal RNA transfection inhibited sensing of the TLR7/8 agonist ssRNA40 in a dose-dependent

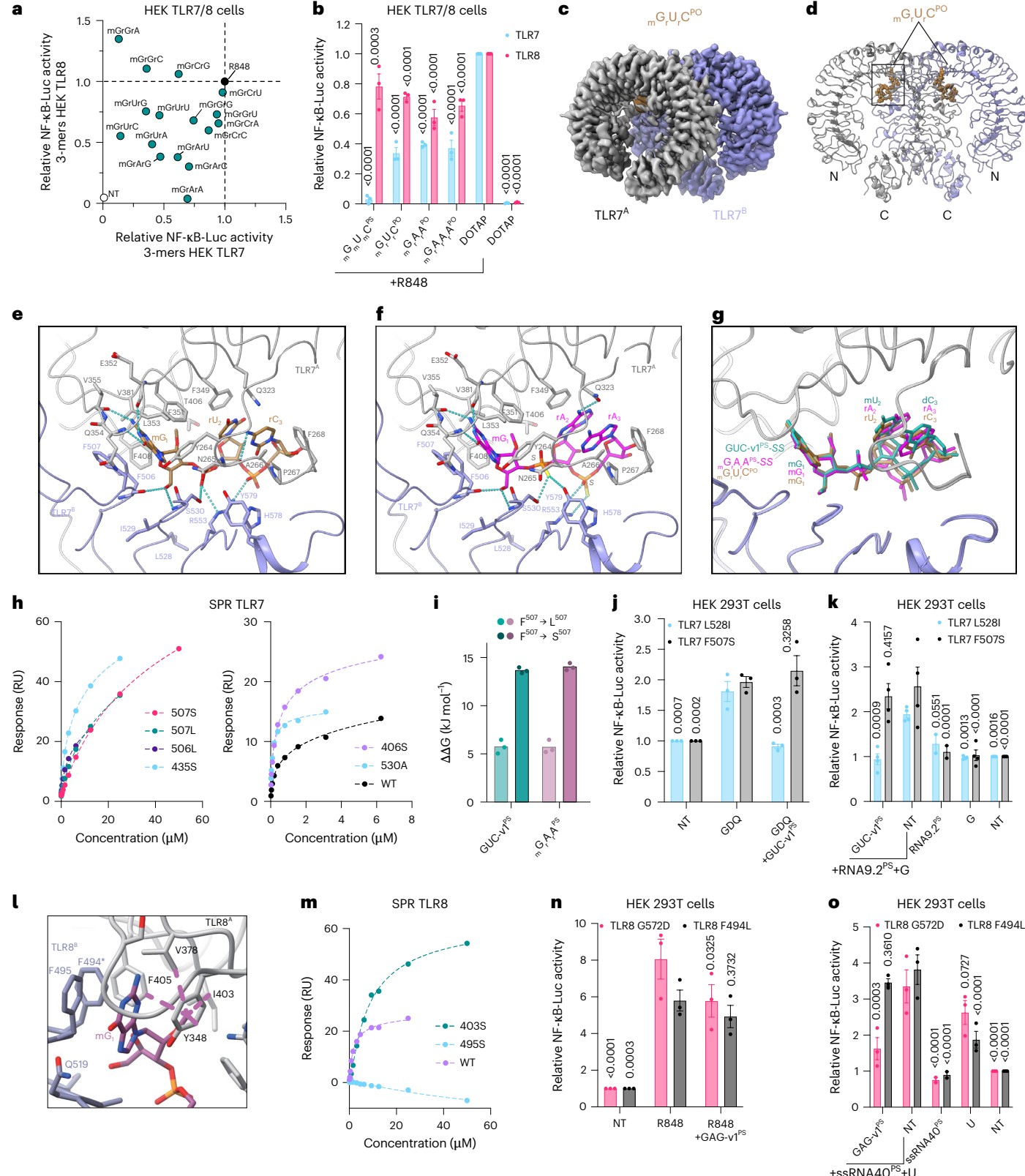

manner in differentiated THP-1 cells (Fig. 7b), supporting it could be a source of natural TLR7/8 antagonists. rRNA also inhibited ssRNA40 sensing by mouse TLR7 and human TLR8 in RAW cells and HEK TLR8 cells, respectively (Fig. 7c,d). Notably, the inhibitory activity of rRNA was also seen on R848-driven TLR8 sensing, confirming that the inhibition operated at the level of the receptor rather as a result of nuclease processing of the RNA (Extended Data Fig. 7a).

2′-OMe modification of rRNA is carried out by a ribonucleic protein complex that includes the enzyme methyltransferase fibrillarin (FBL)[29]. Small interfering RNA (siRNA)-mediated down-regulation of FBL protein levels was sufficient to significantly reduce rRNA 2′-OMe modification, as measured by a 2′-OMe specific PCR assay that favors unmethylated amplification (Extended Data Fig. 7b,c)[30]. Strikingly, in PMA-differentiated THP-1 and HEK TLR7 cells, transfected purified rRNA from siFBL-treated cells was less antagonistic of TLR7/8 than purified rRNA from untreated cells, thereby directly implicating 2′-OMe modifications in the antagonistic effect of rRNA on TLR7/8 sensing (Fig. 7e,f). Moreover, rRNA-driven antagonism of site 1 and 2 cooperativity was significantly impaired in HEK cells stably expressing the F507S mutant of TLR7 or the F494L mutant of TLR8 compared to their WT counterparts (Fig. 7g,h). Collectively, these findings establish the direct activity of rRNA 2′-OMe moieties on antagonism of TLR7/8 sensing through engagement of their antagonist binding sites. Notably, rRNA significantly reduced human TLR9 sensing, modestly impacted mouse TLR13 sensing, but did not affect human TLR3 sensing, suggesting rRNA may have an antagonistic activity on other endosomal nucleic acids sensors (Fig. 7i).

Liquid chromatography-mass spectrometry (LC-MS) analyses of cell lysates following transfection of purified rRNA confirmed an ~30–50% increase in the intracellular concentration of detected 2′-OMe nucleosides ($_mG$, $_mC$ and $_mU$) at 4 to 6 h after transfection, indicative of progressive nuclease fragmentation of rRNA (Fig. 7j). Because site 1 agonists of TLR7 and TLR8 rely on single nucleobases of guanosine (TLR7) and uridine (TLR8), we were interested to see whether single 2′-OMe bases were sufficient to bind to and antagonize TLR7 and TLR8. In vitro SPR analyses of the four 2′-OMe nucleobases with recombinant mmTLR7 and human TLR8 proteins demonstrated that $_mG$ was the only nucleobase robustly binding to TLR7 with an average $K_D$ of 34.4 μM, whereas no meaningful binding was seen with

hTLR8 (Fig. 7k, Extended Data Fig. 7d and Supplementary Table S3e,f). A weak binding of 2′-OMe adenosine to TLR7 was also noted. Aligning with this, MD analyses confirmed the preferential binding of $_mG$ to the TLR7 antagonist binding pocket, whereas binding of $_mA$ was 15-fold less than $_mG$ and $_mU/_mC$ were both dissociated from the antagonist binding site (Fig. 7l and Extended Data Fig. 7e). Functional analyses of the antagonistic activity of the four 2′-OMe bases on TLR7 sensing of R848 or RNA9.2$^{PS}$ combined with guanosine also confirmed preferential antagonism with $_mG$ over the other three nucleosides (Fig. 7m and Extended Data Fig. 7f). Notably, this antagonistic effect of $_mG$ was dependent on its direct binding to the antagonist binding site of TLR7, as revealed by the lack of significant antagonism of $_mG$ in cells stably expressing the TLR7 F507S mutation (Fig. 7m,n). Collectively, these findings establish that rRNA fragmentation generates natural TLR7 antagonists, driven by the direct interaction of 2′-OMe guanosine residues with the antagonistic pocket.

## Discussion

Recent structural studies revealed that the small-molecule inhibitor Cpd-7 binds TLR7 and stabilizes its open, inactive conformation, unlike typical agonists that induce a closed, active form[24]. Here, we show that specific 3-mer RNA fragments with a 5′-end 2′-OMe guanosine bind the same antagonist site, locking TLR7 in an open state and inhibiting its activity. Systematic analysis of 3-base oligos variants confirmed the unique role of 5′-end 2′-OMe guanosine. Structural data revealed aromatic stacking between the 5′-end guanine moiety of our 2′-OMe guanosine 3-base oligos and residues F351$^A$ and F507$^B$. Single 2′-OMe guanosine nucleosides also bound TLR7 via F507, though less potently than 3-mer oligos.

Rare TLR7 GOF mutations linked to systemic lupus erythematosus (SLE) were previously thought to enhance agonism[2,3,26]. Our data suggest instead that mutations at F506/F507 impair antagonism by endogenous 2′-OMe guanosine fragments. Mutant proteins showed reduced binding to $_mG_rU_rC^{PS}$ and no increased affinity for the agonist R848, indicating that loss of antagonism drives autoimmunity. This underscores the importance of TLR7 antagonism for immune homeostasis.

We further demonstrate that transfected rRNA antagonizes TLR7 in an F507-dependent manner, correlating with increased intracellular 2′-OMe guanosine levels. Critically, 2′-OMe guanosine was sufficient to

---

**Fig. 7 | Endogenous 2′-OMe ribosomal RNA fragments act as natural antagonists of TLR7/8. a**, Cumulative plot of the 2′-OMe G sites previously reported in human rRNA. **b**, PMA- and interferon-γ–primed THP-1 cells were transfected for 6 h with purified rRNA with DOTAP before overnight stimulation with 100 nM of transfected ssRNA40$^{PS}$. TNF levels measured by ELISA are shown relative to the ssRNA40-only condition (± s.e.m.). **c**, RAW-ELAM macrophages were transfected with 1.5 μg ml$^{-1}$ purified rRNA with DOTAP for 6 h before overnight stimulation with 250 nM of transfected ssRNA40$^{PS}$. **d**, HEK TLR8 cells were transfected with 1.5 μg ml$^{-1}$ rRNA with DOTAP for 6 h before overnight stimulation with 1 μM transfected ssRNA40$^{PS}$. **e**, PMA- and interferon-γ–primed THP-1 cells were transfected for 1 h with 3 μg ml$^{-1}$ rRNA from siFBL or siNEG treated cells with DOTAP before overnight stimulation with 100 nM of transfected ssRNA40$^{PS}$ and TNF ELISA. **f**, HEK TLR7 cells were transfected with 3 μg ml$^{-1}$ rRNA from siFBL or siNEG treated cells with DOTAP for 5 h before 1 h stimulation with 1 μM naked RNA9.2$^{PS}$ and overnight treatment with 500 μM guanosine (G). **g**, HEK293 cells stably expressing WT or the F507S human TLR7 mutant were transfected with 3 μg ml$^{-1}$ purified rRNA with DOTAP for 5 h before 1 h stimulation with 1 μM naked RNA9.2$^{PS}$ and overnight treatment with 500 μM guanosine (G). **h**, HEK 293 cells stably expressing WT or F494L TLR8 were transfected with 3 μg ml$^{-1}$ purified rRNA with DOTAP for 5 h before 1 h stimulation with 1 μM of naked ssRNA40$^{PS}$, for WT and F494L mutant, respectively, and overnight treatment with 2.5 mM udirine (U). **i**, HEK 293 cells stably expressing human TLR3, 9 or mouse TLR13 were transfected with 3 μg ml$^{-1}$ purified rRNA with DOTAP for 2 h, washed and incubated another 4 h before overnight stimulation with 0.5 μg ml$^{-1}$ pI:C, 200 nM ODN2006, or 0.5 μg ml$^{-1}$ Sa19, respectively. **j**, HEK-293T cells were transfected with 3 μg ml$^{-1}$ purified rRNA with

DOTAP for indicated times and were pelleted and lysed in 0.5 M perchloric acid for LC-MS analyses. 2′-OMe guanosine (mG), 2′-OMe cytosine (mC) and 2′-OMe uridine (mU) were quantified relative to the levels of cytosine. **k**, SPR analyses of mmTLR7 with 2′-OMe guanosine. Data shown are representative of $n = 3$ (Supplementary Table S3). (**l**) Antagonist binding site of mmTLR7 in complex of single nucleotide (upper panel: $_mG$, lower panel: $_mC$) after 400 ns MD simulations. Protomers are purple and gray. Transparent light gray shows MD simulation of $_mG$ from GUC-v1$^{PS}$ binding to TLR7. $_mG$ but not $_mC$ remains at the binding site after 400 ns. **m,n**, HEK 293 cells stably expressing WT (**m**) or F507S TLR7 mutant (**n**) were treated with 500 μM of indicated 2′-OMe nucleoside before stimulation with 1 μM of naked RNA9.2$^{PS}$ and 500 μM guanosine (G). **c,d,g,h,i,m,n**, Data were background-corrected using the NT condition and are shown as expression relative to the ssRNA40$^{PS}$-only condition (**c,d**), DOTAP/RNA9.2$^{PS}$ + G condition (**g**), DOTAP/ssRNA40$^{PS}$ + U condition (**h**), DOTAP/agonist condition (**i**), RNA9.2$^{PS}$ + G [m,n] (± s.e.m. and one-way (**c,d,i,m,n**) or two-way (**g,h**) ANOVA with uncorrected Fisher's LSD tests shown compared to the ssRNA40-only condition (**c,d**), DOTAP/RNA9.2$^{PS}$ + G condition (**g**), WT compared to F494L mutant (**h**), DOTAP+agonist conditions (**i**), RNA9.2$^{PS}$ + G condition (**m,n**); **c**: $P = 0.0036$; **d**: $P = 0.0013$; **i, m, n**: $P < 0.0001$). **e,f,j**, Data are shown relative to the ssRNA40+siFBL rRNA condition (**e**), the NT condition (**f**) or the T = 0 h time point (**j**) (± s.e.m. and one-way (**e,f**) or two-way ANOVA (**j**) with uncorrected Fisher's LSD tests shown compared to the DOTAP/ssRNA40$^{PS}$ condition (**e**), the DOTAP/ RNA9.2$^{PS}$ + G (**f**) or the T = 0 h point of each ratio (**j**); **e**: $P = 0.0004$; **f**: $P < 0.0001$). Data are shown as the mean of $n = 3$ (**b–f**, **h–j**, **m**, **n**) or 4 (**g**) independent experiments. All statistics are available in Source Data Fig. 7.

halve the stimulatory activity of the same concentration of guanosine in the presence of RNA9.2[PS], through engagement of the F507 residue. Given rRNA's abundance, we propose rRNA fragments as the primary source of natural TLR7 antagonists, though other 2′-OMe-modified RNAs (for example, capped mRNAs, tRNAs) likely contribute[31]. Notably, further studies will be required to confirm the unambiguous detection of partial 2′-OMe rRNA degradation products, which we did not evidence here.

Given that the affinity of $_mG_rU_rC^{PS}$ binding to TLR7 was ~10 times greater than that of 2′-OMe guanosine nucleoside in our SPR analyses, we propose that 3′-end extensions enhance TLR7 antagonism. Interestingly, longer TLR7-inhibiting oligos with the optimal TLR7 $_mG_mU_mC/_mG_mU$ inhibiting motif have been reported, such as IMO8400/bazlitoran, which advanced to clinical studies, and miR-224-5p mut2[32,33]. These observations support that diverse RNA fragments can engage the antagonist site.

For TLR8, functional analyses indicate a similar antagonistic mechanism involving binding of 2′-OMe guanosine-containing RNA

fragments to conserved residues F494/495 (equivalent to TLR7 F506/507 residues). A rare TLR8 F494L mutant residue reduced antagonism and was linked to neutropenia in a patient[4]. Notably, the TLR8 F494L variant was partially refractory to GAG-v1[PS] and rRNA antagonism in our assays. In addition, the recombinant TLR8 F495S mutant (mimicking the F507S mutation in TLR7) ablated interaction with $_mG_rA_rA^{PS}$, supporting that F494/495 are essential for TLR8 antagonism. However, the failure of a single 2′-OMe guanosine nucleoside to bind to TLR8, together with the overall weaker antagonistic activity of our 3-mers on human TLR8 compared with human TLR7, raises the possibility that another base modification may bind more favorably to the TLR8 antagonist binding site. Interestingly, some 2′-OMe 3-mers potentiated TLR8 sensing, highlighting a complex interplay between agonism and antagonism for this receptor.

Importantly, structural comparisons confirm that antagonist binding and site 2 engagement are mutually exclusive (Extended Data Fig. 7g), establishing a competitive model: uridine-rich fragments activate TLR7 via site 2 and guanosine via site 1, resulting in

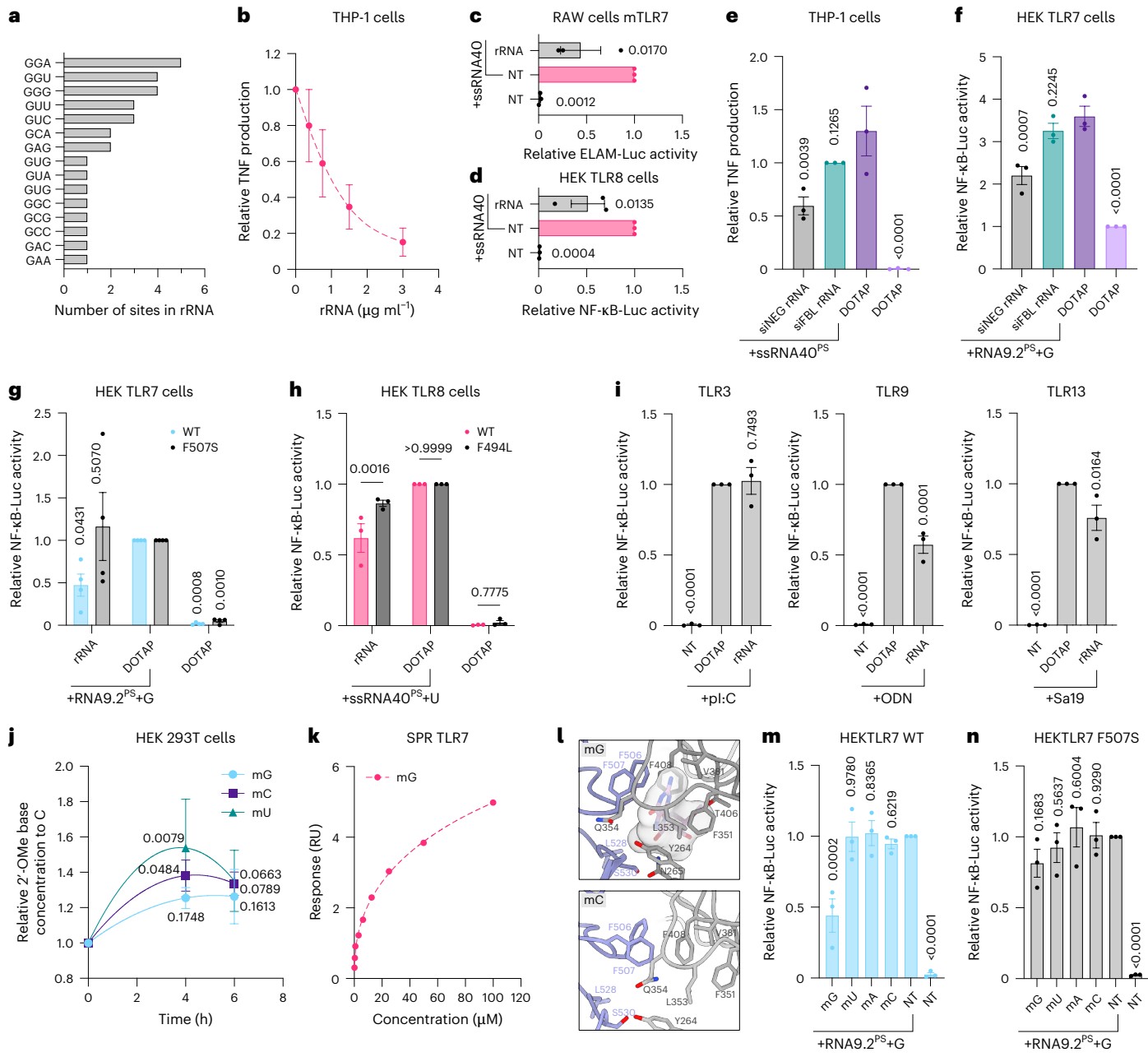

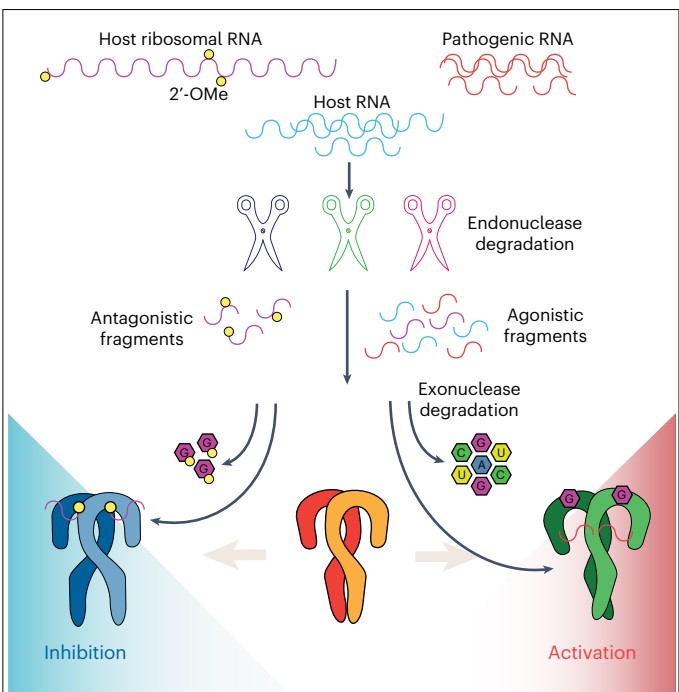

**Fig. 8 | Natural TLR7 antagonism.** Endosomal nucleic acids from various origins (*for example* host or pathogens) are sequentially processed by endo and exonucleases including RNase T2/2/6 and PLD3/4, respectively. Partial fragments (~2-3 bases) and single unmodified guanosine bind to site 2 and site 1 of TLR7, respectively. Cooperative binding to site 1 and 2 leads to a closed conformation of the dimers, allowing for downstream signaling. On the other hand, binding of RNA fragments containing 2′-OMe guanosine residues or 2′-OMe guanosine single nucleosides, originating from abundant ribosomal RNA, bind to the antagonistic sites of TLR7 resulting in an inactive open conformation of the dimers. We show that TLR7 sensing is kept in check by naturally occurring 2′-OMe-modified ribosomal RNA fragments, avoiding autoimmune responses to host RNA in the absence of pathogens.

a closed active form, whereas 2′-OMe guanosine fragments inhibit activation through the antagonist site and the resulting open inactive form.

In conclusion, our results provide new insight into the mechanisms by which TLR7 and TLR8 activation is normally limited to pathogenic contexts and avoided during homeostatic clearance of apoptotic cells and steady-state cell function. These findings imply that activation of TLR7 and TLR8 relies on a displacement of natural antagonism (driven by antagonists such as 2′-OMe guanosine), upon accumulation of endosomal agonistic RNA or DNA fragments, rather than on the detection of 'non-self' RNA features (Fig. 8). Based on our observations that sensing of TLR13 and TLR9 were also inhibited by rRNA, it will be important to define whether the antagonism of TLR7 and TLR8 described in this study represents a more general regulatory mechanism common to other nucleic acid sensors.

## Online content

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

[1]Centre for Innate Immunity and Infectious Diseases, Hudson Institute of Medical Research, Clayton, Victoria, Australia. [2]Department of Molecular and Translational Science, Monash University, Clayton, Victoria, Australia. [3]Department of Clinical Laboratory Sciences, College of Applied Medical Sciences, Taif University, Turabah, Saudi Arabia. [4]Graduate School of Pharmaceutical Sciences, The University of Tokyo, Bunkyō, Japan. [5]Research School of Biology, College of Science, The Australian National University, Acton, Australian Capital Territory, Australia. [6]Mass Spectrometry Facility, St Vincent's Institute of Medical Research, Fitzroy, Victoria, Australia. [7]Noxopharm Limited, Castle Hill, New South Wales, Australia. [8]Pharmorage Pty Limited, Hawthorn East, Victoria, Australia. [9]Sydney Analytical Core Research Facility, University of Sydney, Camperdown, New South Wales, Australia. [10]School of Science, Western Sydney University, Parramatta, New South Wales, Australia. [11]School of Life and Environmental Sciences, The University of Sydney, Camperdown, New South Wales, Australia. [12]Division of Immunology and Infectious Diseases, John Curtin School of Medical Research, The Australian National University, Acton, Australian Capital Territory, Australia. [13]CSIRO Health & Biosecurity, Australian Centre for Disease Preparedness, Geelong, Victoria, Australia. [14]Integrated DNA Technologies Inc, Coralville, IA, USA. [15]Ritchie Centre, Hudson Institute of Medical Research, Clayton, Victoria, Australia. [16]Department of Paediatrics, Monash University, Clayton, Victoria, Australia. [17]The Francis Crick Institute, London, UK. [18]Graduate School of Frontier Sciences, The University of Tokyo, Bunkyō, Japan. [19]These authors contributed equally: Arwaf S. Alharbi, Sunil Sapkota, Zhikuan Zhang, Ruitao Jin. [20]These authors jointly supervised this work: Michael P. Gantier, Toshiyuki Shimizu, Ben Corry. ✉e-mail: sunil.sapkota@hudson.org.au; michael.gantier@hudson.org.au

## Methods

### Cell culture and reagents

293XL-hTLR7 (#293xl-htlr7), 293XL-hTLR8 (293xl-htlr8), 293XL-hTLR9-HA (#293xl-htlr9ha), HEK-Blue hTLR3 (#hkb-htlr3) and HEK-Blue mTLR13 (#hkb-mtlr13) stably expressing human TLR7, TLR8, TLR9, TLR3 or mouse TLR13, respectively, were purchased from Invivogen and maintained in Dulbecco's modified Eagle's medium plus L-glutamine supplemented with 1× antibiotic/antimycotic (Thermo Fisher Scientific) and 10% heat-inactivated fetal bovine serum (referred to as complete DMEM), with 10 to 30 µg ml⁻¹ Blasticidin (Invivogen). HEK-293T cells[34] were also maintained in complete DMEM. HeLa cells (#ATCC CCL-2) were maintained in Gibco minimal essential growth medium (MEM) supplemented with 1% HEPES and 10% FBS. Human acute myeloid leukemia THP-1 cells were grown in RPMI 1640 plus L-glutamine medium (Life Technologies) complemented with 1x antibiotic/antimycotic and 10% heat-inactivated fetal bovine serum (referred to as complete RPMI). THP-1 cells were not differentiated with PMA unless otherwise noted. Overnight THP-1 differentiation was carried out with 20 ng ml⁻¹ PMA (Merck), and the cells were further primed with 20 ng ml⁻¹ recombinant IFNγ (BioLegend) for 6 h before stimulation with TLR7/8 agonists, as described previously[35]. RAW264.7-ELAM macrophages[36] and immortalized *Tlr4*-deficient BMDMs (gift from E. Latz) were grown in complete DMEM. All the cells were cultured at 37 °C with 5% $CO_2$. Cell lines were passaged two or three times a week and tested negative for mycoplasma contamination on routine basis using Mycostrip (Invivogen).

Cells were treated with indicated concentration of oligonucleotides or Enpatoran (MedChemExpress) for 20 to 60 min before R848 (Cayman Chemical), CL075 (Invivogen), Gardiquimod (Invivogen), uridine (Sigma) or Motolimod (MedChemExpress), as indicated. 2′-OMe guanosine, 2′-OMe adenosine, 2′-OMe uridine, 2′-OMe cytosine, guanosine, adenosine and cytosine were resuspended in DMSO and purchased from MedChemExpress for cell-based studies and LC-MS analyses as standards. Immunostimulatory ssRNA40^PS (ref. 12), RNA9.2^PS7 (ref. 12), B-406-AS-1 (ref. 37), ssRNA40^PO and RNA9.2s^PO ssRNAs, trimer and longer oligonucleotides were all commercially synthesized by Integrated DNA Technologies, Syngenis or Wuxi AppTec and resuspended in RNase-free TE buffer, pH 8.0 (Thermo Fisher Scientific). For in vivo experiments, the oligonucleotides were HPLC-purified and confirmed to be endotoxin free by Limulus Amebocyte Lysate gel-clot method. Oligonucleotide sequences and modifications are provided in Supplementary Tables S1 and S6. 2′-MOE is moX, 2′-OMe is mX, DNA is dX, RNA is rX, LNA is lX and phosphorothioate internucleotide linkages are denoted with an asterisk. Where indicated, ssRNAs and purified ribosomal RNA were transfected with DOTAP (Roche) using a ratio of 600 ng RNA for 4.5 µl DOTAP per 96 wells. For transfection of bacterial RNA, we used 400 ng RNA complexed with 4.5 µl DOTAP per 96 wells. For cooperative agonism of TLR7 with RNA9.2^PS and guanosine and agonism of TLR8 with ssRNA40^PS and uridine, the cells were treated in pure DMEM with naked RNA9.2^PS or ssRNA40^PS for 1 h before the addition guanosine and uridine with complete DMEM.

*Tlr7^Y264H* C57BL/6NCrl mice (used under Australian National University animal ethics, reference A2021/29) have a mutation leading constitutive activation of TLR7 and SLE-like disease[3]. Primary BMDMs from 9- to 11-week-old *Tlr7^Y264H* heterozygous female mice were extracted and differentiated for 5 days in complete DMEM supplemented with L929 conditioned medium[38], before 24 h incubation with 5 µM GGC-v1 or 100 nM Enpatoran and total RNA purification for RNA sequencing and RT-qPCRs. Primary bone-marrow-derived DCs were generated from purified bone marrow from WT 10- to 12-week-old male mice (used under Monash Medical Centre B Animal Ethics Committee reference MMCB/2024/30) following an 8-day differentiation in 200 ng ml⁻¹ InVivoMAb recombinant Flt3L-Ig (hum/hum) (BioXCell), before treatment as indicated, as reported previously[39].

Fresh blood was collected from three healthy donors (two males and one female ranging from 40 to 45 years old). Healthy adult participants were recruited exclusively among staff at the Hudson Institute of Medical Research or Monash University. Participation was voluntary, which could introduce a potential self-selection bias, as individuals who elect to participate in biomedical research may differ from the general population in health literacy, education level and motivation. Although this may restrict the generalizability of our healthy reference group to broader populations, the comparison between different treatments within the same donor performed in our study remains valid. The participants provided written informed consent before participation, using the Monash Health Human Research Ethics Committee-approved Participant Information and Consent Form (Protocol RES-18-0000-363A). No financial compensation was provided. PBMCs were purified using Histopaque-gradient (Sigma) centrifugation in SepMate tubes (StemCell Technologies) as described previously[40]. All oligonucleotide screens were performed blinded (with no knowledge of the sequences used).

### Generation of human iPSC-derived macrophages

The HipSci HPSI0114i-kolf_2 (ECACC 77650100) iPS line was a gift from Wellcome Trust Sanger Institute and routinely cultured on growth factor-reduced Matrigel (Corning)-coated 6-well plates in mTeSR Plus medium (StemCell Technologies). The iPS line was regularly validated for normal karyotype and negative mycoplasma. Differentiation of human Kolf2 iPS cells toward the macrophage lineage were adapted from a previous study[41] (H.H.F. et al., manuscript in preparation).

### Plasmids

pCMV6 vectors expressing Human TLR7 F507S and L528I (gifts from C. David and Y. Crow) and the UNC93B1-mCitrine vector were used[2,42]. pRP[Exp]-mCherry-CMV > TLR8 F494L and pRP[Exp]-mCherry-CMV > TLR8-G572D expressing GOF TLR8 variants[4] under the control of a CMV promoter were cloned, amplified and sequence-validated by Sanger sequencing by VectorBuilder Inc. pCMV6-TLR7-F507S and L528I were purified using an EndoFree Plasmid Maxi Kit (Qiagen), and were sequence-validated using nanopore whole-plasmid sequencing service from Micromon Genomics Sanger Sequencing Facility (Monash University).

### siRNA transfection

6.6×10⁵ HeLa cells were seeded into T25 flasks and reverse transfected with 40 nM non-targeting siRNA (siNEG) or SMARTpool of fibrillarin targeting siRNA (siFBL) combined with DharmaFECT1 (DF1 – Dharmacon) transfection reagent and an appropriate volume of Opti-MEM reduced serum media. Final concentration of DF1 was determined as per manufacturer's guidelines. Following 6 h of incubation in the transfection mix, cells were replenished with complete growth media and incubated for 72 to 120 h with media changes at 48-h intervals. siRNA sequences are provided in Supplementary Table S6.

### Protein extraction and Western blotting

2.5×10⁵ HeLa cells were seeded into each well of 6-well plates for protein purification. Following siRNA transfection, as described above, 1x Rippa buffer (50 mM Tris pH 7.4, 150 mM NaCl, 1% IGEPAL, 0.5% sodium deoxycholate, 0.1% SDS) containing 1x Protease Inhibitor Cocktail (PIC – Roche) was used to lyse the cells. Lysates were centrifuged at 16,000 g for 10 mins at 4 °C followed by BCA Protein Assay to quantify the protein concentrations. 16 µg protein along with PAGERuler ladder (Thermo Fisher Scientific) were loaded onto NuPAGE 4-12% Bis-Tris gels (Thermo Fisher Scientific) and electrophoresed using 1x NuPAGE MES−transfer (Bio-Rad) onto nitrocellulose membrane (Amersham). Membranes were blocked for 1 hour using 5% BSA in PBS then incubated overnight at 4 °C in primary antibodies (Anti-Fibrillarin, 1:500 – Abcam ab4566 Lot 1088391-1, and Anti-beta Actin, 1:10,000 – Abcam ab8227

Lot 1103556-1) diluted in blocking buffer. Following several washes with PBST wash buffer, membrane was incubated for 1 hr in the appropriate HRP conjugated secondary antibodies (goat anti-mouse secondary, 1:5,000 – Abcam ab205719 Lot 1036603-15; or goat anti-rabbit secondary, 1:10,000 – Sigma A0545 Lot 069M4835V) also diluted in blocking buffer. The membrane was then imaged using the Clarity western ECL kit (Bio-Rad) on the Bio-Rad ChemiDoc machine with ImageLab software (v6.1). Protein densitometries were quantified using ImageJ software and normalized to β-actin expression. Densitometry graphs were plotted relative to protein expression in siNEG transfected cell.

## Stable expression of TLR7 and TLR8 variants

Cells stably expressing the Human TLR7 F507S and L528I variants were generated by transfecting PvuI-linearized pCMV6-TLR7 vectors[2] in HEK-Blue IFN-α/β Cells (Invivogen #hkb-ifnabv2-b) and by selecting them with Geneticin (1 mg ml⁻¹) for 14 days. Stable cells expressing human TLR8 F494L and G572D variants were generated by co-transfecting PvuI-linearized pRP[Exp]-mCherry-CMV and PvuI-linearized pEGFP-N2 (Clontech) vectors in HEK 293 cells, and by selecting them with Geneticin (1 mg ml⁻¹) for 18 days. Once stably growing, the cells were separated into single-cell clones by dilution and expanded before screening for maximum TLR7 or TLR8 function. TLR7/8 expression of the mutants in the stable clones maintained in 1 mg ml⁻¹ Geneticin was validated by RT-qPCR, and expression levels were comparable between the two mutants for each gene as shown in Extended Data Fig. 7h.

## Luciferase assays

HEK293 cells stably expressing hTLR8, hTLR7, hTLR9, hTLR3 or mTLR13 were reverse transfected with pNF-κB-Luc4 reporter (Clontech), with Lipofectamine 2000 (Thermo Fisher Scientific), according to the manufacturer's protocol. Briefly, 500,000 to 700,000 cells were reverse transfected with 200 to 400 ng pNF-κB-Luc4 reporter with 1.2 μl Lipofectamine 2000 per well of a 6-well plate and incubated for 3 to 24 h at 37 °C with 5% $CO_2$. Following transfection, the cells were collected from the 6-wells and aliquoted into 96 wells, just before oligo and overnight TLR stimulation. Similarly, the RAW264.7 cells stably expressing an ELAM-Luc reporter were treated overnight. As presented in Fig. 6, HEK-293T cells were co-transfected with 300 ng or 200 ng TLR7 GOF or TLR8 GOF vectors, respectively, along with 100 to 150 ng human UNC93B1-mCitrine[42] and 50 ng pNF-κB-Luc4 reporter per well of a 6-well plate with 1.5 μl lipofectamine 2000. Following overnight incubation, the cells were collected from the 6-wells and aliquoted into 96 wells, just before oligo and overnight (TLR7) or 6 to 8 h (TLR8) TLR stimulation. In all cases, the cells were lysed in 40 μl (for a 96-well plate) of 1X Glo Lysis buffer (Promega) for 10 min at room temperature. 15 μl of the lysate was then subjected to firefly luciferase assay using 35 μl Luciferase Assay Reagent (Promega). Luminescence was quantified with a Fluostar OPTIMA (BMG LABTECH) luminometer.

## Cytokine analyses

Production of human IP-10, IL-6 or TNF levels were measured in supernatants from iPSC-macrophages or THP-1 cells using the IP-10 (BD Biosciences, #550926), IL-6 (BD Biosciences, #555220) and TNF (BD Biosciences, #555212) ELISA kits, respectively. Mouse TNF and IFNα levels were measured using TNF (BD Biosciences, #550534) and IFNα (Invivogen, LumiKine Xpress mIFN-a 2.0) specific ELISA Kits. Tetramethylbenzidine substrate (Thermo Fisher Scientific) or Quanti-Luc reagent (Invivogen) was used for quantification of the cytokines on a Fluostar OPTIMA (BMG LABTECH) plate-reader with OPTIMA-Control v2.2R2 software. Data analysis was conducted with MARS Data analysis software 3.01R2. All ELISAs were performed according to the manufacturers' instructions. For Flt3L-DC data, IFNα was only detected in two out of three mice (however, TNFα was detected in all three mice). Concentration of TNF in mouse serum samples (Fig. 5b) was

quantified using LEGENDplex Mouse TNF-α Capture Bead A6 (BioLegend, #740066) as part of the Mouse Anti-Virus Response Mix and Match Panel (BioLegend #740625, #740624, #740623) according to the manufacturer's instructions. Sample acquisition was performed using a BD LSR-II flow cytometer (BD Biosciences) and the data analyzed with the LEGENDplex Data Analysis Software Suite (BioLegend). Concentration of human IFNα, TNF, IL12p70 and IFNγ (Extended Data Fig. 3) were quantified using cytometric bead arrays (BD Biosciences, IFNA #560379 lot 3117527, TNF #560112 lot 5013222, IL12p70 #558283 lot 5121732 and IFNG # 558269 lot 5031631) on an Attune NxT Flow Cytometer (Thermo Fisher Scientific), according to the manufacturer's instructions, with data analysis performed using FlowJo Software version 10.9. For PBMCs, IFNγ levels from one donor saturated the assay and were omitted in the calculations of the averages in Supplementary Table S2.

## Preparation of cell lysates for LC-MS analyses

About 2 million HEK 293 T cells were treated as indicated and pelleted at 300 $g$ in 15 ml tubes before washing with 1 ml chilled PBS. Cells were pelleted again at 300 g for 5 min at 4 °C and ~800 μl PBS was removed. The remaining pellet in ~200 μl PBS was transferred to a 1.5 ml tube and pelleted further at 300 g to remove the remaining PBS. Each pellet was completely lysed for 5 min with 50 μl freshly prepared 0.5 M perchloric acid on ice. The lysed samples were centrifuged at 17,000 $g$ for 15 min at 4 °C, and the cleared supernatants were collected in a clean 1.5 ml tube. The supernatants were neutralized by adding 12.5 μl of ice-cold 2.3 M $KHCO_3$ and centrifuged at 17,000 $g$ for 15 min at 4 °C. Following the final centrifugation, the cleared supernatants were collected for further LC-MS analyses.

## LC-MS/MS analyses of nucleosides and 2'-OMe nucleosides

LC-MS/MS analyses were performed using a Shimadzu LC-30AD binary pump system (Shimadzu) coupled to a hybrid triple quadrupole/linear ion trap mass spectrometer (QTRAP 5500, Sciex). The curtain gas, ion source gases 1 and 2, and collision gas were optimized for analysis. The ion spray voltage and source temperature were set at 5,000 V and 300 °C, respectively. The target compounds were analyzed in multiple-reaction monitoring mode with specific parameters listed in Supplementary Table 7. Each multiple-reaction monitoring transition (precursor ion to product ion) was monitored with a dwell time of 50 ms. Chromatographic separation was achieved using a Synergi Hydro-RP column (100 mm × 2 mm, 2.5 μm). The column temperature was set at 30 °C, and the flow rate was set at 0.3 ml min⁻¹. The mobile phase consisted of 0.1% formic acid in water (mobile phase A) and 90% acetonitrile with 0.1% formic acid (mobile phase B). The gradient elution program was as follows: 2.5% B held for 2 mins followed by increased to 20% B in 2 min; then sharply increased to 90% B in 0.2 min and held for 3 min at 90% B; finally, the column was re-equilibrated to initial conditions for 3 min, resulting in a total analysis time of 10 min per sample. The injection volume was 10 μl. Quantification was performed using external standard calibration. A series of standard working solutions were prepared to establish a calibration curve, which exhibited linearity with $R^2 > 0.995$. The working solutions of analytes were obtained by a series dilution from 100 μmol ml⁻¹ stock solution in water, resulting in a final concentration range of 0.4 to 200 nmol ml⁻¹. Data analysis was conducted using MultiQuant Software 2.0. Multiple-reaction monitoring parameters for nucleosides and their derivatives are provided in Supplementary Table S7 and examples of overlayed chromatograms of nucleosides and derivatives in standard solution are shown in Supplementary Fig. S1.

## RNA and RT-qPCR analyses

For Figs. 3c and 5c and Extended Data Fig. 7, total RNA was purified from $Tlr7^{Y264H}$ primary BMDMs, mouse skin biopsies or HEK TLR7/8 cells using the PureLink RNA Mini Kit (Thermo Fisher Scientific) and DNase-treated using the Purelink DNASE set (Thermo Fisher Scientific). For ribosomal

RNA enrichment, 5 µg total RNA from HEK 293 cells obtained with the PureLink kit was purified using the Ribominus eukaryote kit v2 (Thermo Fisher Scientific) with minor adaptations to the manufacturer's instructions. Briefly, the beads bound to rRNA were resuspended in 300 µl RNase-free water, and heated 5 min at 70 °C to elute the rRNA. The beads were collected with the DynaMag 2 Magnetic Stand (Thermo Fisher Scientific), and the remaining rRNA solution purified further with the PureLink kit. Total bacterial RNA was extracted from *Escherichia coli* JM109 (Promega) using the PureLink RNA Mini Kit (Thermo Fisher Scientific) according to the manufacturer's instructions with minor modifications. Briefly, 10 ml overnight bacterial culture was harvested by centrifugation and the resulting pellet was resuspended in 0.5 ml of the supplied lysis buffer supplemented with 1% (v/v) 2-mercaptoethanol. The suspension was vigorously vortexed and subjected to two cycles of freezing and thawing for better cell lysis. The lysate was centrifuged at 12,000 *g* for 5 min at 4 °C, and approximately 400 µl of the supernatant was transferred to a clean 1.5 ml microcentrifuge tube. An equal volume of ice-cold RNase-free 70% ethanol was added to the supernatant and mixed thoroughly by pipetting five or six times. The resulting mixture which contains bacterial RNA was then loaded to the purification column, and RNA isolation was completed following the manufacturer's protocol. On column DNase treatment was performed using the PureLink DNase Set (Thermo Fisher Scientific).

Random hexamer cDNA was synthesized from isolated RNA using the High-Capacity cDNA Archive kits (Thermo Fisher Scientific) according to the manufacturer's instructions. RT-qPCR was carried out with the Power SYBR Green Master Mix (Thermo Fisher Scientific) on a QuantStudio 6 Flex RT-PCR system (Thermo Fisher Scientific) with the QuantStudio Real-Time PCR Software v1.7.2. Each PCR was performed in technical duplicate and mouse and human *18S* were used as the reference gene. Each amplicon was gel-purified and used to generate a standard curve for the quantification of gene expression. Melting curves were used in each run to confirm specificity of amplification. The following primers were used: mouse 18 s: Rn18s-FWD 5′-GTAACCCGTTGAACCCCATT-3′; Rn18s-REV 5′-CCATCCAATCGGTAGTAGCG-3′; M-F-Slc13a3 5′-GGA AGG CCG ATG CCT CTA TG-3′; M-R-Slc13a3 5′-GGA AGT TGG TGT CGA GGA AGT-3′; M-F-Itgal 5′-CCA GAC TTT TGC TAC TGG GAC-3′; M-R-Itgal 5′-GCT TGT TCG GCA GTG ATA GAG-3′; M-F-Fpr1 5′-CAT TTG GTT GGT TCA TGT GCA A-3′; M-R-Fpr1 5′-AAT ACA GCG GTC CAG TGC AAT-3′; M-F-Fpr2 5′-GAG CCT GGC TAG GAA GGT G-3′; M-R-Fpr2 5′-TGC TGA AAC CAA TAA GGA ACC TG-3′; M-F-Cd300e 5′-TGG GTC TTA CTG TGT CAA GAT-3′; M-R-Cd300e 5′-CTT ACA CTG ACC GAT GGA TCA C-3′; Ms Cxcl1-F1 5′-CCT TGA CCC TGA AGC TCC CT-3′; MsCxcl1-R1 5′-CAG GTG CCA TCA GAG CAG TCT-3′; mIL17aF1 5′-ACC GCA ATG AAG ACC CTG AT-3′; mIL17aR1 5′-TCC CTC CGC ATT GAC ACA-3′; MsTnfaF1 5′-CAA AAT TCG AGT GAC AAG CCT G-3′; MsTnfaR1 5′-GAG ATC CAT GCC GTT GGC-3′; hTLR7-RT-FWD 5′-CCT TTC CCA GAG CAT ACA GC-3′; hTLR7-RT-REV 5′-GGA CAG AAC TCC CAC AGA GC-3′; hTLR8-RT-FWD 5′-CAG AGC ATC AAC CAA AGC AA-3′; hTLR8-RT-REV 5′-GCT GCC GTA GCC TCA AAT AC-3′; Hu18s F 5′-CGG CTA CCA CAT CCA AGG AA-3′; Hu18s R 5′-GCT GGA ATT ACC GCG GCT-3′.

### RNA purification and reverse transcription for RTL-P assay
RNA was harvested from HeLa cell monolayer using Tri Reagent (Sigma) according to the manufacturer's instructions and eluted in RNase-free water. 1,000 ng RNA were reverse transcribed using the SuperScript III First-Strand Synthesis System (Invitrogen), as per manufacturer's instructions, with the following modifications: 20 µl reactions were set up in duplicates, using a high (10 mM) and a low (4 µM) dNTP concentration and 28S rRNA specific RT primer (5′- ATCGGTCGCGTTACCG-3′). 28S rRNA amplification was used to measure levels in 2′-*O*-methylation at low dNTP concentration (4 µM) by qPCR analyses using amplification of a target 2′-OMe site with 28S-FD1 region forward primer (5′-TTGAACATGGGTCAGTCGGTCC-3′) expressed relative to a non-methylated region amplified with 28S-FU3

region forward primer (5′-CAGGTGCAGATCTTGGTGGTAG-3′) and a common reverse primer for both forward primers (28S rRNA reverse primer: 5′-ATCGGTCGCGTTACCGCACT-3′)[30,43]. 20 µl qPCR reactions were set up with 1x SYBR Green Universal Master Mix, 10 µM forward primers specific to 28S-FD1 (control) region or 28S-FU3 region (containing methylation sites), 10 µM 28S rRNA reverse primer common to both the forward primers and 2 µl cDNA template. qPCR was performed using standard cycling conditions on an Applied Biosystems QuantStudio 3 machine using QuantStudio Design & Analysis Software v1.5.2. Post qPCR calculations for the RTL-P Assay were performed using the $2^{(-(28S\text{-}FU3\,Ct)\,-\,(28S\text{-}FD1\,Ct))}$ for each sample, then expressed relative to the siNEG condition of each independent experiment.

### Statistical analyses
Statistical analyses were carried out using Prism 10 (GraphPad Software). Data distribution was assumed to be normal but this was not formally tested. One-way and two-way ANOVA with uncorrected Fisher's LSD were used when comparing groups of conditions, whereas unpaired two-tailed *t*-tests were used when comparing selected pairs of conditions. Data collection and analysis were not performed blind to the conditions of the experiments unless otherwise stated.

### Reporting summary
Further information on research design is available in the Nature Portfolio Reporting Summary linked to this article.

### Data availability
RNA-sequencing data have been deposited in the NCBI Gene Expression Omnibus under the accession code GSE291606. Cryo-EM maps have been deposited in the Electron Microscopy Data Bank under the accession codes EMD-60515 (TLR7/GUC-v1$^{PS}$ complex), EMD-60541 (TLR7/$_m$G$_r$A$_r$A$^{PS}$ complex) and EMD-63406 (TLR7/$_m$G$_r$U$_r$C$^{PO}$ complex). The coordinates of the atomic models have been deposited in the Protein Data Bank under the accession codes TLR7/GUC-v1$^{PS}$-*SS* 8ZW2), TLR7/GUC-v1$^{PS}$-*RR* (8ZW4), TLR7/mGrArA$^{PS}$-*SS* (8ZXE), TLR7/mGrArA$^{PS}$-*RR* (8ZXF) and TLR7/mG$_r$U$_r$C$^{PO}$ (9LUV). All other data are available in the article and Supplementary Information. Source data are provided with this paper.

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

## Acknowledgements

We thank K. Sakaniwa (The University of Tokyo, Tokyo, Japan) for the purified TLR8 protein; V. Hornung (Ludwig-Maximilians-University, Munich, Germany) for the HEK-293T cells; C. David and Y. Crow (Université Paris Cité, Paris, France) for the TLR7 GOF expressing vectors; R. Veedu (Murdoch University, Perth, Australia) for advice on short oligonucleotide synthesis; E. Latz (Deutsches Rheuma-Forschungszentrum Berlin, Berlin, Germany) for the UNC93B1-mCitrine vector and *Tlr4*-deficient iBMDMs; K. Jeppe, J. Steele and C. Barlow (Monash University, Clayton, Australia) for help with LC-MS/MS analyses; M. Sweet (The University of Queensland, Brisbane, Australia) for the RAW-ELAM cells; T. Wilson and L. J. Gearing (Hudson Institute of Medical Research, Clayton, Australia) for RNA sequencing and help with analyses; Eurofins Panlabs Discovery Services Taiwan (New Taipei City, Taiwan) for generating mRNA LNPs and conducting associated in vivo studies; E. Pinto, I. Rudloff, and M. Tate (Hudson Institute of Medical Research, Clayton, Australia) for their help with PBMC studies; G. Ng (Hudson Institute of Medical Research, Clayton, Australia) for technical advice on multiplex cytokine analyses; B.R.G. Williams and S. Masters (Hudson Institute of Medical Research, Clayton, Australia) for comments on the manuscript; M. Kikkawa and Y. Sakamaki (The University of Tokyo, Tokyo, Japan) for managing and supporting the Graduate School of Medicine cryo-EM facility at the University of Tokyo. We acknowledge use of equipment and technical assistance of Monash Histology Platform (Department of Anatomy and Developmental Biology, Monash University, Clayton, Australia); use of equipment and technical support from RNAte (Hudson Institute of Medical Research, Clayton, Australia); the Monash Proteomics & Metabolomics Platform (Monash University, Clayton, Australia) for the provision of technical support and infrastructure, which has been enabled by Bioplatforms Australia and the National Collaborative Research Infrastructure Strategy; the Phenomics Translational Initiative teams for technical assistance with *in vivo* mouse studies including: K. Kwong and F-J. Li (The Australian National University, Acton, Australia) for the mouse intravenous and intraperitoneal injections and A. Davies and K. Diamand (The Australian National University, Acton, Australia) for cytokine and RT-qPCR measurements; and the Sydney Analytical Core Research Facility (University of Sydney, Sydney, Australia) for access to SPR infrastructure. This work was supported by the Australian National Health and Medical Research Council Ideas Grant (2020565 to M.P.G., J.I.E. and B.C.) and Centre for Research Excellence in Nucleic Acid Sensing (GNT2035500; M.P.G., J.I.E.); mRNA Victoria Research Acceleration Fund (M.P.G.), the Victorian Government's Operational Infrastructure Support Program and COVID-19 Treatments Medical Research Fund (M.P.G.); Grant-in-Aid from the Japanese Ministry of Education, Culture, Sports, Science, and Technology (grants 22K15046 and 24K09349 to Z.Z., 22H02556 to U.O., 22H05184 and 23H00366 to T.S.); CREST, JST (grant JPMJCR21E4 to T.S.); Open Philanthropy; the Francis Crick Institute (CC2228), which receives its core funding from Cancer Research UK, the UK Medical Research Council, and the Wellcome Trust (C.G.V.); and Noxopharm Limited (M.P.G., B.C.). Cryo-EM analyses were supported by the Basis for Supporting Innovative Drug Discovery and Life Science Research (BINDS) from the Japan Agency of Medical Research and Development (AMED) (grant JP21am0101115; support 1570, 1846, 1848) (T.S.). This research project was undertaken with the assistance of resources and services from the National Computational Infrastructure and the Pawsey Supercomputing Research Centre, which are supported by the Australian Government and the Government of Western Australia accessed through the National Merit Allocation and ANU Merit Allocation schemes (B.C.).

## Author contributions

Conceptualization was performed by M.P.G., S.S., A.S.A., O.F.L., D.S.W., R.G., M.S., R.J., B.C., Z.Z. and T.S. Investigation was carried out by A.S.A., S.S., R.J., Z.Z., U.O., W.S.N.J., E.R., L.C., L.W.W., R.G., J.I.E., A.L.M., R.R., R.G., M.S., D.S.W., R.M.S., M.R.A., L.Y., H.H.F., J.B., S.H., D.Y., O.F.L. and M.P.G. Resources were provided by C.R.S., K.A.L., P.H., C.G.V., M.A.B., U.O., O.F.L., D.S.W., C.A.N.P., M.S., B.C., T.S. and M.P.G. Data curation was conducted by A.L.M. and Z.Z. The original draft was written by M.P.G., B.C., Z.Z., U.O. and T.S.; review and editing of the manuscript were completed by S.S., M.S., O.F.L., R.J., D.S.W., E.R., W.S.N.J., S.H., J.I.E., H.H.F., L.W.W., A.L.M., D.Y., R.G., L.C., L.Y., Z.Z., U.O., C.G.V., K.A.L., C.R.S., C.A.N.P., M.A.B., T.S., B.C. and M.P.G. Supervision was provided by M.P.G., B.C. and T.S. Project administration was handled by A.S.A., S.S., Z.Z., U.O., R.J., M.S., R.G., O.F.L., B.C., T.S. and M.P.G. Funding acquisition was carried out by O.F.L., B.C., T.S. and M.P.G.

## Competing interests

O.F.L., D.S.W. and M.S. are employees of Noxopharm and Pharmorage. M.P.G. and B.C.'s groups receive funding from Noxopharm to study the activity of oligonucleotides on TLR7/8. M.P.G. receives consulting and advisory fees from Noxopharm. M.P.G. does not personally own shares and/or equity in Noxopharm. M.P.G., O.F.L., D.S.W., M.S. and S.S. are named inventors of a patent relating to the trimer oligonucleotide technology developed herein (WO2024077351). O.F.L. owns shares and/or equity in Noxopharm. The other authors declare no competing interests.

## Additional information

**Extended data** Extended data are available for this paper at https://doi.org/10.1038/s41590-026-02429-2.

**Correspondence and requests for materials** should be addressed to Sunil Sapkota or Michael P. Gantier.

**Peer review information** *Nature Immunology* thanks Gunther Hartmann and the other, anonymous, reviewer(s) for their contribution to the peer review of this work. Primary Handling Editor: Nick Bernard was the primary editor on this article and managed its editorial process and peer review in collaboration with the rest of the editorial team. Peer reviewer reports are available.

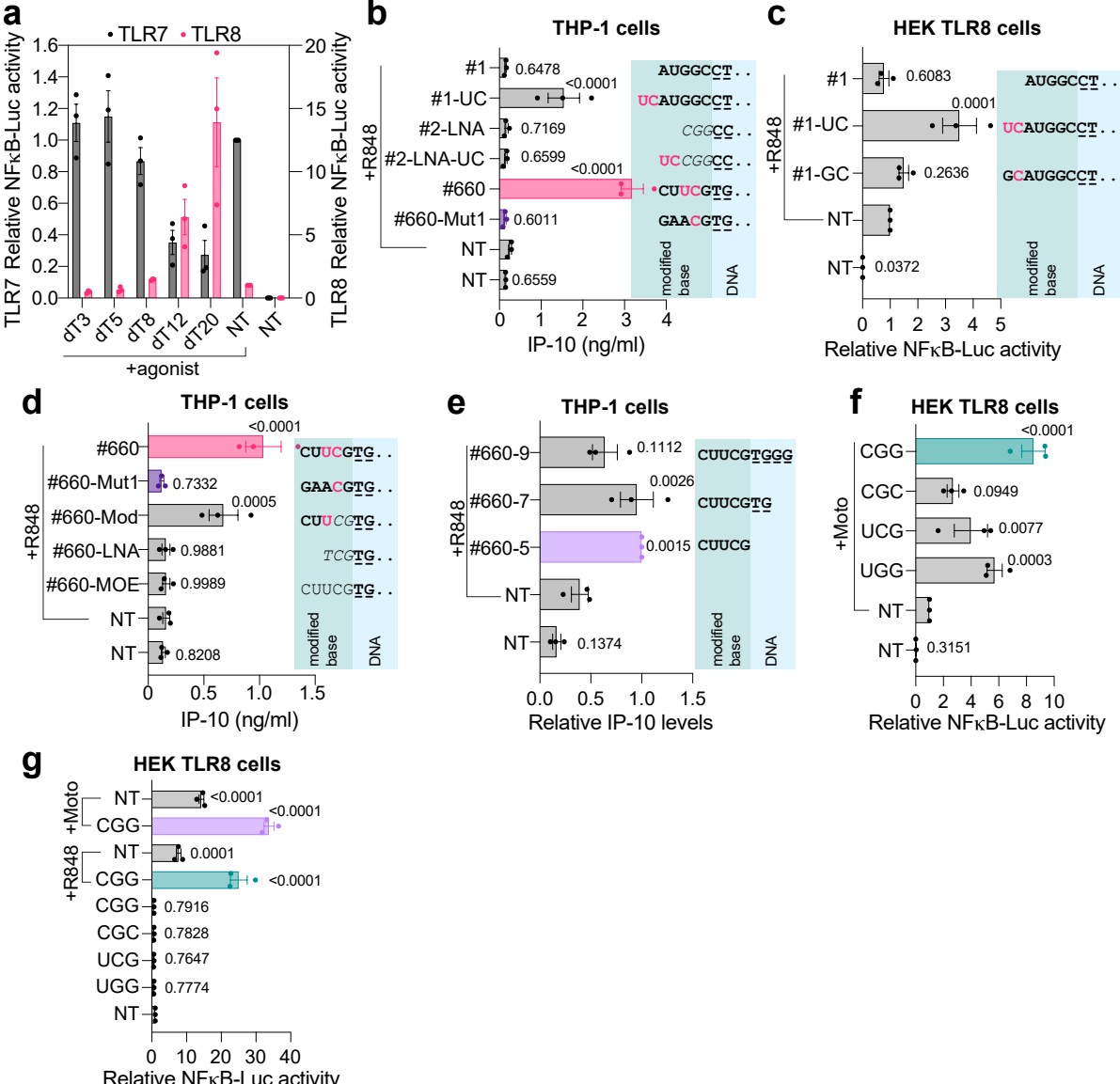

**Extended Data Fig. 1 | TLR8 potentiation by 3-mer oligos. (a)** HEK TLR8 (right Y-axis) and HEK TLR7 (left Y-axis) cells were pretreated for ~30 min with 1 μM dT$^{PS}$ series oligos prior to overnight stimulation with 1 μg/ml of R848 (TLR7) or 20 mM Uridine (TLR8) followed by luciferase assay. Data were background-corrected using the non-treated (NT) condition and are shown as expression relative to R848 or uridine only (± s.e.m.). **(b, d, e)** Monocytic THP-1 cells were incubated overnight with 100 nM of oligo **(b, d)** or 1 μM of oligo **(e)** and stimulated with 1 μg/ml of R848 for 7 h prior to IP-10 ELISA analysis (± s.e.m. and one-way ANOVA with uncorrected Fisher's LSD tests shown compared to the R848-only condition; **b, d**: P < 0.0001; **e**: P = 0.0006). **(e)** Data were normalized to #660-5 oligo condition. **(c, f, g)** HEK TLR8 cells were pretreated ~30 min with 500 nM **(c)** or 5 μM **(f, g)** of the indicated oligos prior to overnight stimulation with 1 μg/ml of

R848 **(c, g)** or 600 nM of motolimod (Moto) **(f, g)** followed by luciferase assay. **(c, f)** Data were background-corrected using the non-treated (NT) condition and are shown as relative expression to motolimod or R848 only (± s.e.m. and one-way ANOVA with uncorrected Fisher's LSD tests shown compared to the agonist-only condition; **c, f**: P < 0.0001). **(g)** Data are shown relative to the NT condition (± s.e.m. and one-way ANOVA with uncorrected Fisher's LSD tests shown compared to the NT condition; **g**: P < 0.0001). **(a-g)** Data are shown as mean of n = 3 independent experiments. All the oligos were PS modified. Oligos are modified as follows: bold pink or black is 2′-OMe, italic is LNA, non-bold is 2′-MOE. Pink highlights $_mU_mC$ motifs. DNA bases are underlined, and "· ·" denotes that the sequence is truncated – see Table S6 for full-length sequences. All statistics are available in Source Data Ext. Data Fig. 1.

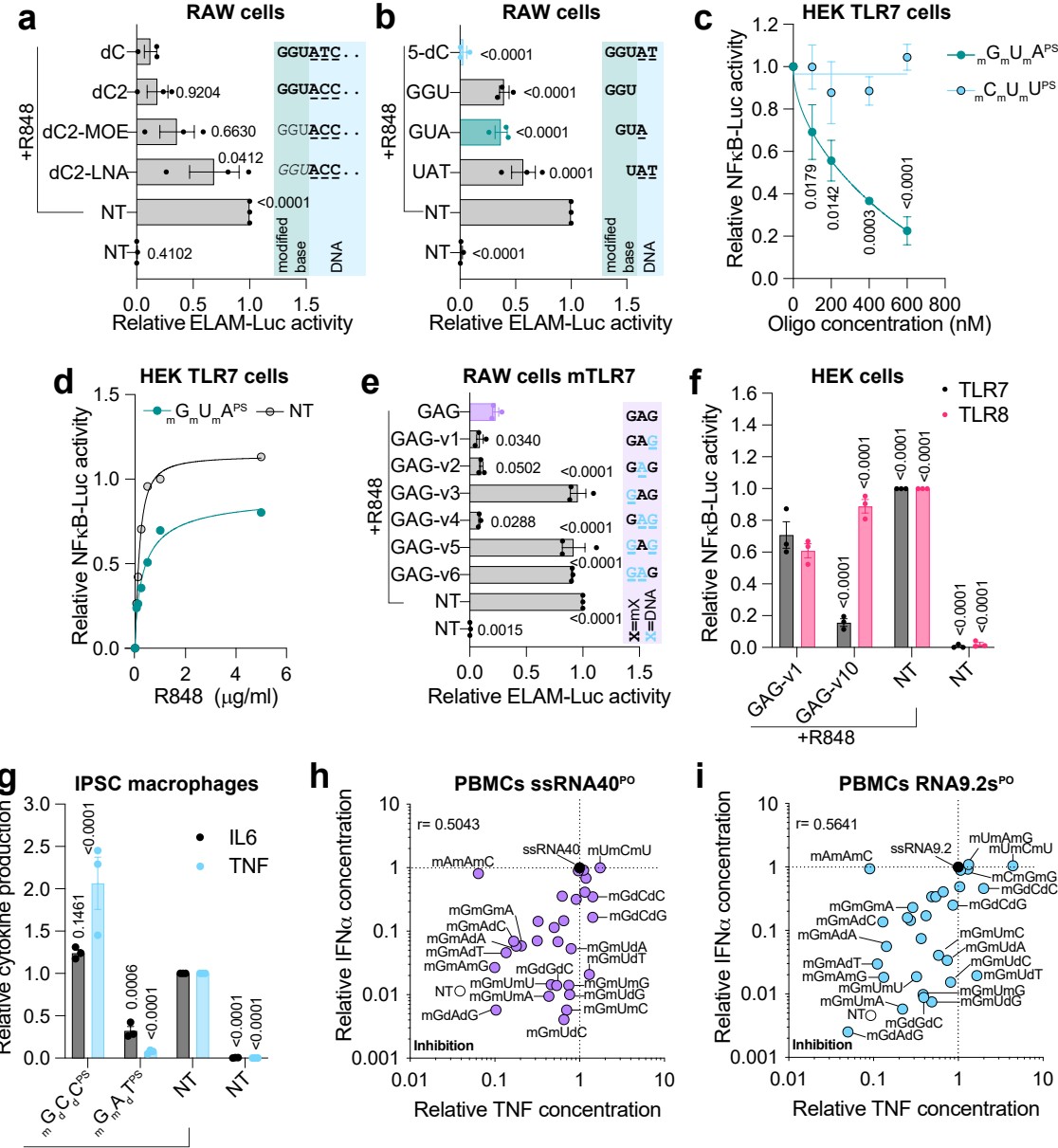

**Extended Data Fig. 2 | TLR7/8 modulation by 3-mer oligos. (a, b)** Mouse RAW264.7 cells stably expressing an ELAM-Luciferase reporter (RAW-ELAM) were pretreated for ~60 min with 1 μM (**a**) or 5 μM (**b**) of oligos, prior to overnight stimulation with 0.125 μg/ml R848 followed by luciferase assay. (**c, d**) HEK TLR7 cells were pretreated for ~30 min with the indicated concentration (0, 100, 200, 400 or 600 nM) (**c**) or 400 nM (**d**) of the oligos prior to overnight stimulation with 1 μg/ml (**c**) or indicated concentration (0, 0.062, 0.125, 0.25, 0.5, 1 μg/ml) of R848 followed by luciferase assay. (**e**) RAW-ELAM macrophages were pretreated for 1 h with 5 μM GAG[PS] variants prior to overnight stimulation with 0.125 μg R848 followed by luciferase assay. (**f**) HEK TLR8 and HEK TLR7 cells were pretreated for ~30 min with 5 μM (TLR8) or 1 μM (TLR7) GAG-v1[PS] or GAG-v10[PS] oligos prior to overnight stimulation with 1 μg/ml of R848 followed by luciferase assay. GAG-v10[PS] sequence is mGmAdGdCdCdC[PS]. (**g**) iPSC-derived macrophages were pretreated for 30 min with 5 μM of oligos prior to stimulation with 400 nM motolimod (Moto) for 6 h and IL-6 and TNF ELISAs. Cytokine levels were normalized to the motolimod-only condition. (**h-i**) PBMCs from healthy donors were incubated with 5 μM of 3-mers for ~60 min prior to overnight transfection of 400 nM ssRNA40[PO] (**h**) or RNA9.2s[PO] (**i**) with DOTAP. Cytokines

were measured using specific cytometric bead arrays (CBA). Data shown are averaged from 2 independent experiments in 3-blood donors, and are shown as expression relative to the ssRNA40[PO] (**h**) or RNA9.2s[PO] (**i**) conditions. Data were not corrected (**d, h, i**) or were background-corrected using the non-treated (NT) condition (**a, b, c, e**) and are shown as expression relative to the R848/ motolimod-only conditions (± s.e.m. and one-way [**a, b, e**] or two-way [**c, f, g**] ANOVA with uncorrected Fisher's LSD tests shown compared to dG [**a**], R848-only [**b, d**], mCmUmU[PS] [**c**], mGmAmC[PS] [**e**], GAG-v1[PS] + R848 [**f**] or Moto [**g**] conditions; a: P = 0.0001; b, e: P < 0.0001). (**h, i**) r values comparing TNF/IFNα relative levels are provided on each graph and correlation P values were **h:** P = 0.0033; **i:** P = 0.0008. Data are mean of n = 2 (**d, h, i**) or n = 3 (**a, b, c, e, f**) independent experiments. (**a, b**) Oligos as follows: bold is 2′-OMe, italic is LNA, non-bold is 2′-MOE. DNA bases are underlined, and "··" denotes that the sequence is truncated – see Table S6 for full-length sequences. (**e**) Black bases in bold denote 2′-OMe modification and light blue bases underlined denote DNA modifications. (**h, i**) mX is 2′-OMe-modified, and dX is DNA. All the oligos were PS modified unless otherwise indicated. All statistics are available in Source Data Ext. Data Fig. 2.

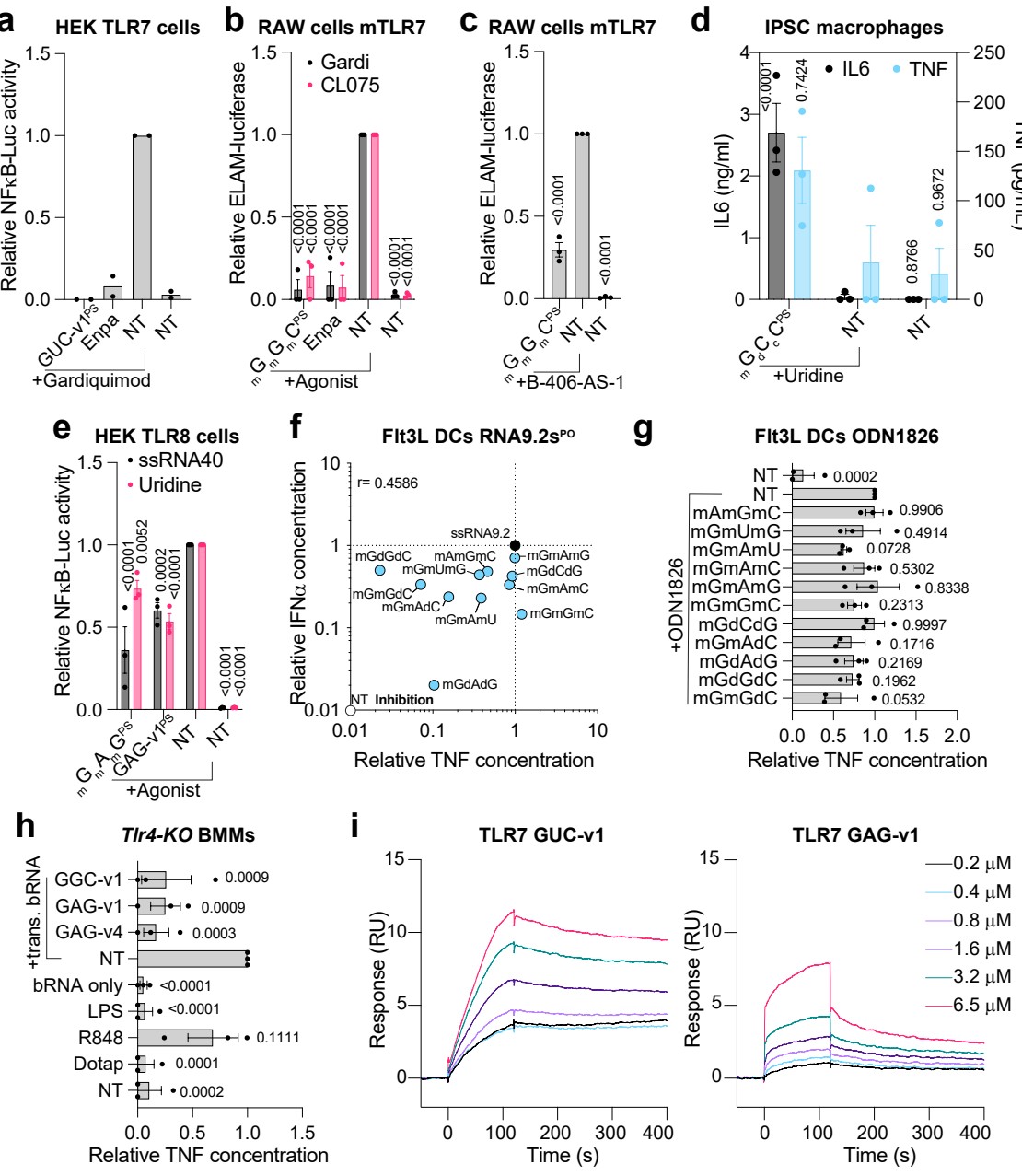

**Extended Data Fig. 3 | 3-mer oligos bind to TLR7/8 to modulate their response to agonists. (a)** HEK TLR7 cells were pretreated for ~30 min with 5 µM GUC-v1[PS] or 50 nM enpatoran (Enpa) prior to overnight stimulation with 1 µg/ml of gardiquimod followed by luciferase assay. **(b, c)** RAW-ELAM macrophages were pretreated for ~30 min with 5 µM **(b)** or 500 nM **(c)** mGmGmC[PS] prior to overnight stimulation with 0.5 µg/ml gardiquimod (Gardi) or CL075 **(b)** or DOTAP transfection with 500 nM of B-406-AS-1[PO] ssRNA **(c)** followed by luciferase assay. **(d)** iPSC-derived macrophages were pretreated 30 min with 5 µM of GCC-v4[PS] oligo prior to overnight stimulation with 20 mM uridine and IL-6 and TNF ELISAs. **(e)** HEK TLR8 cells were pretreated for ~120 min with 5 µM of mGmAmC[PS] or GAG-v1[PS] prior to overnight stimulation with 20 mM of uridine or DOTAP transfection with 1 µM of ssRNA40[PS] followed by luciferase assay. **(f-g)** Flt3L-derived DCs were pretreated for ~30 min with 5 µM of 3-mers prior to overnight transfection of 400 nM RNA9.2s[PO] with DOTAP **(f)** or stimulation with 500 nM ODN1826 **(g)**. Cytokines were measured using specific ELISAs. Data are shown as expression relative to the RNA9.2s[PO] **(f)** or ODN1826 **(g)** conditions (± s.e.m. and

one-way ANOVA with uncorrected Fisher's LSD tests shown compared to ODN1826 condition – also see Table S2). **(h)** Tlr4-deficient iBMDMs were pretreated for ~30 min with 5 µM of 3-mers prior to overnight transfection of 2 µg/ml purified bacterial RNA (bRNA) with DOTAP ( + trans) or without DOTAP (bRNA only), or stimulation with 1 µg/ml of R848 or 1 µg/ml of LPS. Data were not corrected **(d, f-h)** or were background-corrected **(a-c, e)** using the non-treated (NT) condition and are shown as expression relative to the agonist-only condition **(a-c, e-h)** (± s.e.m. and one-way [c, g, h] or two-way ANOVA [b, d, e] with uncorrected Fisher's LSD tests shown compared to agonist condition; **c:** P < 0.0001; **g:** P = 0.0102; **h:** P = 0.0007). Data are shown as mean of n = 2 **(a)** or n = 3 **(b-h)** independent experiments. **(f, g)** mX is 2′-OMe modified, and dX is DNA. **(i)** Surface plasmon resonance (SPR) analyses of recombinant mmTLR7 (with indicated concentrations of 3-mers). Data shown are representative of 5-6 independent analyses (Supplementary Table S3). All the oligos were PS modified unless otherwise indicated. All statistics are available in Source Data Ext. Data Fig. 3.

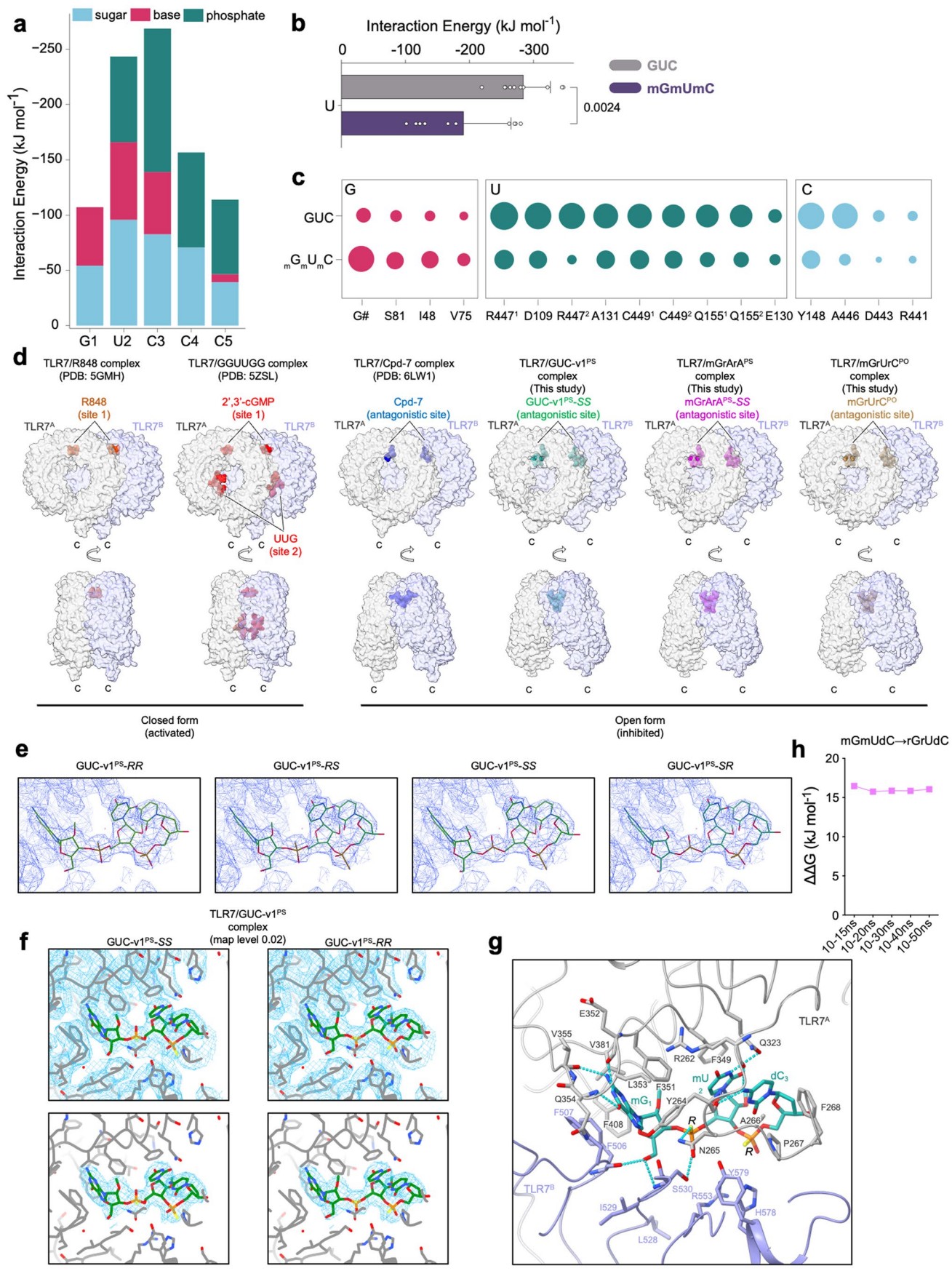

**Extended Data Fig. 4 | See next page for caption.**

**Extended Data Fig. 4 | GUC-v1 binds at the antagonist binding site of TLR7 leading to an open conformation.** (**a**) The decomposed interaction energy of $_rG_rU_rC_rC^{PO}$ with the protein residues that form binding site 2 from CphMD simulations at pH 5, suggesting the first three bases are essential for RNA binding and recognition. (**b**) The attractive interactions between the receptor and uridine are decreased by ~90 kJ mol$^{-1}$ with the uracil base leaving its binding site in $_mG_mU_mC^{PO}$ simulations. Data are shown as mean of n = 10 independent simulations (± standard deviation and two-sided unpaired t-test shown). (**c**) The overall percentage of interactions between the RNA bases and the protein residues that form binding site 2 retained during simulation are shown (the relative size of the circle represents the % time for observing each interaction pair formed along whole simulation trajectories). The intermolecular interactions between the nucleotides and receptor are decreased overall in $_mG_mU_mC^{PO}$ simulations at site 2 compared with $_rG_rU_rC^{PO}$. Labels on protein residues indicate different pairs of interactions between RNA ligands and residue side chains. (**d**) Structural comparison of various TLR7/ligand complex structures. In each structure, the two TLR7 protomers TLR7A and TLR7B and ligands are colored by gray, purple and different colors, respectively. (**e**) Views of ligand fitting of four GUC-v1$^{PS}$ stereoisomers into the cryo-EM map using COOT software. Cryo-EM map (level = 0.031) of the TLR7/GUC-v1$^{PS}$ complex and the fitted GUC-v1$^{PS}$ stereoisomers are shown in mesh and stick representations, respectively. (**f**) Full densities (upper panels) and zoned densities (within 2 Å of the ligands, lower panels) at the antagonistic sites of the TLR7/GUC-v1$^{PS}$ complex shown in mesh representations. Map contour values are shown in parentheses. (**g**) Overall structure of the TLR7/GUC-v1$^{PS}$-RR complex (upper). Close-up view of GUC-v1$^{PS}$-RR recognition at the antagonist binding site. (**h**) Convergence of the relative binding free energy difference between 3-mers $_mG_mU_dC^{PO}$ and $_rG_rU_dC^{PO}$ using different length of trajectories in free energy perturbation calculations (ΔΔG is 16.06 kJ.mol$^{-1}$ for $_rG_rU_dC^{PO}$ compared to $_mG_mU_dC^{PO}$, representing reduced binding affinity of $G_rU_dC^{PO}$ to the antagonist binding site of TLR7). (**h**) Data are shown as the mean of n=3 independent experiments. All statistics are available in Source Data Ext. Data Fig. 4.

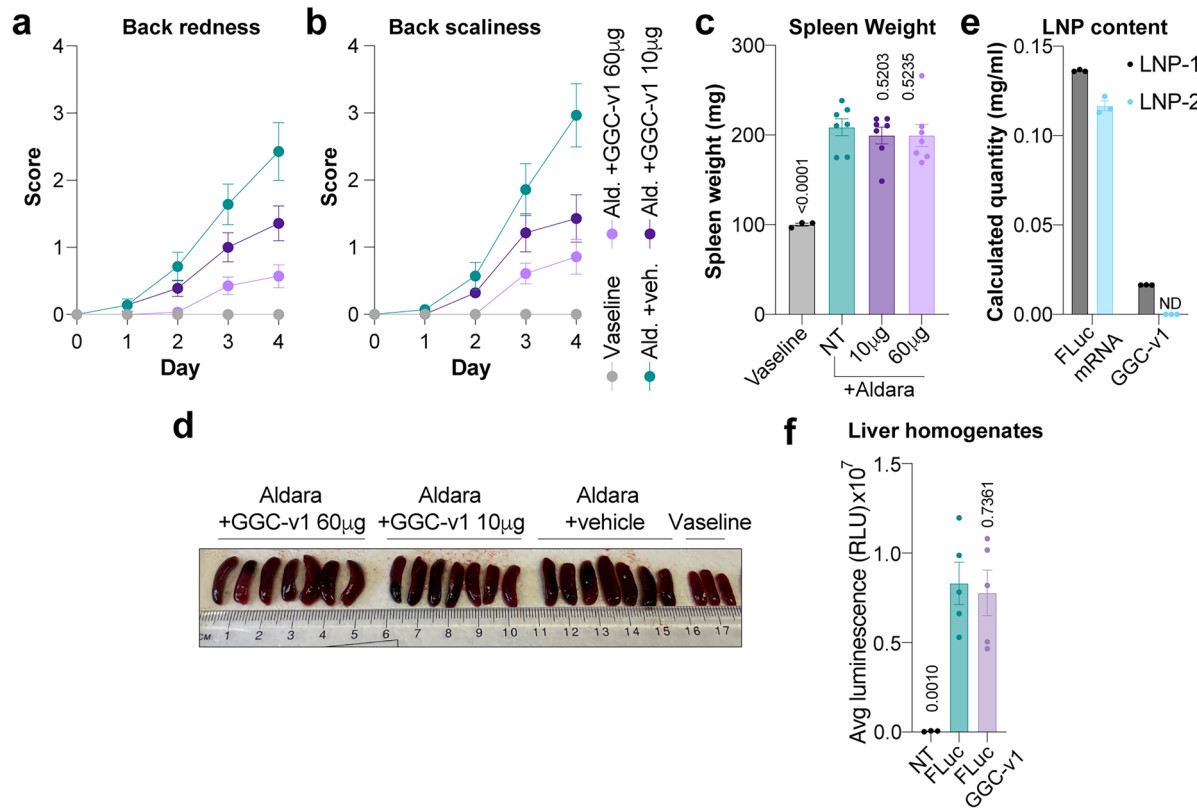

**Extended Data Fig. 5 | 2′-OMe 3-mer oligos act as TLR7 antagonists *in vivo*.** (**a**-**c**) WT C57/BL6 mice were treated with Aldara cream directly following, or not (n = 7 mice), application of 10 μg (n = 7 mice) or 60 μg (n = 7 mice) GGC-v1PS formulated in F127 Pluronic gel for four days. Non-Aldara treated control mice received Vaseline (n = 3 mice). Redness and scaliness were assessed by blinded investigators. After four days, mice were humanely euthanised and the spleens collected, weighed (**c**), and photographed (**d**). (**a**-**c**) Mean of n = 3 (Vaseline) or n = 7 (all other groups) mice/group are shown (± s.e.m. [**a**-**c**] and one-way ANOVA with uncorrected Fisher's LSD tests shown compared to Aldara-only group [**c**]; **c**: P < 0.0001). (**e**) RT-qPCR of FLuc mRNA and LC-MS quantification of GGC-v1PS

from two LNP preparations – both are absolute quantification with the standard curve for FLuc mRNA, or GGC-v1PS. ND: Not Detected at limit level of detection (10 ng/mL). LNP1 contains FLuc + GGC-v1PS and LNP2 contains FLuc mRNA only (data is averaged each from 3 replicate analyses of the same LNP preparation, (± s.e.m.). (**f**) Luciferase activity measured from liver homogenates at 24 h post injection. Mean of n = 3 (non-treated) and n = 5 (other groups) mice /group are shown (± s.e.m. and one-way ANOVA with uncorrected Fisher's LSD tests shown compared to FLuc mRNA-only group; **f**: P = 0.002). All statistics are available in Source Data Ext. Data Fig. 5.

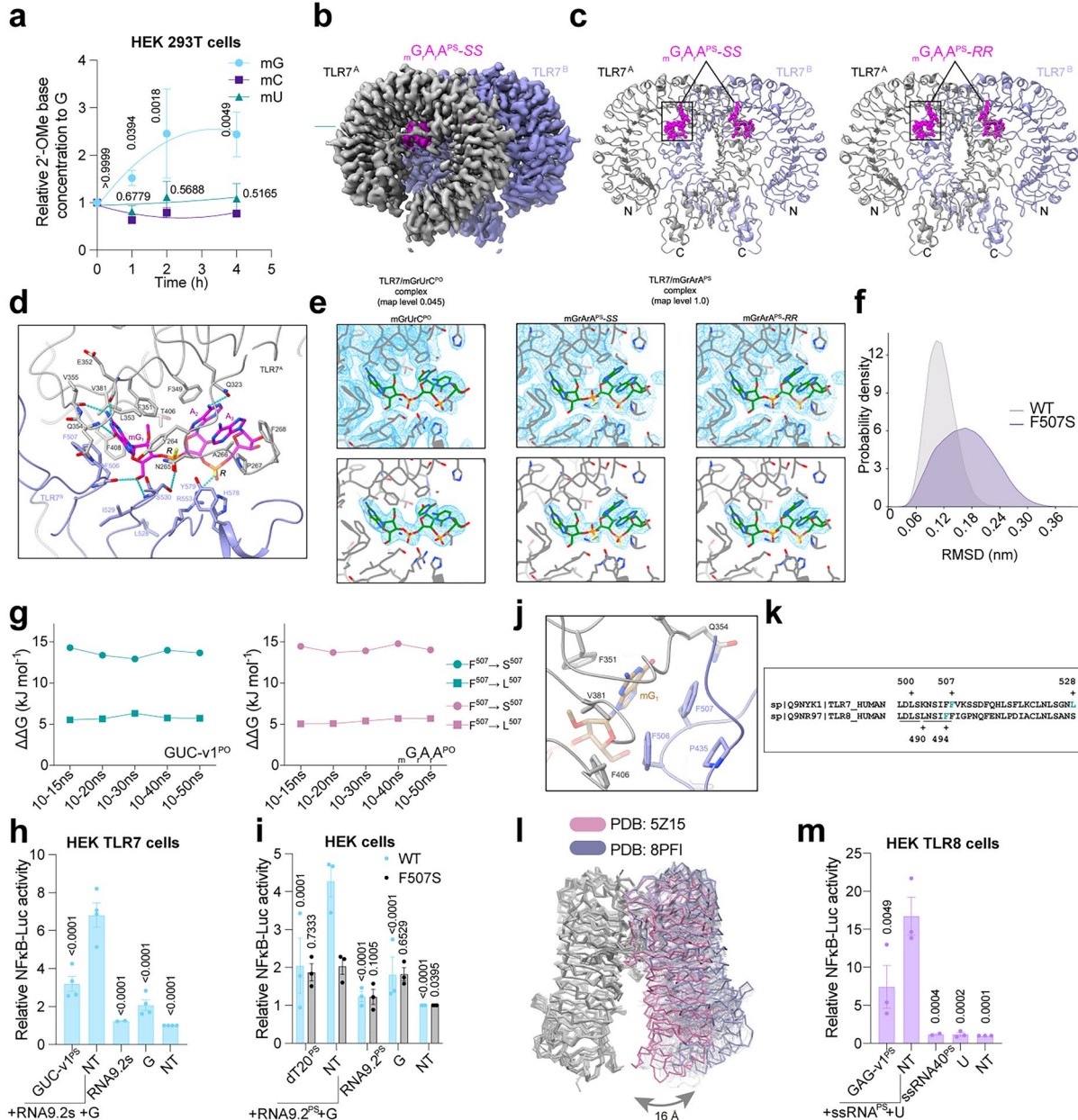

**Extended Data Fig. 6 | See next page for caption.**

**Extended Data Fig. 6 | 2′-OMe RNA fragments bind to a conserved TLR7/8 antagonistic pocket.** (**a**) HEK-293T cells were transfected with 5 μM of $_mG_rA_rA_rA^{PO}$ rRNA with DOTAP for indicated times and cells were pelleted and lysed directly in 0.5 M perchloric acid prior to $KHCO_3$ neutralization and LC-MS analyses (see Methods). 2′-OMe guanosine (mG), 2′-OMe cytosine (mC) and 2′-OMe uridine (mU) were robustly detected and quantified relative to the levels of guanosine. Data are shown as relative to the T = 0 h time point (± s.e.m and two-way ANOVA with uncorrected Fisher's LSD tests shown compared to the mC/G levels). (**b**, **c**) Unsharpened cryo-EM map (**b**), and structures (**c**) of the TLR7/$_mG_rA_rA^{PS}$-*SS* (left) and TLR7/$_mG_rA_rA^{PS}$-*RR* (right) complexes. The two TLR7 protomers and $_mG_rA_rA^{PS}$ are colored gray, purple, and magenta, respectively. (**d**) Close-up view of $_mG_rA_rA^{PS}$-*RR* (right) recognition at the antagonist binding site. Residues (within 4.5 Å from the ligand) are shown in stick representations and are colored by atom, with the N, O, and S atoms colored blue, red, and orange, respectively. Dashed lines in cyan indicate hydrogen bonds (cutoff distance < 3.5 Å). (**e**) Full densities (upper panels) and zoned densities (within 2 Å of the ligands, lower panels) at the antagonist binding site of the TLR7/$_mG_rA_rA^{PS}$-*SS* (middle), the TLR7/$_mG_rA_rA^{PS}$-*RR* complex (right) and the TLR7/$_mG_rU_rC^{PO}$ complex (left) shown in mesh representations. Map contour values are shown in parentheses. (**f**) The distribution of RMSD value of $_mG_1$ of GUC-v1$^{PS}$ in wild-type (gray) and F507S mutant (steel blue) in an MD simulation at a constant pH of 5. (**g**) Convergence of the relative binding free energy difference of the antagonistic 3-mers to F507S and F507L mutants using different length of trajectories in free energy perturbation calculations.

(**h**) HEK TLR7 cells were pretreated 30 min with 1 μM of GUC-v1$^{PS}$ prior to addition of 1 μM of naked RNA9.2$^{PS}$ for 1 h and overnight stimulation with 500 μM guanosine (**G**) and luciferase assay. (**i**) HEK cells stably expressing TLR7 F507S or WT were pretreated 30 min with 5 μM of (dT)$_{20}^{PS}$ prior to addition of 1 μM of naked RNA9.2$^{PS}$ for 1 h and overnight stimulation with 500 μM guanosine (**G**) and luciferase assay. (**j**) Close-up view of the P435 residue near the F506 and F507 residues in the TLR7/$_mG_rU_rC^{PO}$ complex. (**k**) Protein sequence alignments of human TLR7 and TLR8 antagonistic regions – the residues underlined are entirely conserved. TLR7 F507, L528, and TLR8 F494 are highlighted in blue. (**l**) The different conformations of an antagonist binding to inactive TLR8 dimers superimposed using single protomer chain with structures from PDB 5WYX, 5WYZ, 5Z14, 5Z15, 6KYA, 6TY5, 6V9U, 6ZJZ, 7CRF, 7R52, 7R53, 7R54, 7RC9, 7YTX, and 8PFI. The structures of inhibitory molecules bound in the most compact inactive dimer (5Z15, pink) and the most open inactive dimer (8PFI, steel blue) are shown on the right, with $_mG_rA_rA^{PO}$ docking best to the most open dimer. (**m**) HEK TLR8 cells were pretreated 30 min with 5 μM of GAG-v1$^{PS}$ prior to addition of 1 μM of naked ssRNA40$^{PS}$ for 1 h and overnight stimulation with 2.5 mM uridine (**U**) followed by luciferase assay. (**h**, **i**, **m**) Data are shown normalized to NT condition (± s.e.m. and one-way [**h**, **m**] or two-way [**i**] ANOVA with uncorrected Fisher's LSD tests shown compared to the RNA9.2$^{PS}$ + G condition [**h**, **i**], or ssRNA40$^{PS}$ + U condition [**m**]; **h**: P < 0.0001; **m**: P = 0.0006). (**a**, **g**, **h**, **i**, **m**) Data are shown as the mean of n=3 independent experiments. All statistics are available in Source Data Ext. Data Fig. 6.

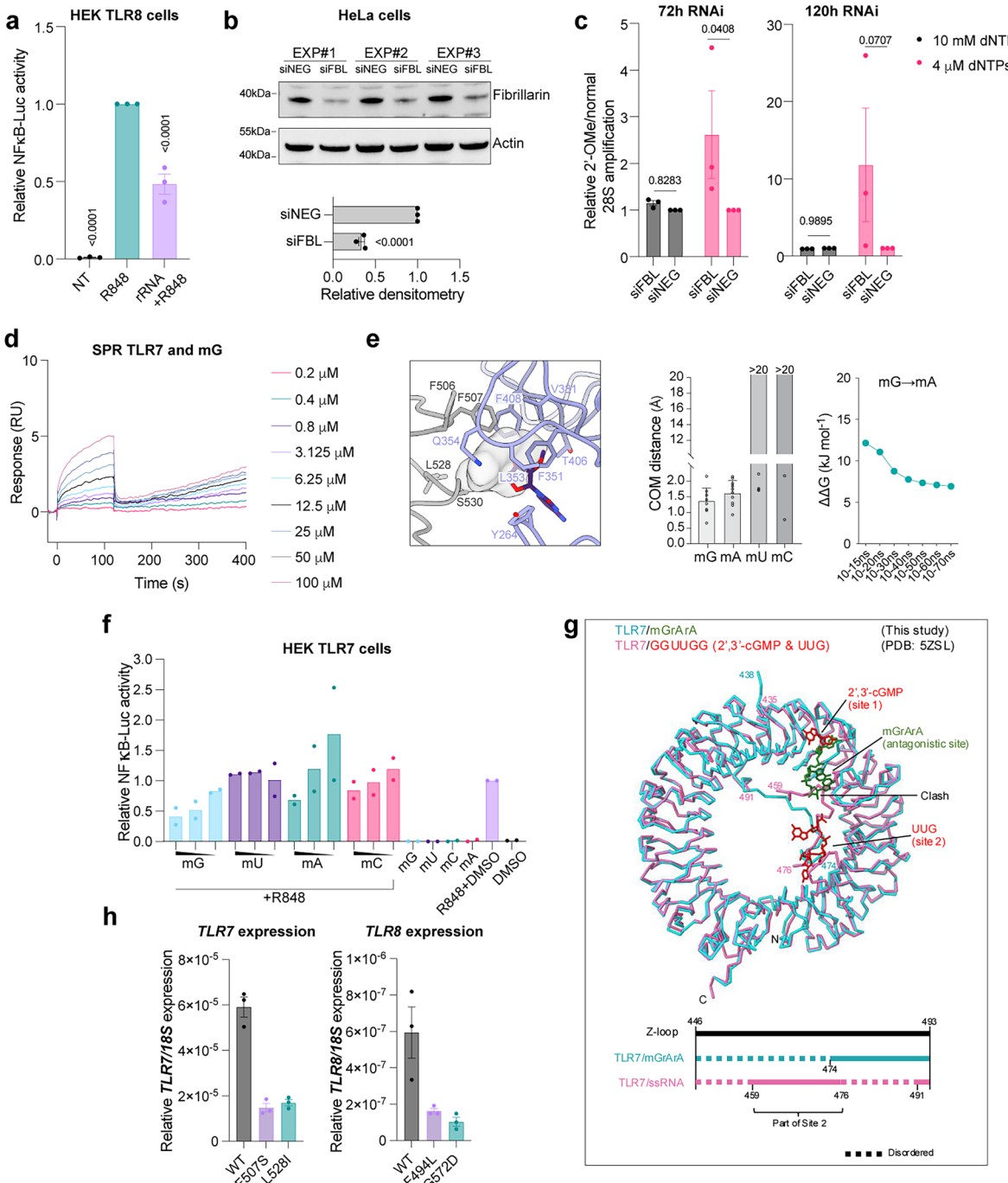

**Extended Data Fig. 7 | See next page for caption.**

**Extended Data Fig. 7 | Endogenous 2′-OMe ribosomal RNA fragments act as natural antagonists of TLR7/8.** (**a**) HEK TLR8 cells were transfected for 6 h with 3 µg/ml purified rRNA prior to overnight stimulation with 1 µg/ml of R848 followed by luciferase assay. (**b**) HeLa cells were transfected for 72 h with 40 nM of siFibrillarin (siFBL) or control siRNA (siNEG) prior to western blotting of the cell lysates. Blots of anti-fibrillarin and anti-actin are shown and relative densitometry of Fibrillarin to Actin reported to siNEG conditions are shown (± s.e.m. and two-sided unpaired t-test are shown). Each pair of siFBL and siNEG samples are from an independent experiment. (**c**) RNA from siFBL or siNEG treated HeLa cells was purified at 72 or 120 h and analyzed by 2′-OMe specific 28S RT-qPCR method (see Methods). Increased amplification at low dNTP (4 µM) versus high dNTP (10 mM) correlates with a loss of 2′-OMe mark in the targeted region. Relative amplification of 2′-OMe versus non 2′-OMe target sites in 28S is shown normalized to siNEG condition (± s.e.m. and two-way ANOVA with uncorrected Fisher's LSD tests shown comparing siNEG to siFBL conditions). (**d**) SPR analyses of recombinant mmTLR7 (with indicated concentrations of 2′-OMe guanosine). Data shown are representative of 3 independent analyses. (**e**) MD analyses of single 2′-OMe base binding to the antagonistic site of TLR7. (Left panel) Antagonist binding site of human TLR7 in complex of single nucleotide, $_mC$, after 400 ns MD simulations. Two protomers are colored in purple and gray, respectively. The molecular surface shown in transparent light gray was obtained using MD simulations of an $_mC$ moiety binding TLR7, inferred by mutating the first $_mG$ from the GUV-v1$^{PS}$ structure into $_mC$. (**e**)(middle panel) The averaged center-of-mass (COM) distance profile between single nucleotide ligands ($_mG$, $_mA$, $_mU$ and $_mC$) at the end of 400 ns MD simulations with the antagonist binding site comprised by residues F506, F507, L528, I529, S530, Q531, T532, R553 from one protomer and Y264, N265, F349, F351, E352, L353, Q354, V355, Y356, G379, V381, T406, N407, F408 from the other protomer (data are shown as mean of n = 10 independent simulations ± standard deviation; points above 20 are not shown). (**e**) (Right panel) Convergence of the relative binding free energy difference between single nucleotide $_mG$ and $_mA$ using different length of trajectories in free energy perturbation calculations (ΔΔG is 6.94 kJ. mol$^{-1}$ for $_mA$ compared to $_mG$, representing reduced binding affinity of $_mA$ to the antagonist binding site of TLR7). (**f**) HEK 293 cells stably expressing WT TLR7 were treated with 500, 250 or 125 µM of indicated 2′-OMe nucleosides prior to overnight stimulation with 1 µg/ml R848 followed by luciferase assay. (**g**) Structural comparison of the protomer structures from the TLR7$_{/mG_rA_rA}$-*SS* complex and the TLR7$_{/rG_rG_rU_rU_rG_rG}^{PO}$ complex (PDB: 5ZSL). One TLR7 protomer in each dimer structure was aligned using the ChimeraX matchmaker tool and is shown in main chain trace. The ordered and disordered regions of Z-loop in each structure are indicated at the bottom. (**h**) HEK cells stably expressing TLR7 WT or mutants, or TLR8 and mutants were lysed and specific expression of TLR7 and TLR8 relative to 18S was measured by RT-qPCR (data is averaged from 3 independent cell lysates from the same cell line - ± s.e.m.). (**a**, **f**) Data were background-corrected using the NT condition and are shown as expression relative to the R848-only condition [a] or R848 + DMSO condition [f] (± s.e.m. and one-way ANOVA with uncorrected Fisher's LSD tests shown compared to R848-only condition [a]; a: P < 0.0001. Data are shown as the mean of n=3 (**a**, **b**, **c**, **e**) or n = 2 (**f**) independent experiments. All statistics are available in Source Data Ext. Data Fig. 7.

**a**

TLR7/GUC-v1$^{PS}$ complex

**6,615 movie stacks (pixel size = 0.83 Å)**

(Data processing with Relion)

↓ **Motion correction** (MotionCor2)

↓ **CTF estimation** (CTFFIND-4.1)

↓ **Auto-picking** (Laplacian-of-Gaussian, LoG filter 80/120 Å)

↓ **Extraction** (box = 240 pixels, re-scaled to 48 pixels)

**~7,993k particles**

↓ **2D classification** (40 subsets) (Ignore CTFs until first peak, 30 classes, T = 2, 100 VDAM mini-batches, mask 150 Å)

↓ **Re-extraction** (box = 240 pixels, re-scaled to 100 pixels)

**~3,050k particles**

↓ **3D classification** (6 subsets) (Reference map: EMD-35258; C1, initial low pass filter 20 Å, 3 classes, T =4 , 50 iterations, mask 150 Å)

**~1,570k particles**

↓ **3D classification** (3 subsets) (C1, initial low pass filter 20 Å, 3 classes, T =4, 50 iterations, mask 150 Å)

↓ **Re-extraction** (box = 300 pixels, re-scaled to 192 pixels)

**~1,219k particles**

↓ **3D classification** (2 subsets) (C2, initial low pass filter 15 Å, 3 classes, T =4, 50 iterations, mask 150 Å)

**686,274 particles**

↓ **3D auto-refine and Post-processing** (C2, initial low pass filter 20 Å, mask 150 Å, solvent-flattened FSCs) → 3.2 Å

↓ **2D classification** (for assessment purpose only) (50 classes, T = 2, 200 VDAM mini-batches, mask 150 Å)

↓ **Bayesian polishing** (default own parameters; box = 300 pixels, re-scaled to 192 pixels)

↓ **3D auto-refine** (C2, initial low pass filter 20 Å, mask 150 Å, solvent-flattened FSCs) → 3.2 Å

↓ **3D classification** (C2, initial low pass filter 15 Å, 2 classes, T = 4, 25 iterations, mask 150 Å, no image alignment)

**173,248 particles**

↓ **3D auto-refine and Post-processing** (C2, initial low pass filter 20 Å, mask 150 Å, solvent-flattened FSCs) → 3.1 Å

↓ **Bayesian polishing** (s_vel: 0.807, s_div: 7635, s_acc: 4.59; box = 300 pixels, re-scaled to 192 pixels)

↓ **3D auto-refine and Post-processing** (C2, initial low pass filter 20 Å, mask 150 Å, solvent-flattened FSCs) → 3.1 Å

↓ **CTF refinement** (fit defocus/astigmatism/B-factor per-micrograph, estimate beamtilt/trefoil)

↓ **3D auto-refine and Post-processing** (C2, initial low pass filter 20 Å, mask 150 Å, solvent-flattened FSCs) → 3.0 Å

↓ **CTF refinement** (estimate anisotropic magnification, fit defocus/astigmatism/B-factor per-micrograph, estimate beamtilt/trefoil)

↓ **3D auto-refine and Post-processing** (C2, initial low pass filter 20 Å, mask 150 Å, solvent-flattened FSCs) → 3.0 Å

↓ **Bayesian polishing** (box = 300 pixels, re-scaled to 192 pixels)

↓ **3D auto-refine and Post-processing** (C2, initial low pass filter 20 Å, mask 150 Å, solvent-flattened FSCs) → 3.0 Å

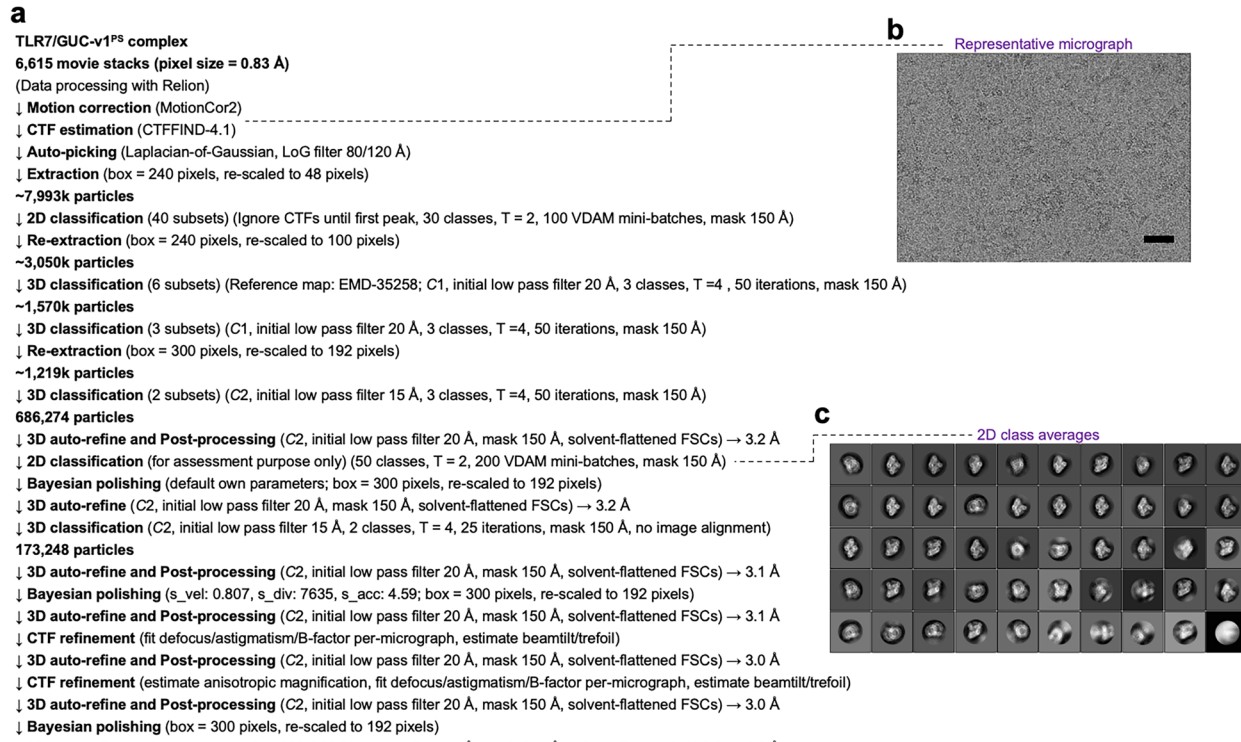

**b** Representative micrograph

**c** 2D class averages

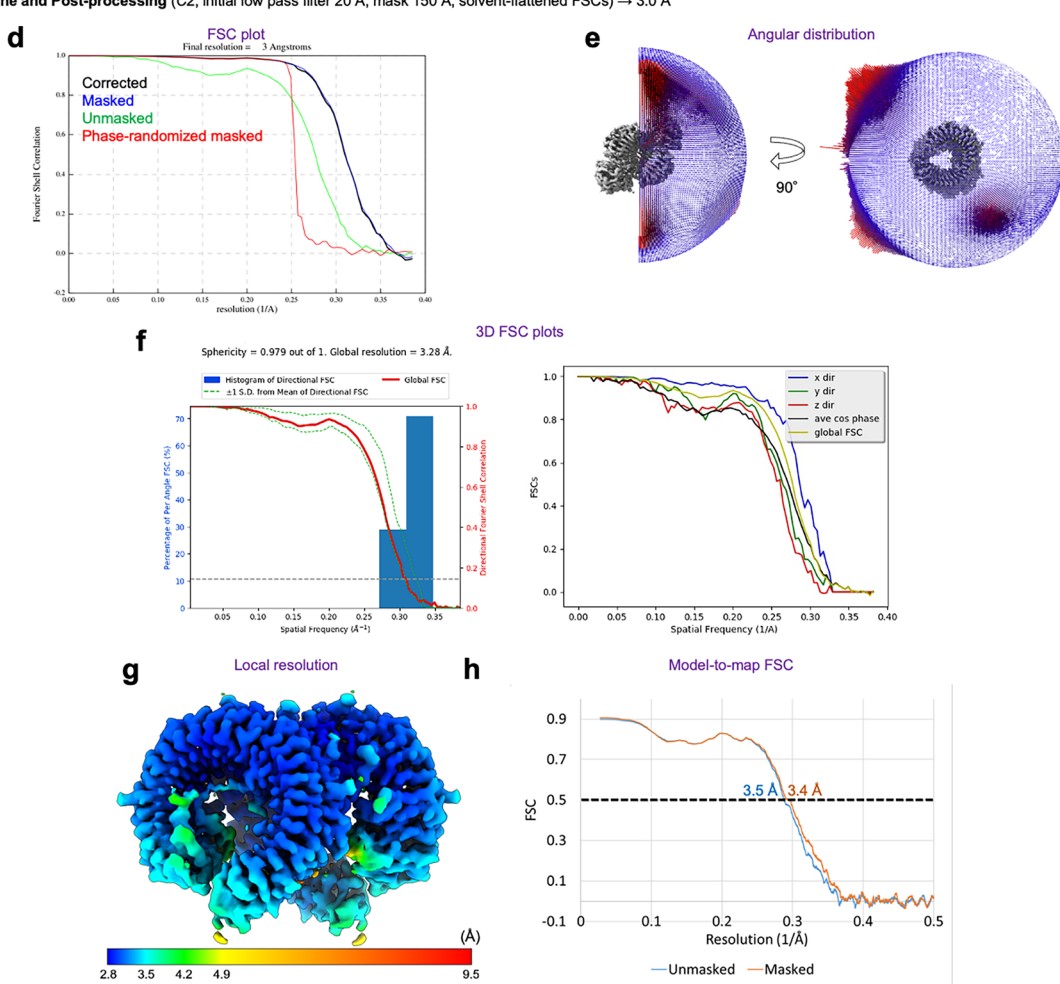

**d** FSC plot
Final resolution = 3 Angstroms

Corrected
Masked
Unmasked
Phase-randomized masked

**e** Angular distribution

90°

**f** 3D FSC plots

Sphericity = 0.979 out of 1. Global resolution = 3.28 Å.

Histogram of Directional FSC — Global FSC
±1 S.D. from Mean of Directional FSC

x dir
y dir
z dir
ave cos phase
global FSC

**g** Local resolution

(Å)
2.8  3.5  4.2  4.9  9.5

**h** Model-to-map FSC

3.5 Å  3.4 Å

Unmasked  Masked

**Extended Data Fig. 8 | Cryo-EM data processing of the TLR7/GUC-v1$^{PS}$ complex.** (**a**) Data processing workflow of cryo-EM analysis of the TLR7/GUC-v1$^{PS}$ complex. (**b**) Representative motion-corrected micrographs (out of 6,615 total micrographs). Black bar, 50 nm. (**c**) 2D class averages. (**d**) Fourier shell correlation (FSC) plot of the final 3D reconstruction (resolution cutoff at FSC = 0.143) (**e**) Angular distribution of the particles for final 3D reconstruction. (**f**) 3D FSC plots. (**g**) Local resolution of the final 3D reconstruction. (**h**) Model-to-map FSC plot of the TLR7/GUC-v1$^{PS}$-SS structure.

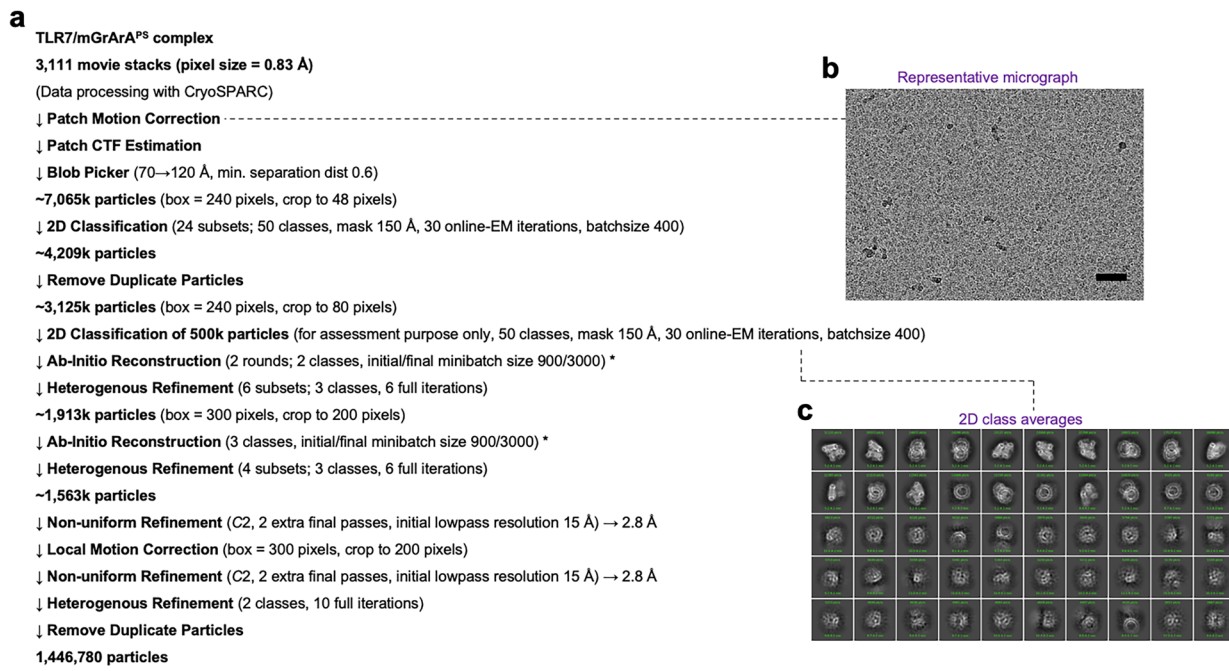

**a**

TLR7/mGrArA$^{PS}$ complex

**3,111 movie stacks** (pixel size = 0.83 Å)

(Data processing with CryoSPARC)

↓ **Patch Motion Correction** ----------------------

↓ **Patch CTF Estimation**

↓ **Blob Picker** (70→120 Å, min. separation dist 0.6)

**~7,065k particles** (box = 240 pixels, crop to 48 pixels)

↓ **2D Classification** (24 subsets; 50 classes, mask 150 Å, 30 online-EM iterations, batchsize 400)

**~4,209k particles**

↓ **Remove Duplicate Particles**

**~3,125k particles** (box = 240 pixels, crop to 80 pixels)

↓ **2D Classification of 500k particles** (for assessment purpose only, 50 classes, mask 150 Å, 30 online-EM iterations, batchsize 400)

↓ **Ab-Initio Reconstruction** (2 rounds; 2 classes, initial/final minibatch size 900/3000) *

↓ **Heterogenous Refinement** (6 subsets; 3 classes, 6 full iterations)

**~1,913k particles** (box = 300 pixels, crop to 200 pixels)

↓ **Ab-Initio Reconstruction** (3 classes, initial/final minibatch size 900/3000) *

↓ **Heterogenous Refinement** (4 subsets; 3 classes, 6 full iterations)

**~1,563k particles**

↓ **Non-uniform Refinement** (*C2*, 2 extra final passes, initial lowpass resolution 15 Å) → 2.8 Å

↓ **Local Motion Correction** (box = 300 pixels, crop to 200 pixels)

↓ **Non-uniform Refinement** (*C2*, 2 extra final passes, initial lowpass resolution 15 Å) → 2.8 Å

↓ **Heterogenous Refinement** (2 classes, 10 full iterations)

↓ **Remove Duplicate Particles**

**1,446,780 particles**

↓ **Non-uniform Refinement** (*C2*, 2 extra final passes, initial lowpass resolution 15 Å, optimize per-group CTF params, EDS correction) → 2.7 Å

* For generation of 3D maps as initial volumes for heterogenous refinements

**b** Representative micrograph

**c** 2D class averages

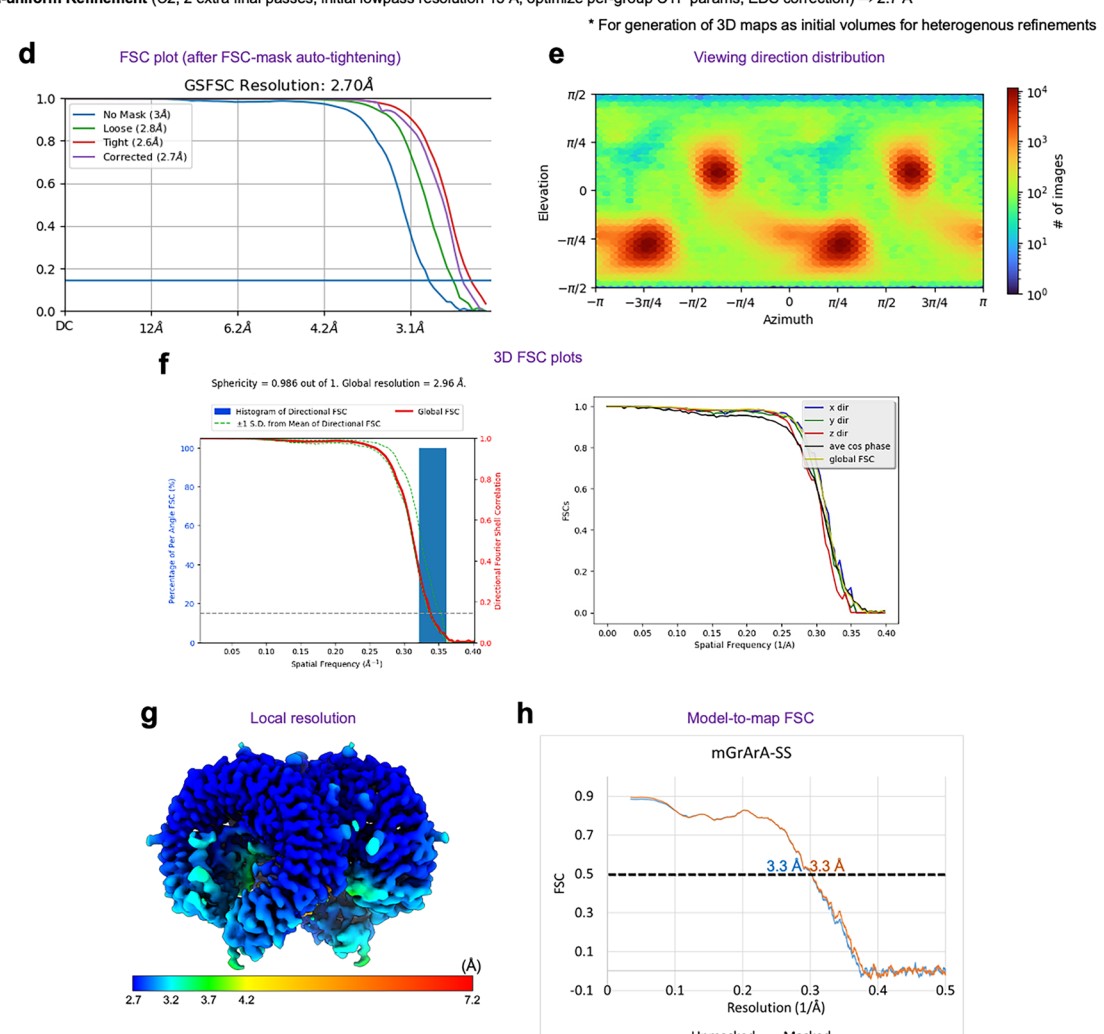

**d** FSC plot (after FSC-mask auto-tightening)

**e** Viewing direction distribution

**f** 3D FSC plots

**g** Local resolution

**h** Model-to-map FSC

**Extended Data Fig. 9 | Cryo-EM data processing of the TLR7/$_m$G$_r$A$_r$A$^{PS}$ complex.** (**a**) Data processing workflow of cryo-EM analysis of the TLR7/$_m$G$_r$A$_r$A$^{PS}$ complex. (**b**) Representative motion-corrected micrographs (out of 3,111 total micrographs). Black bar, 50 nm. (**c**) 2D class averages. (**d**) Fourier shell correlation (FSC) plot of the final 3D reconstruction (resolution cutoff at FSC = 0.143) (**e**) Viewing direction distribution of the particles for final 3D reconstruction. (**f**) 3D FSC plots. (**g**) Local resolution of the final 3D reconstruction. (**h**) Model-to-map FSC plot of the TLR7/$_m$G$_r$A$_r$A$^{PS}$-SS structure.

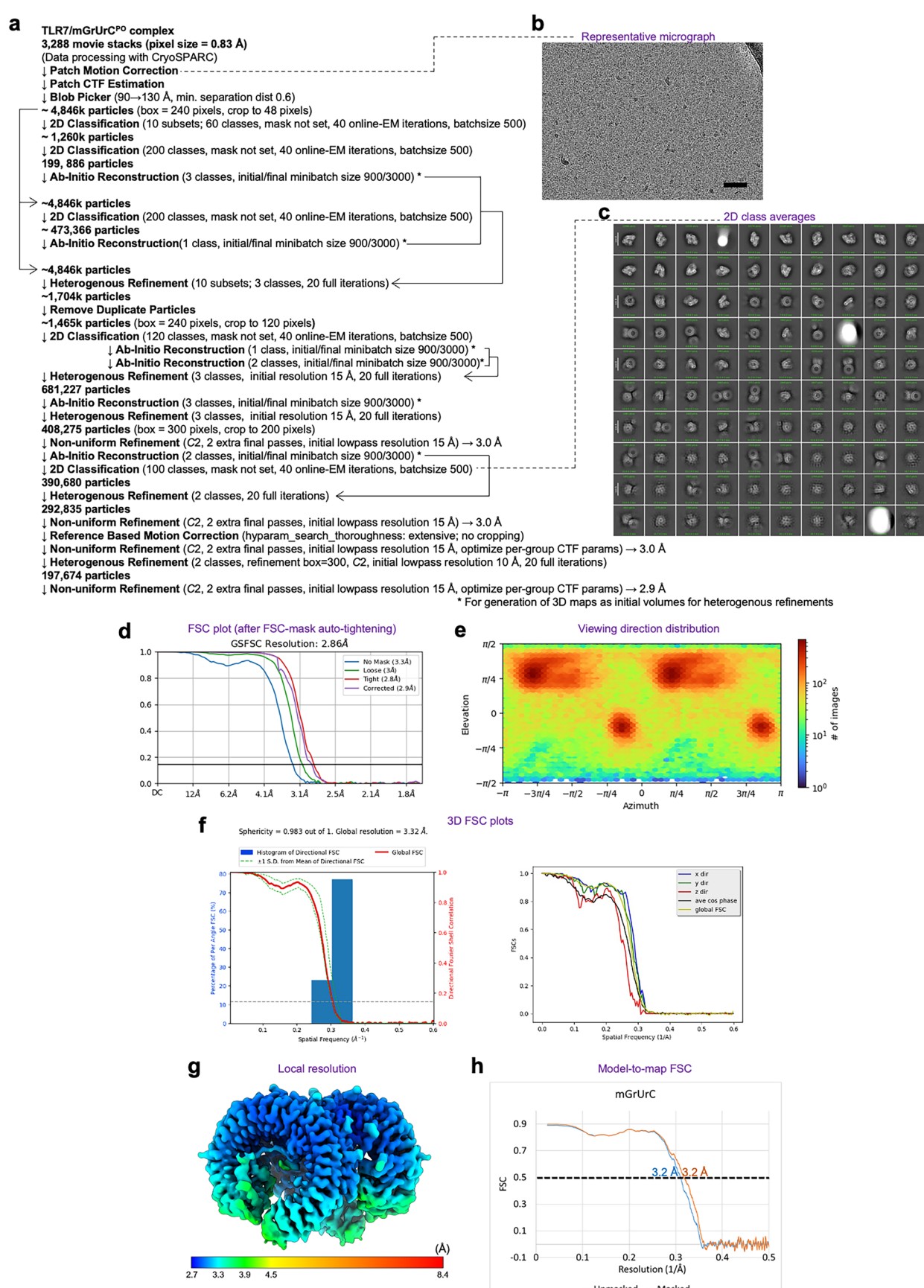

**Extended Data Fig. 10 | Cryo-EM data processing of the TLR7/$_m$G$_r$U$_r$C$^{PO}$ complex.** (**a**) Data processing workflow of cryo-EM analysis of the TLR7/$_m$G$_r$U$_r$C$^{PO}$ complex. (**b**) Representative motion-corrected micrographs (out of 3,288 total micrographs). Black bar, 50 nm. (**c**) 2D class averages. (**d**) Fourier shell correlation (FSC) plot of the final 3D reconstruction (resolution cutoff at FSC = 0.143) (**e**) Viewing direction distribution of the particles for final 3D reconstruction. (**f**) 3D FSC plots. (**g**) Local resolution of the final 3D reconstruction. (**h**) Model-to-map FSC plot of the TLR7/$_m$G$_r$U$_r$C$^{PO}$ structure.

# Reporting Summary

## Statistics

For all statistical analyses, confirm that the following items are present in the figure legend, table legend, main text, or Methods section.

| n/a | Confirmed | |
|---|---|---|
| ☐ | ☒ | The exact sample size (*n*) for each experimental group/condition, given as a discrete number and unit of measurement |
| ☐ | ☒ | A statement on whether measurements were taken from distinct samples or whether the same sample was measured repeatedly |
| ☐ | ☒ | The statistical test(s) used AND whether they are one- or two-sided<br>*Only common tests should be described solely by name; describe more complex techniques in the Methods section.* |
| ☒ | ☐ | A description of all covariates tested |
| ☐ | ☒ | A description of any assumptions or corrections, such as tests of normality and adjustment for multiple comparisons |
| ☐ | ☒ | A full description of the statistical parameters including central tendency (e.g. means) or other basic estimates (e.g. regression coefficient) AND variation (e.g. standard deviation) or associated estimates of uncertainty (e.g. confidence intervals) |
| ☐ | ☒ | For null hypothesis testing, the test statistic (e.g. *F*, *t*, *r*) with confidence intervals, effect sizes, degrees of freedom and *P* value noted<br>*Give P values as exact values whenever suitable.* |
| ☒ | ☐ | For Bayesian analysis, information on the choice of priors and Markov chain Monte Carlo settings |
| ☒ | ☐ | For hierarchical and complex designs, identification of the appropriate level for tests and full reporting of outcomes |
| ☒ | ☐ | Estimates of effect sizes (e.g. Cohen's *d*, Pearson's *r*), indicating how they were calculated |

*Our web collection on statistics for biologists contains articles on many of the points above.*

## Software and code

Policy information about availability of computer code

Data collection

1. For SPR analyses, a Biacore T200 (Cytiva) was used
2. For ELISA and luminescence collection with Fluostar OPTIMA the OPTIMA-Control v2.2R2 software was used.
3. For LegendPlex TNF analyses, Sample acquisition was performed using a BD LSR-II flow cytometer (BD Biosciences)
4. LC-MS/MS analyses of 3-mer in LNP were conducted on Waters Acquity premier UPLC system (Waters Corporation, Milford, MA) coupled QTRAP 6500 mass spectrometer (Sciex, Framingham, MA)
5. For 2'OMe nucleoside detection, LC-MS/MS analyses were performed using a Shimadzu LC-30AD binary pump system coupled to a hybrid triple quadrupole/linear ion trap mass spectrometer (QTRAP 5500). Chromatographic separation was achieved using a Synergi Hydro-RP.
6. RT-qPCR were conducted on QuantStudio 6 Flex RT-PCR system (Thermo Fisher)
7. IVIS Spectrum was used for in vivo luciferase quantification
8. A Bio-Rad ChemiDoc was used for Western blot imaging
9. serum quantification of IFN-by ELISA was analysed on TECAN Infinite F50
10. BIO-RAD Bio-Plex 200 Luminex was used for 23-plex assay analyses
11. Cryo-EM movies were recorded by using a Titan Krios G4 microscope equipped with a Gatan Quantum-LS Energy Filter
12. VS120 Slide Scanning System (Olympus) for scanning histology slides
13. Analyses of PBMC CBA were conducted with Attune NxT (Thermo Fisher).
14. LNP luciferase qPCR was carried out on QuantStudio Design
15. Applied Biosystems 7900 machine (Thermo Fisher Scientific) for R848 qPCR experiments
16. Applied Biosystems QuantStudio 3 machine for RTL-P assay PCRs

Data analysis

Software
The software used below match the equipments above.
1. For SPR analyses, the Biacore T200 Evaluation Software Version 3.2 (Cytiva) was used.
2. MARS Data analysis software 3.01R2 (BMG Labtech) was used for luminescence and absorbance analyses.
3. Legendplex data were analysed with the LEGENDplex™ Data Analysis Software Suite (BioLegend)
4. LNP LC-MS/MS analyses of 3mers were performed on Analyst Software v 1.6.3.
5. LC-MS/MS Data analysis of nucleosides was conducted using MultiQuant Software™ 2.0
6. QuantStudio™ Real-Time PCR Software v1.7.2 for qPCR analyses from QuantStudio 6 Flex
7. Living Image™ software v4.5.2 for IVIS analyses
8. For Western blot analyses, the ImageLab software v6.1 was used.
9. Tecan i-control software for analyses of IFN ELISA.
10. Bio-Plex Manager Software 6.0 was used for 23-plex analyses.
11. For processing the TLR7/GUC-v1PS dataset, RELION v4.0.1 was used. The CTF parameters were determined using the CTFFIND4. For processing the TLR7/mGrArAPS dataset, cryoSPARC v4.0.3 was used. The TLR7/mGrUrCPO dataset was processed in cryoSPARC v4.4.0.
12. IImageJ 1.53 was used to analyse histological microscopy images.
13. FlowJo Sowftare v10.9 (BD) was used for analyses of CBA results on Attune
14. Analysis Software v1.5.2 for Luciferase qPCR was used for luciferase qPCRs
15. 7900 SDS v2.4.1 software for R848 qPCR experiments
16. QuantStudio Design & Analysis Software v1.5.2

Statistical analyses were carried out using Prism 10 (GraphPad Software Inc.)

Molecular Dymanics:
homology model of wild-type human TLR7 active dimer complex was generated using MODELLER (version 10.4)
All simulations were performed using the GPU-accelerated GROMACS software package (version 2023.1 and version 2021 with CpHMD module)
Autodock VINA and Autodocktools were also used for docking predictions (see methods).

For RNA sequencing analyses:
Dragen BCLConvert (v3.7.4) for base calling
R (v4.1.0) data analysis was conducted with the following packages:
scPipe package (v1.14.0) read processing, demultiplexing, gene counting
Rsubread package (v2.6.1) read alignment
biomaRt package (v2.48.3) gene annotation
edgeR package (v3.34.0) count filtering, normalisation, linear model fitting

For manuscripts utilizing custom algorithms or software that are central to the research but not yet described in published literature, software must be made available to editors and reviewers. We strongly encourage code deposition in a community repository (e.g. GitHub). See the Nature Portfolio guidelines for submitting code & software for further information.

# Data

Policy information about availability of data

All manuscripts must include a data availability statement. This statement should provide the following information, where applicable:
- Accession codes, unique identifiers, or web links for publicly available datasets
- A description of any restrictions on data availability
- For clinical datasets or third party data, please ensure that the statement adheres to our policy

RNA sequencing data has been deposited in the NCBI Gene Expression Omnibus (GEO) with accession GSE291606.
The cryo-EM maps have been deposited at the Electron Microscopy Data Bank under the following accession codes: EMD-60515 (TLR7/GUC-v1PS complex), EMD-60541 (TLR7/mGrArAPS complex) and EMD-63406 (TLR7/mGrUrCPO complex). The coordinates of the atomic models have been deposited at the Protein Data Bank (PDB) under the following accession codes: TLR7/GUC-v1PS-SS (8ZW2), TLR7/GUC-v1PS-RR (8ZW4), TLR7/mGrArAPS-SS (8ZXE), TLR7/mGrArAPS-RR (8ZXF) and TLR7/mGrUrCPO (9LUV).

# Research involving human participants, their data, or biological material

Policy information about studies with human participants or human data. See also policy information about sex, gender (identity/presentation), and sexual orientation and race, ethnicity and racism.

| | |
|---|---|
| Reporting on sex and gender | The PBMC experiments were conducted in 2 males and one female healthy volunteers (40-45 years old) |
| Reporting on race, ethnicity, or other socially relevant groupings | NA |
| Population characteristics | NA |
| Recruitment | The participants provided written informed consent before participation, using the Monash Health Human Research Ethics Committee–approved Participant Information and Consent Form (Protocol RES-18-0000-363A). No financial compensation was provided. |

| Ethics oversight | Monash Health Human Research Ethics Committee approval #RES-18-0000-363A |

Note that full information on the approval of the study protocol must also be provided in the manuscript.

# Field-specific reporting

Please select the one below that is the best fit for your research. If you are not sure, read the appropriate sections before making your selection.

☒ Life sciences          ☐ Behavioural & social sciences          ☐ Ecological, evolutionary & environmental sciences

For a reference copy of the document with all sections, see nature.com/documents/nr-reporting-summary-flat.pdf

# Life sciences study design

All studies must disclose on these points even when the disclosure is negative.

| Sample size | No statistical methods were used to pre-determine sample sizes but our sample sizes are similar to those reported in previous publications. For Aldara experiments, we used 7 mice for experimental groups and 3 mice for non-treated controls which had previously found was sufficient to achieve statistical power in aldara-treated mice (PMID: 19380832). For R848 systemic injection studies in Fig 5A, we used 4/5 mice per treatment group as we anticipated it was the minimum required for these analyses to reach significance based on literature (PMID: 11812998). For LNP injections in Figure 5E/F/G, we used 5 mice per group as as we anticipated it was the minimum required to reach significance based on literature (PMID: 35332327). |

| Data exclusions | For Flt3L-DC data, IFNa was only significantly detected in 2 out of 3 mice (however TNFa was detected in 3/3 mice). For PBMCs, IFNg levels from one donor saturated the assay and were omitted in the calculations of the averages. |

| Replication | In vitro experiments were all reliably reproduced a minimum of two independent times except for the DNA trimer screens shown in Table S1 which were only conducted once due to limited activities seen. The aldara in vivo studies were independently replicated more than 2 times. Similarly the R848 systemic challenge was independently confirmed with another TLR7 inhibiting 3-base oligonucleotide. The LNP in vivo studies were only conducted once as robust significance was reached with 5 animals in each group. |

| Randomization | Mice were randomly allocated to their group for all in vivo studies. For preparation of bone marrow derived macrophages and DCs from Kika mutant mice or WT mice, mice from the same genotype and same age were used. |

| Blinding | Data collection and analysis were not performed blind to the conditions of the experiments unless otherwise stated in the methods. Mice studies: For Aldara studies, the treatments were not blinded but scoring of the mice was conducted blinded. For analyses of sera from LNP studies, the treatment and analyses of LNP with FLuc or FLuc+GGCv1 were conducted blinded. For the systemic R848 challenge experiments, the treatments were not blinded but similarly, analyses of the TNF and RTqPCRs in collected samples were conducted blinded. Cells studies: All oligonucleotide screens were performed blinded (with no knowledge of the sequences used). |

# Reporting for specific materials, systems and methods

We require information from authors about some types of materials, experimental systems and methods used in many studies. Here, indicate whether each material, system or method listed is relevant to your study. If you are not sure if a list item applies to your research, read the appropriate section before selecting a response.

## Materials & experimental systems

| n/a | Involved in the study |
|---|---|
| ☐ | ☒ Antibodies |
| ☐ | ☒ Eukaryotic cell lines |
| ☒ | ☐ Palaeontology and archaeology |
| ☐ | ☒ Animals and other organisms |
| ☒ | ☐ Clinical data |
| ☒ | ☐ Dual use research of concern |
| ☒ | ☐ Plants |

## Methods

| n/a | Involved in the study |
|---|---|
| ☒ | ☐ ChIP-seq |
| ☒ | ☐ Flow cytometry |
| ☒ | ☐ MRI-based neuroimaging |

## Antibodies

| Antibodies used | We used:<br>1. CD45 (D3F8Q) antibody (#70257S Cell signalling Technology - Lot #4) (1:200),<br>2. Anti-Fibrillarin (38F3) antibody (Abcam ab4566 Lot 1088391-1) (1:500), |

3. Anti-beta Actin (SP124) (Abcam ab8227 - Lot 1103556-1) (1:10,000),
4. Anti-mouse secondary (Abcam ab205719 Lot 1036603-15) (1:5,000),
5. Goat anti-rabbit secondary (Sigma A0545 Lot 069M4835V).(1:10,000),

Cytometric bead arrays (BD Biosciences) - as recommended by manufacturer
6. IFNA #560379 (lot 3117527),
7. TNF#560112 (lot 5013222),
8. IL12p70#558283 (lot 5121732),
9. IFNG # 558269 (lot 5031631).

| Validation | Antibodies were validated by manufacturer as follows:<br>1.https://www.cellsignal.com/products/primary-antibodies/cd45-d3f8q-rabbit-monoclonal-antibody/70257<br>2.https://www.abcam.com/en-us/products/primary-antibodies/fibrillarin-antibody-38f3-nucleolar-marker-ab4566<br>3.https://www.abcam.com/en-us/products/primary-antibodies/beta-actin-antibody-loading-control-ab8227<br>4.https://www.abcam.com/en-us/products/secondary-antibodies/goat-mouse-igg-h-l-hrp-ab205719<br>5.https://www.sigmaaldrich.com/AU/en/product/sigma/a0545<br>6.https://www.bdbiosciences.com/en-au/products/reagents/immunoassay-reagents/cba/cba-kits/human-ifn-flex-set.560379<br>7.https://www.bdbiosciences.com/en-au/products/reagents/immunoassay-reagents/cba/cba-kits/human-tnf-flex-set.560112<br>8.https://www.bdbiosciences.com/en-au/products/reagents/immunoassay-reagents/cba/cba-kits/human-il-12p70-flex-set.558283<br>9.https://www.bdbiosciences.com/en-au/products/reagents/immunoassay-reagents/cba/cba-kits/human-ifn-flex-set.558269 |
|---|---|

# Eukaryotic cell lines

Policy information about cell lines and Sex and Gender in Research

| Cell line source(s) | HEK-Blue™ IFN-α/β Cells (Invivogen #hkb-ifnabv2-b) (used to make TLR7/8 mutant cells), 293XL-hTLR7 (#293xl-htlr7), 293XL-hTLR8 (293xl-htlr8), 293XL-hTLR9-HA (#293xl-htlr9ha), HEK-Blue™ hTLR3 (#hkb-htlr3) and HEK-Blue™ mTLR13 (#hkb-mtlr13) were purchased from Invivogen.<br>RAW ELAM macrophages were reported in PMID: 11686851<br>WT THP-1 cells were reported in PMID: 34057477<br>HEK-293T cells were published in PMID: 23722158<br>HipSci HPSI0114i-kolf_2 were from Sanger Institute<br>HeLa cells were purchased from ATCC #CCL-2<br>TLR4 KO iBMDMs were a gift from E. Latz |
|---|---|
| Authentication | None of the cells lines were authenticated. |
| Mycoplasma contamination | All cell lines tested negative for mycoplasma contamination using Mycostrip (Invivogen). |
| Commonly misidentified lines<br>(See ICLAC register) | HeLa cells - these were purchased from ATCC - and they were only used to obtain RNA as a modulator of TLR7/8 sensing |

# Animals and other research organisms

Policy information about studies involving animals; ARRIVE guidelines recommended for reporting animal research, and Sex and Gender in Research

| Laboratory animals | Mice were housed in an specific pathogen free vivarium and fed normal chow ad libitum, mice experienced a 12 hour light/dark cycle with temperature ~22oC and humidity of ~40%.<br>1. 24x 8-week-old wild type C57Bl/6J female mice (Monash Animal Research Platform) were used for Aldara model experiments.<br>2. 3x 12-week-old wild type C57Bl/6J males (Monash Animal Research Platform) were used for Flt3L-derived DC preparations.<br>3. 12x 8-week-old female C57BL/6NCrl mice (ANU) were used for systemic R848 experiments.<br>4. 13x 8-week-old 129X1/SvJ female mice (JAX Lab:000691) were used for LNP experiments.<br>5. 3x 9-11-week old TLR7(WT/Kika)C57BL/6NCrl mutant female mice (ANU) were used for BMM purifications. |
|---|---|
| Wild animals | No wild animals were used in these studies. |
| Reporting on sex | TLR7 is expressed on Chromosome X and is linked to heightened auto-immunity in females; For this reason we selectively used female mice in all our experimentations. |
| Field-collected samples | No field collected samples were used in these studies. |
| Ethics oversight | 1. LNP injection studies were approved by Institutional Animal Care and Use Committee reference IN020-08202020-27736.<br>2. Systemic R848 injections in C57BL/6NCrl were approved by Australian National University animal ethics, reference A2022/18.<br>3. Aldara driven skin inflammation studies were approved in advance by an Animal Ethics Committee at Monash Medical Centre (MMCB/2022/18 and MMCB/2023/19).<br>4. BMMs were collected from Tlr7Y264H C57BL/6NCrl mice (used under Australian National University animal ethics, reference A2021/29).<br>5. BMMs were collected from C57Bl/6J males for Flt3L-derived DC preparations (under Monash Medical Centre B Animal Ethics Committee reference MMCB/2024/30) |

Note that full information on the approval of the study protocol must also be provided in the manuscript.

## Plants

| | |
|---|---|
| Seed stocks | NA |
| Novel plant genotypes | NA |
| Authentication | NA |

