## [Peer Review File · Nature Immunology]

2'-O-Methyl-guanosine RNA fragments mediate essential natural TLR7/8 antagonism

Corresponding Author: Professor Michael Gantier

Version 0:

Decision Letter:

10th Jul 2025

Dear Dr Gantier,

Your Article, "2'-O-Methyl-guanosine RNA fragments mediate essential natural TLR7/8 antagonism" has now been seen by 2 referees. You will see from their comments copied below that while they find your work of considerable potential interest, they have raised substantial concerns.

In light of these comments, we cannot accept the manuscript for publication, but would be very interested in considering a revised version that addresses these serious concerns.

We do expect you to address reviewer 1 comments in full experimentally, but given the difficulty of addressing reviewer 2 main concern experimentally we will be satisfied if you tone down conclusions to ensure nothing is overstated and insert caveats and a clear discussion of the limitations of your methodology.

If you choose to revise your manuscript taking into account all reviewer and editor comments, please highlight all changes in the manuscript text file in Microsoft Word format.

* If you have not done so already please begin to revise your manuscript so that it conforms to our Article format instructions at <http://www.nature.com/ni/authors/index.html>. Refer also to any guidelines provided in this letter.

The Reporting Summary can be found here:

When submitting the revised version of your manuscript, please pay close attention to our [href="https://www.nature.com/nature-portfolio/editorial-policies/image-integrity">Digital Image Integrity Guidelines. and to the following points below:](https://www.nature.com/nature-portfolio/editorial-policies/image-integrity)

Extended Data figures and tables are online-only (appearing in the online PDF and full-text HTML version of the paper), peer-reviewed display items that provide essential background to the Article but are not included in the printed version of the paper due to space constraints or being of interest only to a few specialists. A maximum of ten Extended Data display items

(figures and tables) is typically permitted. When re-submitting your manuscript, please ensure that any supplementary figures and tables that are more critical to the manuscript's conclusions are converted to Extended data to increase these data's visibility.

Link Redacted

If you wish to submit a suitably revised manuscript we would hope to receive it within 6 months. If you cannot send it within this time, please let us know. We will be happy to consider your revision so long as nothing similar has been accepted for publication at Nature Immunology or published elsewhere.

Nature Immunology is committed to improving transparency in authorship. As part of our efforts in this direction, we are now requesting that all authors identified as 'corresponding author' on published papers create and link their Open Researcher and Contributor Identifier (ORCID) with their account on the Manuscript Tracking System (MTS), prior to acceptance. ORCID helps the scientific community achieve unambiguous attribution of all scholarly contributions. You can create and link your ORCID from the home page of the MTS by clicking on 'Modify my Springer Nature account'. For more information please visit please visit www.springernature.com/orcid.

Thank you for the opportunity to review your work.

Sincerely,

Nick Bernard, PhD
Senior Editor
Nature Immunology

Reviewers' Comments:

Reviewer #1 (Remarks to the Author):

In their manuscript "2'-O-Methyl-guanosine RNA fragments mediate essential natural TLR7/8 antagonism", Alharbi et al. report 2'-O-methylated trinucleotide RNA molecules that modulate TLR7 and TLR8 activity. They report sequence- and modification requirements for the inhibition of TLR7 and TLR8. On the other hand they describe modified trinucleotides that augment TLR8 activity. They identified a novel inhibitory binding site in TLR7 and TLR8 that is distinct from the known binding sites for nucleosides/small molecule agonists and non-modified trinucleotide RNAs. The fact that genetic variants at this identified binding site (F506 and F507 gain of function mutations) are involved in systemic lupus erythematosus (SLE) support the author's concept that 2'-O-methylated self-RNA constantly binds to this inhibitory binding site thereby preventing TLR7-mediated autoinflammation. The authors describe a number of inhibitory trinucleotides that potently inhibit the response to TLR7/8 agonists in vitro in human and murine cell models, and in a mouse model in vivo. Together, these findings are highly interesting and provide new insight in the physiological control of TLR7/8 activity. The new inhibitory binding site of TLR7/8 could potentially serve as target for novel treatments of TLR7-dependent autoimmunity, with the described inhibitory trinucleotides as promising candidates.

However, most of the data rely on small molecule agonists, HEK cell models and macrophages. To understand the full potential of this immune modulatory approach and the biological role of the inhibitory site, actual RNA agonists should be investigated more comprehensively, and primary human immune cells, most importantly plasmacytoid dendritic cells for TLR7 and monocytes for TLR8, should be used to confirm the functional relevance of the results obtained with small molecule ligands and cell lines.

Specific comments:

1. Currently, the assays used mainly focus on various HEK-TLR cell lines and the macrophage cell lines THP-1 and RAW264.7. A small subset of experiments is performed in iPSC-derived macrophages. However, the prototypical TLR7-responsive cells in humans are plasmacytoid dendritic cells (pDC) and B cells. Therefore it is important to demonstrate that the immunomodulatory trimer-oligonucleotides also function on primary human PBMC to inhibit or enhance the TLR7-dependent IFN- (pDC) and TLR8-dependent TNF or IL-12p70 (monocytes) response. Flt3L-bone marrow derived DC could

be used as a reliable source of murine pDC .

2. As discussed by the authors, from the data is not entirely clear whether the effects of the different oligos screened are mediated by the agonistic trinucleotide binding site or by the novel antagonistic binding site. Furthermore, additional TLR independent activities can not be ruled out by the experimental settings used. In human PBMC and murine Flt3L-DC, TLR9 agonists can be used to control for TLR7/8 specificity of the immunomodulatory trimer-oligonucleotides. The use of TLR9 agonists allows to distinguish TLR7/8 specific effects from other effects such as a general disturbance of endosomal nucleic acid metabolism, TLR processing and TLR signaling.

3. Most screening experiments and the in vivo model in this work investigate the modulation of TLR7/8 responses upon stimulation by small molecule agonists. Only a small subset of experiments investigates natural RNA ligands such as ssRNA40 and RNA9.2s. In some ways the reasons for the use of small molecule agonists in this study is unclear (e.g. Fig S1A, TLR8 is stimulated with uridine but TLR7 with R848). One way to interpret the data presented is that the inhibitory site mainly functions to prevent autoactivation by site 1-ligands like uridine alone, thus stronger site 1 small molecule ligands would be more susceptible to inhibition. Of note, in general HEK cells are easier to activate with small molecule agonists than with RNA. Whether such studies are representative for the natural TLR7/8 responses in pDC and monocytes is remains unclear. Thus, it is important to assess the potency of all investigated immunomodulatory trinucleotides in human PBMC using TLR7/8 agonistic RNAs ssRNA40 and RNA9.2s, and ideally natural TLR7/8 agonistic RNAs, e.g. E.coli totalRNA.

4. The authors screen large panels of different oligonucleotides. For better clarity and readability, tables summarizing the effects of different oligos on different TLRs and cell lines should be provided. Such tables would provide an overview which sequences inhibit or enhance TLR7 or TLR8 or both, and whether those sequences have the same effects across HEK cells, macrophages, monocytes and plasmacytoid dendritic cells, as well as in mouse and human.

5. Inhibitory (longer) oligonucleotides with 2'-O-methyl modifications have been described before. The authors should provide information whether the optimal trinucleotides identified here are also contained in previously described antagonistic RNA oligonucleotides. Such previously identified antagonistic RNA oligonucleotides could serve as "prodrugs" that are cleaved into inhibitory short trimers described in this work.

Reviewer #2 (Remarks to the Author):

This manuscript addresses a longstanding immunological question. Why do Toll-like receptors 7 and 8, which are essential for sensing pathogenic RNA, not respond to self RNA? In fact, they may even become inhibited by it. The authors propose a model in which 2'-O-methyl guanosine-modified RNA fragments derived from host ribosomal RNA act as natural TLR7/8 antagonists. This prevents unwarranted immune activation. Through a combination of biochemical assays and structural analyses, the authors propose that these modified RNAs bind to a distinct inhibitory site on TLR7/8, which was previously identified as a negative allosteric regulatory pocket occupied by inhibitory small molecules. The authors refer to this site as a physiological immune checkpoint. The authors argue that mutations in this site disrupt inhibition and are associated with autoimmune phenotypes.

This study offers an intriguing concept for how endogenous RNA might subvert TLR: host-derived, modified RNA fragments actively suppress TLR7/8 signaling. Part of this concept is supported by robust structural data, including cryo-EM analysis demonstrating the occupancy of a third binding pocket on TLR7/8 by 2'-OMe-G-initiated oligonucleotides. These insights provide a compelling mechanistic framework for the idea that RNA modifications can mediate immune tolerance to self-RNA.

Despite the strength of some of the underlying data and the conceptual appeal of the idea, the manuscript clearly overstates key aspects.

Major:

The authors demonstrate that 2'-O-methyl guanosine 3-mer oligonucleotides primarily block TLR activation using phosphorothioate oligonucleotides that do not exist in nature. In contrast, physiologically relevant phosphodiester oligonucleotides exhibited weaker suppression of TLR responses. Furthermore, the authors never demonstrate that 2'-O-methyl guanosine oligonucleotides are produced in cells under steady-state conditions or in the context of efferocytosis. Instead, the manuscript shifts its focus to the single nucleoside 2'-OMe-G, while the physiological relevance of this molecule also remains uncertain. The concentrations required for partial inhibition in vitro (0.5 mM) are substantially higher than what one would expect from endogenous rRNA methylation levels, especially since only a small fraction of guanosines in rRNA are methylated. To this end, it remains speculative whether inhibitory RNA fragments do exist in endolysosomal compartments.

Minor:

The results section, especially the first half, is overly complex. I recommend focusing on the paper's key aspect: the inhibition of RNA-sensing TLRs by 2'-O-methyl guanosine-containing RNA.

The discussion is too long and speculative. I recommend that the authors tone down their central conclusions and more clearly delineate what has been demonstrated versus what has been hypothesized.

Version 1:

Decision Letter:

Our ref: NI-A40568A

24th Nov 2025

Dear Dr. Gantier,

Thank you for submitting your revised manuscript "2'-O-Methyl-guanosine RNA fragments mediate essential natural TLR7/8 antagonism" (NI-A40568A). It has now been seen by the original referees and their comments are below. The reviewers find that the paper has improved in revision, and therefore we'll be happy in principle to publish it in Nature Immunology, pending minor revisions to satisfy the referees' final requests and to comply with our editorial and formatting guidelines.

We will now perform detailed checks on your paper and will send you a checklist detailing our editorial and formatting requirements in about a week. Please do not upload the final materials and make any revisions until you receive this additional information from us.

If you had not uploaded a Word file for the current version of the manuscript, we will need one before beginning the editing process; please email that to immunology@us.nature.com at your earliest convenience.

Thank you again for your interest in Nature Immunology Please do not hesitate to contact me if you have any questions.

Sincerely,

Nick Bernard, PhD
Senior Editor
Nature Immunology

Reviewer #1 (Remarks to the Author):

The authors have performed the additional experiments recommended by the reviewer, and have appropriately addressed all concerns raised by this reviewer.

Reviewer #2 (Remarks to the Author):

The authors addressed my concerns. I have no further comments.

Response to Reviewers – Manuscript NI-A40568

We sincerely thank the Editor and Reviewers for their assessment of our work. The Reviewer comments are presented in **bold**, while our responses are in *blue italics*. Note that edits of our manuscript corresponding to new results/data/text have been highlighted in **yellow**, and text changes made to streamline the paper have been highlighted in **green**.

Editor:

We do expect you to address reviewer 1 comments in full experimentally, but given the difficulty of addressing reviewer 2's main concern experimentally we will be satisfied if you tone down conclusions to ensure nothing is overstated and insert caveats and a clear discussion of the limitations of your methodology.

We thank the editor for their comment. Accordingly, our revised manuscript incorporates several new experiments addressing all the points raised by reviewer #1. We have also halved our discussion and clearly emphasised the limitations of our study raised by reviewer #2 and future research directions.

Reviewer #1:

1. Currently, the assays used mainly focus on various HEK-TLR cell lines and the macrophage cell lines THP-1 and RAW264.7. A small subset of experiments is performed in iPSC-derived macrophages. However, the prototypical TLR7-responsive cells in humans are plasmacytoid dendritic cells (pDC) and B cells. Therefore it is important to demonstrate that the immunomodulatory trimer-oligonucleotides also function on primary human PBMC to inhibit or enhance the TLR7-dependent IFN- α (pDC) and TLR8-dependent TNF or IL-12p70 (monocytes) response. Flt3L-bone marrow derived DC could be used as a reliable source of murine pDC.

We thank the reviewer for their valuable suggestion. Our revised manuscript now incorporates new analyses of 30 immunomodulatory trinucleotides (selected from Table S1) on TLR7/8 sensing stimulated by transfected phosphodiester RNA9.2s and ssRNA40 RNAs in human PBMCs (Extended Figure S2H, S2I, and Table S2). We compared levels of human IFN- α with TNF (Figure S2H and S2I), and assessed IL-12p70 and IFN- γ (Table S2). Notably, the inhibitory activities of the 3-mers were remarkably similar for both ssRNAs, consistent with the dual activity of these molecules on TLR7 and TLR8 (PMID: 32294405).

As requested, we also analysed IFN- α and TNF levels from Flt3L-derived murine DCs stimulated with RNA9.2s and treated with a panel of 11 immunomodulatory 3-mers (Extended Figure 3F and Table S2).

These analyses in human and mouse primary cells corroborated our previous analyses with synthetic TLR7/8 agonists, while providing a broad characterisation of the immunomodulatory activities of the most potent 3-mers on TLR7/8 sensing in the relevant immune cells.

The results section of our manuscript presents the novel findings as follows:

“To confirm the relevance of our observations on sensing of the transfected TLR7 and TLR8 RNA agonists, RNA9.2s^{PO9} and ssRNA40^{PO16}, we selected the top 30 3-mer oligos modulating

TLR7 and TLR8 and tested their activity in healthy donor peripheral blood mononuclear cells (PBMCs). These analyses confirmed that $mG_mA_mG^{PS}/mG_dA_dG^{PS}$ oligos strongly reduced TLR7 (IFN α) and TLR8 (TNF) sensing (Extended Figures 2h, 2i and Table S2). Notably, $mG_mU_mX/mG_mU_dX^{PS}$ oligos selectively blunted IFN α levels with a minimal effect on TNF levels, highlighting their preferential activity on TLR7. Analysis of TLR8-selective IL-12p70 and IFN γ levels confirmed the more selective inhibitory effect of $mG_dA_dG^{PS}$ and related sequences on TLR8 sensing, which was much weaker for $mG_mU_mC^{PS}/mG_mU_dC^{PS}$ oligos (Table S2). The potentiating effect of $mG_dC_dC^{PS}$ was also confirmed for both RNA ligands, with a preference for TNF, although this was more limited than in the context of pure site 1 agonists, and $mU_mC_mU^{PS}$ was consistently superior across TNF/IL-12p70/IFN γ levels (Extended Figures 2h, 2i and Table S2).”

And

“We also validated the activity of a panel of eleven 3-mers on RNA9.2s^{PO}-sensing by mouse TLR7 using primary bone marrow-derived DCs, which revealed that $mG_mG_dC^{PS}$ and $mG_dG_dC^{PS}$ had the strongest inhibitory effect on TNF production, while also halving IFN α production (Extended Figure 3f and Table S2). Notably, $mG_dA_dG^{PS}$ had the strongest inhibitory effect on IFN α , but not TNF, indicating the 3-mers may have different activities in different Flt3L-derived-DC subsets.”

2. As discussed by the authors, from the data it is not entirely clear whether the effects of the different oligos screened are mediated by the agonistic trinucleotide binding site or by the novel antagonistic binding site. Furthermore, additional TLR independent activities cannot be ruled out by the experimental settings used. In human PBMC and murine Flt3L-DC, TLR9 agonists can be used to control for TLR7/8 specificity of the immunomodulatory trimer-oligonucleotides. The use of TLR9 agonists allows to distinguish TLR7/8 specific effects from other effects such as a general disturbance of endosomal nucleic acid metabolism, TLR processing and TLR signaling.

To test the specificity of our 3-mers, we assessed the effect of our lead 3-mers (30 in human cells and 11 in mouse cells) on TLR9 signalling induced with ODN2216 in human PBMCs, and with ODN1826 in Flt3L-derived cells (Extended Figure 3g and Table S2).

Notably, only IFN- α was significantly detected with ODN2216 in human PBMCs. Conversely, we only detected TNF with ODN1826 in Flt3L-derived cells. None of the 3-mers tested significantly modulated human or mouse TLR9 sensing, confirming that the inhibitory effect seen on TLR7/8 is not due to off-target endosomal disruption. The results section of our manuscript presents the novel findings as follows:

“Analyses of the 30 3-mer oligo panel on TLR9 sensing in PBMCs did not reveal any significant IFN α inhibition by the 3-mers, confirming their selective activity on TLR7/8 over TLR9 (Table S2).”

And

“Similar to what was seen in PBMCs, none of these 3-mers significantly impacted TLR9-driven TNF production in Flt3L-derived-DCs (Extended Figure 3g and Table S2).”

3. Most screening experiments and the in vivo model in this work investigate the modulation of TLR7/8 responses upon stimulation by small molecule agonists. Only a small subset of experiments investigates natural RNA ligands such as ssRNA40 and RNA9.2s. In some ways the reasons for the use of small molecule agonists in this study is unclear (e.g. Fig S1A, TLR8 is stimulated with uridine but TLR7 with R848). One way to interpret the data presented is that the inhibitory site mainly functions to prevent autoactivation by site 1-ligands like uridine alone, thus stronger site 1 small molecule ligands would be more susceptible to inhibition. Of note, in general HEK cells are easier to activate with small molecule agonists than with RNA. Whether such studies are representative for the natural TLR7/8 responses in pDC and monocytes is remains unclear. Thus, it is important to assess the potency of all investigated immunomodulatory trinucleotides in human PBMC using TLR7/8 agonistic RNAs ssRNA40 and RNA9.2s, and ideally natural TLR7/8 agonistic RNAs, e.g. E.coli totalRNA.

This comment is aligned with the first point raised by this reviewer. As per the response in point #1, in this revision we assessed the activity of the 30 (human) and 11 (mouse) lead immunomodulatory 3-mers in primary cells to ssRNA40 and RN9.2s phosphodiester RNAs. These 3-mer panels were chosen because they had the strongest activity on TLR7/8 from previous screens. These experiments demonstrated that the inhibitory activity of our 3-mers on RNA agonists and small molecule agonists was very similar. As such, GUX oligos predominantly blocked TLR7-driven IFN- α , while GAX oligos had a broader effect on TLR8-driven cytokines (TNF, IFN- γ and IL-12p70).

To address the comment about natural TLR7/8 agonistic RNAs, we now present analyses of the effect of three lead mouse TLR7 antagonists on sensing of purified E. coli RNA in TLR4-deficient immortalised macrophages (to exclude potential endotoxin contaminants in the purified RNA). These analyses confirmed the antagonistic effect of the three 3-mers on sensing of purified E. coli RNA (Extended Figure 3h). The results section of our manuscript presents the novel findings as follows:

“In addition, mouse TLR7-sensing of transfected bacterial RNA was also significantly inhibited by GGC-v1^{PS}, mG_dA_dG^{PS} (GAG-v4^{PS}) and GAG-v1^{PS} (Extended Figure 3h).”

4. The authors screen large panels of different oligonucleotides. For better clarity and readability, tables summarizing the effects of different oligos on different TLRs and cell lines should be provided. Such tables would provide an overview which sequences inhibit or enhance TLR7 or TLR8 or both, and whether those sequences have the same effects across HEK cells, macrophages, monocytes and plasmacytoid dendritic cells, as well as in mouse and human.

We thank the reviewer for this comment and have now replaced our initial supplementary Table S1 with two supplementary Tables provided as excel files with tabs. In Supplementary Table S1, we present the data of all our screens in immortalised cell lines. To visually clarify the identification of the strongest modulators of TLR7 and TLR8, we have added colour coding based on a heat map for each screen (where low values are blue and high values are red). We also bolded the most potent 3-mers, which were selected for further testing in primary PBMCs and mouse Flt3L-DCs. These new data are shown in Supplementary Table S2, and similar colour coding was used to highlight the sequences broadly inhibiting or activating TLR7/8.

Notably we have added a sorting button for each column allowing readers to sort these tables according to the desired parameter (e.g., low TLR7 sensing in HEK-TLR7, high TLR8 sensing in HEK-TLR8, etc).

5. Inhibitory (longer) oligonucleotides with 2'-O-methyl modifications have been described before. The authors should provide information whether the optimal trinucleotides identified here are also contained in previously described antagonistic RNA oligonucleotides. Such previously identified antagonistic RNA oligonucleotides could serve as "prodrugs" that are cleaved into inhibitory short trimers described in this work.

This is an interesting point: how do the 2'-OMe motifs identified here perform in longer antagonistic RNAs? We previously discovered that ~20-base PO fully-2'-OMe modified oligonucleotides were inhibiting TLR7/8 sensing in a motif-dependent manner (Sarvestani et al, NAR 2015). As evidenced with miR-224-5p mut2 in that paper, addition of a 5' end mGmUmC increased the inhibition of TLR7, consistent with our present findings. We also note that

*IMO8400 (dC*dT*dA*dT*dC*dT*mG*mU*5-MedC*d(c7G)*dT*dT*dC*dT*dC*dT*mG*mU), which was used in clinical studies of TLR7/8/9 antagonism has a central mGmUC motif and a 3'-end mGmU motif.*

We have added a sentence about miR-224-5p mut2 and IMO8400 in the discussion:

“Interestingly, longer TLR7-inhibiting oligos with the optimal TLR7 mGmUmC/mGmU inhibiting motif have been reported, such as IMO8400/Bazlitoran, which advanced to clinical studies, and miR-224-5p mut2^{23, 42}.”

Reviewer #2:

Major:

The authors demonstrate that 2'-O-methyl guanosine 3-mer oligonucleotides primarily block TLR activation using phosphorothioate oligonucleotides that do not exist in nature. In contrast, physiologically relevant phosphodiester oligonucleotides exhibited weaker suppression of TLR responses.

We thank the reviewer for raising this point. Notably, the lower efficacy of phosphodiester (PO) 2'-OMe oligos on TLR7/8 is consistent with previous reports using longer oligos and most likely related to a stability/rapid degradation issue of PO oligos rather than a defective activity, as hinted by the reviewer's comment. As such, previous works testing the activity of poly-dT oligos on TLR7 inhibition also reported a lower inhibitory effect of PO poly-dT oligos over phosphorothioate (PS) ones (Ref 29). It is noteworthy that we, and others, have also utilised PS-modified ssRNA40 and RNA9.2 agonists and discovered that these molecules exhibit strong TLR7/8 activating capacity (Ref 16). This observation challenges the notion of an off-target inhibitory effect of the PS-modification.

Furthermore, the authors never demonstrate that 2'-O-methyl guanosine oligonucleotides are produced in cells under steady-state conditions or in the context of efferocytosis.

We acknowledge this limitation of our studies, and have now added the following sentence to address this in the discussion:

"However further studies will be required to confirm the unambiguous detection of such partial 2'-OMe rRNA degradation products, which we did not evidence here."

However, we did evidence increased levels of 2'-OMe nucleosides upon transfection of ribosomal RNA, which supports processing into fragments. This has been stressed in the text of the discussion:

"Since 2'-OMe nucleosides originate from nuclease processing of longer fragments, we speculate that rRNA processing generates partial (2-3 bases) and complete degradation fragments containing 2'-OMe guanosine, inhibiting TLR7 sensing."

Instead, the manuscript shifts its focus to the single nucleoside 2'-OMe-G, while the physiological relevance of this molecule also remains uncertain. The concentrations required for partial inhibition in vitro (0.5 mM) are substantially higher than what one would expect from endogenous rRNA methylation levels, especially since only a small fraction of guanosines in rRNA are methylated. To this end, it remains speculative whether inhibitory RNA fragments do exist in endolysosomal compartments.

We acknowledge the high concentration of 2'-OMe guanosine employed in this study. However, it is crucial to emphasise that nucleosides must undergo active intracellular transport via nucleoside transporters (PMID: 29928232). Consequently, the extracellular concentration utilised in the cell medium differs significantly from that reaching the endosome. Furthermore, in accordance with the active transport mechanism, it is noteworthy that higher concentrations of uridine and guanosine are necessary to activate TLR7/8, ranging from 1 mM to 20 mM, as observed in our experiments and previous reports (Refs 5, 16).

Here we successfully show that at an equimolar concentration of for both Guanosine (+ssRNA) and 2'-OMe-Guanosine (both nucleosides used at 500 μ M), TLR7 is significantly antagonised, arguing that this is a biologically relevant observation.

The following sentence has been added to the discussion to address these concerns:

“Critically, 2'-OMe guanosine was sufficient to halve the stimulatory activity of the same concentration of guanosine in the presence of RNA9.2^{PS}, through engagement of the F507 residue.”

Minor:

The results section, especially the first half, is overly complex. I recommend focusing on the paper's key aspect: the inhibition of RNA-sensing TLRs by 2'-O-methyl guanosine-containing RNA.

We thank the reviewer for this suggestion. Accordingly, we have simplified the results description of the first sections of our manuscript relating to the oligonucleotide discoveries – Figs 1 and 2 – while retaining the important data.

The discussion is too long and speculative. I recommend that the authors tone down their central conclusions and more clearly delineate what has been demonstrated versus what has been hypothesized.

We appreciate the reviewer's comment and have removed speculative sections from our discussion, toned down claims, and emphasised our study's limitations and future research needs. As such, we have nearly halved the length of the discussion.